

# The postcranial skeleton of *Cerrejonisuchus improcerus* (Crocodyliformes: Dyrosauridae) and the unusual anatomy of dyrosaurids

Isaure Scavezzoni and Valentin Fischer

Evolution and Diversity Dynamics Lab, University of Liège, Liège, Belgium

## ABSTRACT

Dyrosauridae is a clade of neosuchian crocodyliforms that diversified in terrestrial and aquatic environments across the Cretaceous-Paleogene transition.
The postcranial anatomy of dyrosaurids has long been overlooked, obscuring both their disparity and their locomotive adaptations. Here we thoroughly describe of the postcranial remains of an unusually small dyrosaurid, *Cerrejonisuchus improcerus*, from the middle-late Paleocene Cerrejón Formation of Colombia, and we provide a wealth of new data concerning the postcranial anatomy of the key dyrosaurids: *Congosaurus bequaerti* and *Hyposaurus rogersii*. We identify a series of postcranial autapomorphies in *Cerrejonisuchus improcerus* (an elliptic-shaped odontoid laterally wide, a ulna possessing a double concavity, a fibula bearing a widely flattened proximal end, a pubis showing a large non-triangular distal surface) as well as functionally-important traits such as a relatively long ulna (85% of the humerus' length), short forelimb (83% of hindlimb's length), or thoracic vertebra bearing comparatively large lateral process (with widened parapophysis and diapophysis) along with strongly arched thoracic ribs allowing a more sturdy and cylindrical rib cage. These indicate a more terrestrial lifestyle for *Cerrejonisuchus* compared to the derived members of the clade. We also built a dataset of 187 traits on 27 taxa, that extensively samples the cranial and postcranial architectures of exemplar crocodyliforms. We analyze these data in via Principal Coordinate Analysis (PCoA) to visualize the postcranial morphospace occupation of Dyrosauridae, Thalattosuchia, and Crocodylia. Our data reveal the existence of a distinctive postcranial anatomy for Dyrosauridae that is markedly distinct from that of crocodylians. As a result, modern crocodylians are probably not good functional analog for extinct crocodyliformes. Postcranial data should also be more widely used in phylogenetic and disparity analyses of Crocodyliformes.

## INTRODUCTION

Dyrosauridae is an extinct family of neosuchian crocodyliforms that is first recorded in the Campanian–Maastrichtian Shendi Formation of Sudan (*Salih et al., 2015*). Dyrosaurids

Corresponding author
Isaure Scavezzoni,
isaure.scavezzoni@doct.uliege.be

survived the Cretaceous–Paleogene mass extinction (*Bronzati, Montefeltro & Langer, 2012, 2015; Hastings, Bloch & Jaramillo, 2014; Wilberg, Turner & Brochu, 2019; Jouve & Jalil, 2020*), and disappeared during the Eocene (presumably at the Ypresian–Lutecian boundary) (*Buffetaut, 1978a; Jouve et al., 2006; Jouve, 2007; Martin, Sarr & Hautier, 2019*). The origin of dyrosaurids is placed in Africa (*Barbosa, Kellner & Viana, 2008; Jouve, Bouya & Amaghzaz, 2008; Hastings, Bloch & Jaramillo, 2014; Jouve et al., 2020*), with their apparition dating from the Late Cretaceous (*Hastings, Bloch & Jaramillo, 2014; Jouve et al., 2020*). Dyrosauridae showed several dispersal episodes during the Late Cretaceous, with at least three exchanges with America (*Hastings, Bloch & Jaramillo, 2014; Jouve et al., 2020*).

Dyrosaurid remains have been found both in marine (*Buffetaut, 1976, 1978a; Jouve et al., 2005, 2006; Schwarz, Frey & Martin, 2006; Barbosa, Kellner & Viana, 2008; Jouve et al., 2008; Schwarz-Wings, Frey & Martin, 2009; Shiller, Porras-Muzquiz & Lehman, 2016; Sena et al., 2017; Martin, Sarr & Hautier, 2019; de Souza et al., 2019; Jouve & Jalil, 2020*) and freshwater (*Buffetaut, 1978a; Khosla et al., 2009; Hastings, Bloch & Jaramillo, 2011; Hastings, Bloch & Jaramillo, 2014; de Souza et al., 2019*) deposits, and are usually pictured as large 'crocodiles' (*Buffetaut, 1976, 1978a, 1980; Langston, 1995, Schwarz-Wings, Frey & Martin, 2009*), although some taxa were fairly small (≤3 m) (*Jouve, Bouya & Amaghzaz, 2005; Hastings et al., 2010; Hastings, Bloch & Jaramillo, 2014*). The relative importance of fossils in marine deposits suggests that dyrosaurids mainly thrived in coastal environments (*Troxell, 1925; Buffetaut, 1976; Denton, Dobie & Parris, 1997; Jouve et al., 2006; Schwarz, Frey & Martin, 2006; Jouve et al., 2005; Barbosa, Kellner & Viana, 2008; Jouve et al., 2008, Salih et al., 2015; Shiller, Porras-Muzquiz & Lehman, 2016; Sena et al., 2017; Martin, Sarr & Hautier, 2019; Jouve & Jalil, 2020*); two lineages from South America (*Cerrejonisuchus–Anthracosuchus*, and *Acherontisuchus*) likely inhabited freshwater environments (*Hastings, Bloch & Jaramillo, 2011, 2014; Wilberg, Turner & Brochu, 2019*), whereas other freshwater dyrosaurids have also been found in Asia (*Buffetaut, 1978a*) and India (*Khosla et al., 2009*). Yet, 'marine dyrosaurids' may have actually inhabited both environments during their lifespan, with youngs living in freshwater environments and adults transitioning to the marine realm (e.g. like *Dyrosaurus*, see *Jouve et al. (2008)*) as few juveniles have been found along adults in marine deposits (*Denton, Dobie & Parris, 1997; Jouve, Bouya & Amaghzaz, 2005; Jouve et al., 2006, 2008; Schwarz, Frey & Martin, 2006; Hastings et al., 2010; Hastings, Bloch & Jaramillo, 2011, 2014; Salih et al., 2015; Shiller, Porras-Muzquiz & Lehman, 2016; Sena et al., 2017; Martin, Sarr & Hautier, 2019*). This environmental tolerance of dyrosaurids presumably helped them cross the Cretaceous–Paleogene crisis (*Jouve et al., 2008; Hastings, Bloch & Jaramillo, 2014; Jouve et al., 2020*).

Marine dyrosaurids have been considered both shore dwellers (*Buffetaut, 1976*) or, on the contrary, as open-sea residents spending only short periods of time on the coast (*Denton, Dobie & Parris, 1997*). In parallel, dyrosaurids have been interpreted as pursuit hunters (*Buffetaut, 1979*) and ambush predators (*Denton, Dobie & Parris, 1997; Hua, 2003*), but (*Schwarz-Wings, 2014*) suggested that the shallow waters, from either marine or freshwater environments, would actually enable a wide range of possible feeding behaviors. Yet, the peculiar freshwater dyrosaurids *Anthracosuchus* (e.g. short and wide snout,
with widely spread orbits (*Hastings, Bloch & Jaramillo, 2014*)) and *Cerrejonisuchus* (e.g. small-bodied and short snouted (*Hastings et al., 2010*)) presumably showed a slightly different spectrum of feeding strategies (*Hastings, Bloch & Jaramillo, 2014*). Several factors helped infer the habitat of fossil dyrosaurids, notably the study of their surrounding deposits (i.e. nature of sediments, plus fossil flora and fauna) (*Denton, Dobie & Parris, 1997*; *Jouve & Schwarz, 2004*; *Jouve et al., 2005*, *2006*; *Barbosa, Kellner & Viana, 2008*; *Jouve et al., 2008*; *Hastings et al., 2010*; *Hastings, Bloch & Jaramillo, 2011*, *2014*; *Salih et al., 2015*; *Shiller, Porras-Muzquiz & Lehman, 2016*; *Sena et al., 2017*; *Martin, Sarr & Hautier, 2019*), as well as cranial anatomy (e.g. relative position of the orbits, elongated snout (*Troxell, 1925*), complex inner ear (*Schwarz-Wings, 2014*)) and a couple of postcranial features (e.g. presence of gastroliths (*Denton, Dobie & Parris, 1997*; *Martin, Sarr & Hautier, 2019*), light osteodermal shield, long and laterally flat tail, amphicoelous vertebrae (*Troxell, 1925*)).

Dyrosaurids have a long history of detailed craniodental studies (*Denton, Dobie & Parris, 1997*; *Jouve, 2007*; *Barbosa, Kellner & Viana, 2008*; *Hastings et al., 2010*; *Martin, Sarr & Hautier, 2019*). However, their postcrania was believed to be undiagnostic (*Buffetaut, 1976*, *Buffetaut, 1978c*, *Buffetaut, 1980*; *Parris, 1986*; *Storrs, 1986*; *Norell & Storrs, 1989*; *Denton, Dobie & Parris, 1997*), and thus was often overlooked in anatomical descriptions and diagnoses (*Langston, 1995*; *Godoy et al., 2016*; *de Souza et al., 2019*). This, in turn, hampers a thorough assessment of the ecological diversity of the group as a whole, which likely colonized several niches (i.e. marine, freshwater, terrestrial) (*Hastings, Bloch & Jaramillo, 2014*; *Wilberg, Turner & Brochu, 2019*). More recently, postcranial remains have started to show their importance in dyrosaurids (*Langston, 1995*; *Jouve & Schwarz, 2004*; *Schwarz, Frey & Martin, 2006*; *Schwarz-Wings, Frey & Martin, 2009*; *Martin, Sarr & Hautier, 2019*; *de Souza et al., 2019*; *Jouve & Jalil, 2020*; *Jouve et al., 2020*), notably for systematics (*Langston, 1995*; *de Souza et al., 2019*; *Jouve & Jalil, 2020*; *Jouve et al., 2020*) and ecological inferences (*Jouve & Schwarz, 2004*; *Schwarz, Frey & Martin, 2006*; *Schwarz-Wings, Frey & Martin, 2009*; *Martin, Sarr & Hautier, 2019*).

Here, we thoroughly describe the postcranial anatomy of the early dyrosaurid *Cerrejonisuchus improcerus* (*Hastings et al., 2010*) from the middle–late Paleocene of Colombia, and provide a series of novel anatomical observations on *Congosaurus bequaerti* and *Hyposaurus natator*. We use these new data to discuss the regionalization of the axial skeleton in Dyrosauridae, and to try to infer the possible lifestyle of *Cerrejonisuchus improcerus*. We also built a morphological dataset containing 187 traits that describe the cranial and postcranial anatomy of 23 selected taxa (27 specimens) in Dyrosauridae, Thalattosuchia, and Crocodylia. Our multivariate analyses of this dataset reveal that dyrosaurids have a distinctive postcranial anatomy and that the morphological signal in craniodental and postcranial regions are concordant, confirming that postcranial characters should make their way in phylogenetical analyses.

Extant crocodylians are naturally used for morphological comparison when it comes to fossil crocodyliforms (*Langston, 1995*; *Denton, Dobie & Parris, 1997*; *Hua, 2003*; *Jouve, Bouya & Amaghzaz, 2005*; *Jouve et al., 2006*; *Schwarz, Frey & Martin, 2006*; *Schwarz-Wings, Frey & Martin, 2009*; *Hastings et al., 2010*; *Hastings, Bloch & Jaramillo, 2011*, *2014*;

*Schwarz-Wings, 2014*; *Martin, Sarr & Hautier, 2019*) or other archosaurs (*Maidment & Barrett, 2011*; *Liparini & Schultz, 2013*; *Voegele et al., 2020*), and also serve as behavioral and functional archetypes (e.g. *Hua & Buffetaut (1997)*; *Jouve et al. (2008)*; *Schwarz-Wings, Frey & Martin (2009)*; *Schwarz-Wings (2014)*). Indeed, the completeness of their remains, and the extended studies of their anatomy and behaviors (e.g. *Gans & Clark (1976)*; *Cong et al. (1998)*; *Reilly & Elias (1998)*; *Farmer & Carrier (2000) Farmer et al. (2000)*; *Uriona & Farmer (2006, 2008)*; *Claessens & Vickaryous (2012)*; *Munns et al. (2012)*; *Allen et al. (2014)*; *Tsai & Holliday (2014)*; *Hutchinson et al. (2019)*; *Drumheller & Wilberg (2020)*), undoubtedly make extant crocodylians an ideal model to understand fossil remains (especially since the extant crocodylian *bauplan* has been around since at least the Early Jurassic (*Stockdale & Benton, 2021*)). Hence, we compare *Cerrejonisuchus* with several crocodylians, as one of our goals is to assess the presumed dyrosaurids–extant crocodylians resemblance.

## MATERIALS AND METHODS

*Cerrejonisuchus improcerus* comprises four different individuals bearing distinct inventory numbers, which are stored at the Florida Museum of Natural History, University of Florida (**UF**) (*Hastings et al., 2010*):

- **UF/IGM 29**, the *holotype*, a nearly complete skull;
- **UF/IGM 30**, a *referred specimen*, a lower jaw (dentaries, splenials and 11 teeth);
- **UF/IGM 31**, a *referred specimen*, comprises a nearly complete skull and several postcranial elements (left humerus, ulna, left femur, fibula, tibia, left and right pubes, 17 vertebrae, one rib, eight osteoderms);
- **UF/IGM 32**, a *referred specimen*, a partial skull (complete snout up to the anterior portion of orbital region).

The specimen **UF/IGM 31** is the most interesting one as it is the only one possessing postcranial material. The skull of this specimen will not be redescribed here has the skull of the holotype (UF/IGM 29) has already been given an extensive description by *Hastings et al. (2010)*. Here, we compare *Cerrejonisuchus* with *Mecistops cataphractus*, and several other crocodylians. We specifically chose *Mecistops cataphractus* as it represents one of the most aquatic crocodylians (O. Pauwels, 2018, personal communications); according to *Schwarz-Wings (2014)*, *Mecistops caraphractus* also falls into the same range of sizes as most dyrosaurids. Moreover, we wanted not only to analyze the overall crocodylian morphology in relation to other fossil crocodyliforms (Dyrosauridae and Thalattosuchia), but also to assess their suitability as functional or ecological models.

In order to better understand the postcranial anatomy of dyrosaurids, and test its uniqueness, we gathered a series of measurements on 40 specimens of crocodyliforms, representing 23 taxa/OTU's (list see also Table SI 1, 3 in Supplemental Information; Table SI 1 represents the list of specimens possessing both cranial and postcranial remains which where used in the main PCoA analysis, whereas Table SI 3 represents the list of dyrosaurid specimens which were used in the postcranial restricted PCoA and thus did not

necessarily have cranial remains). We used digital calipers to record most of the measurements (approximate error of 0.1 mm) on several relatively complete crocodyliforms (see Supplemental Information for the lists of individuals).

We built an SQLite database to manage our morphological data. We used our measurements to create a morphological dataset containing 187 traits as dimensionless ratios (113 length ratios and 74 area ratios), scored for 27 specimens (see Table SI 1 in Supplemental Information; the dyrosaurid postcranial dataset totals 40 specimens and 170 traits, see Table SI 3 in Supplemental Information). All analyses were then conducted in the R statistical environment (v 3.5.1) using the following packages (see Table SI 4 in Supplemental Information): ape (*Paradis, Claude & Strimmer, 2004*), vegan (*Oksanen et al., 2019*), psych (*Revelle, 2019*), dendextend (*Galili, 2015*), ggdendro (*de Vries, 2016*), pvclust (*Suzuki & Shimodaira, 2006*), DBI (*Wickham & Müller, 2019*) (with RSQLite), and ggplot2 (*Wickham et al., 2020*). The morphological dataset of 187 characters (ratios, of which 9.83% are skull ratios) and 27 specimens (taxa) initially contained 70.17% missing data. This value is reduced to 33.44 % after the application of a completeness threshold of 40% for specimens and 30% for morphological features. These thresholds ensure the establishment of a relatively complete distance matrix and hence prevents non-comparability issues and morphospace distortion by highly incomplete specimens. At this stage, the proportion of skull ratios reaches 15.15% of the whole dataset (10 out of 65 characters in total). We then scaled the data so that each morphological feature has a variance of one and a mean of zero (z-transform). This scaled dataset was then used to compute the distance matrix using Euclidean distances. The R-scripts are all available in the Supplemental Information.

The non-negligible amount of missing data prevents the use of PCA/pPCA. We thus subjected this distance matrix to two ordination methods: cluster dendrograms and a principal coordinate analysis (PCoA) (see "Morphospace Occupation"). For the cluster dendrograms, we employed the pvclust function from the pvclust package from *Suzuki & Shimodaira (2006)*, which is hierarchical agglomerative approach of the cluster analysis. For the clustering criterion located within the hclust function, we chose the argument 'ward.D'. The pvclust function uses columns from the dataset (here our distance matrix) to form a hierarchical cluster, and provides *p*-values to show the degree of support from the data each cluster possesses (so high *p*-values indicate highly supported branches). This hierarchical clustering works with multiscale bootstrap resampling. Indeed, this clustering approach constitutes several subsets differing in size from that of the original dataset, ranging from 0.5 to ten times the size of our original dataset, with 0.5 increments.

We assessed the morphological differences between dyrosaurids, crocodylians and thalattosuchians using Permanova (Permutational multivariate analysis of variance, non-parametric) (*Anderson, 2001*). The distance matrix was set as the dependent variable, and taxonomy served as the independent variable. We also assessed the existence of significant differences between the different ecomorphological groups. For this, the distance matrix was again set as the dependent variable, while the three main morphological clusters served as the independent variable. In each case, we set the

number of permutations to 1,000. The $p$-value we obtained for both results was significant ($p < 0.01$).

The distance matrix was then subjected to a PCoA ( ape package) to analyze patterns of morphospace occupation. We used the 'cailliez' correction for negative eigenvalues; this correction method simply adds a constant to each value of the distance matrix (except the diagonal ones). We also visualized the strength of the ties between cranial and postcranial characters using the tanglegram function from the dendextend R package. The tanglegram was drawn over the clusters obtained from cranial and postcranial limited matrices (respectively possessing 25 and 170 columns of characters initially). The datasets were both subjected to a slightly less stringent completeness threshold of 20% for their characters and 30% for their specimens, thus reducing the amount of missing data to 28% for the skull dataset and 43% for the postcranial dataset, in order to maximize the number of taxa compared to one another.

## AXIAL SKELETON ANATOMY

### General information and morphological conventions

*Cerrejonisuchus improcerus* Hastings et al. (2010) is a dyrosaurid crocodyliform, ranging from the middle-late Paleocene of Colombia. Phylogenetic analyses placed *Cerrejonisuchus improcerus* as a rather primitive dyrosaurid along with *Anthracosuchus balrogus*, where both are intermediate taxa between the more basal *Phosphatosaurus–Sokotosuchus* clade (Young et al., 2016; Wilberg, Turner & Brochu, 2019) and *Arambourgisuchus* (Young et al., 2016). *Cerrejonisuchus improcerus* was recovered from the Cerrejón Formation, Colombia, at the La Puente Pit within the Cerrejón Coal Mine (underclay of Coal Seam 90, see Hastings et al. (2010)). The environment within which the Cerrejón Formation was deposited corresponds to a tropical rainforest of the middle–late Paleocene (Wing et al., 2009). The referred specimen of *Cerrejonisuchus improcerus* (UF/IGM 31) preserves a skull, as well as total of 18 vertebrae plus an odontoid: one odontoid; four cervicals; ten thoracics (one is actually flattened on the ventral side of the skull); two lumbars; two sacrals; and one caudal.

All of the vertebrae are weathered. We have labeled in the text the cervical vertebrae *C*, the thoracics *Th*, the lumbars *L*, the sacrals *S*, and the caudal *Cd*. We have also numbered each vertebra, but it only reflects their relative position in the vertebral column. All vertebral stiffness inferences are based on the works of Molnar, Pierce & Hutchinson (2014) and Molnar et al. (2015), and also Schwarz-Wings, Frey & Martin (2009). The centrum width has been chosen as the reference measurement for the centrum. The length (anteroposteriorly) of the centrum is subject to change too much along the axial region (as for other crocodyliforms such as *Dakosaurus maximus, Cricosaurus suevicus* (Fraas, 1902), *Steneosaurus leedsi* (Andrews, 1913), or any modern crocodylian (Grigg & Kirshner, 2015)). The height of the centrum also varies a lot for this specimen, whereas the width remains more constant.

The investigation of the skeletons of the hyposaurines *Congosaurus bequaerti* (holotype from MRAC Tervuren), *Hyposaurus natator* (NJSM 23368; YPM VP.000380—heautotype, VP.000753—holotype of subspecies *Hyposaurus natator oweni* (Troxell, 1925),

VP.000985 – holotype) and Hyposaurinae indet. (all previously referred to *Hyposaurus rogersii*: AMNH FARB 1416, 1421, 1432, 2389, 2390; ANSP 8629–8669, 9631–9693, 13656; YPM VP.000764) (*Jouve et al., 2020*), and the crocodylians *Crocodylus porosus* (Aquarium–Museum Liège R.G.294), and *Mecistops cataphractus* (RBINS 18374) helped elaborate the identification strategies for ordering the vertebrae of *Cerrejonisuchus improcerus*.

Hyposaurine dyrosaurids possess at least 22 pre-sacral vertebrae (*Langston, 1995*; *Schwarz-Wings, Frey & Martin, 2009*), but there is evidence from *Rhabdognathus* (*Storrs, 1986*; *Langston, 1995*) and *Dyrosaurus maghribensis* (*Jouve et al., 2006*) that some dyrosaurids had at least 25 pre-sacrals. Modern crocodylians possess eight to nine cervicals and 15 to 16 dorsals (thoracics and lumbars) (*Mook, 1921*; *Grigg & Kirshner, 2015*; *de Souza, 2018*), whereas crocodylomorphs are considered to possess nine cervical vertebrae (*Steel, 1973*). *Jouve et al. (2006)* interpreted that *Dyrosaurus maghribensis* possesses nine cervicals (including the atlas–axis complex as two separate vertebra). By observing several partial and more complete dyrosaurid skeletons (e.g. *Dyrosaurus maghribensis* NHM VP R36759; *Hyposaurus natator* NJSM 23368; *Congosaurus bequaerti* MRAC 1839,1870,1840,1868,1850,1871,1869,1872,1873,1849), we confirm that dyrosaurids possessed seven post atlas-axis cervicals like in the hyposaurine skeletal reconstruction of *Schwarz, Frey & Martin (2006)*. Indeed, some anterior thoracic vertebrae are sometimes mistaken for cervicals due to the shifting position of the parapophysis in this area (e.g. as in *Jouve & Schwarz (2004)*; *Callahan et al. (2015)*). A great particularity of dyrosaurids is the presence of large hypapophyses among the anterior thoracic vertebrae (*Owen, 1849*; *Langston, 1995*), just like *Hyposaurus* (*Owen, 1849*).

The cervicals were identified following the presence of a parapophysis and a diapophysis on the lateral sides of the centrum, or the presence of a cervical rib as it is the case for modern crocodylians (*Grenard, 1999*; *Grigg & Kirshner, 2015*; *de Souza, 2018*).

The dorsal vertebrae were identified as such using the shape and position of the lateral process: it is generally single (i.e. single base) and borne on the neural arch, just like in crocodylians where it is often called 'the transverse process' (*Romer, 1956*; *Grenard, 1999*; *Grigg & Kirshner, 2015*; *de Souza, 2018*). Among thoracics, the lateral process splits into two processes distally which resemble two rami of a single structure: the parapophyseal process (anterior), and the diapophyseal process (posterior). Each process bears a distinct end, the parapophysis and diapophysis, corresponding to two different attachment sites on the thoracic rib just like modern crocodylians (*Mook, 1921*; *Grigg & Kirshner, 2015*; *de Souza, 2018*). We chose to follow the terminologies from *de Souza (2018)* because we found the definition of transverse processes of dorsals too ambiguous for this work on a more basal dyrosaurid; also we decided to use the general term of 'lateral process' for all bony structures of the vertebrae laterally emerging (either from the centrum or neural arch) instead of sporadically using 'transverse process' which has a restricted meaning among Crocodylia (and it is not always possible to meet with the necessary requirements with fossils to use this definition) (*de Souza, 2018*). For each one of the thoracics of *Cerrejonisuchus* the lateral process is attached to the centrum, this indicating the relative maturity of the specimen (*Hastings et al., 2010*), even if dyrosaurids

are known to possess weak neurocentral sutures (*Buffetaut, 1978b*). The anterior portion of the lateral process, called the parapophyseal process, is always shorter than the posterior one, the diapophyseal process. Both processes are also distinguishable in parasagittal section (i.e. if the process is broken) as the diapophyseal process is dorsoventrally thicker than the parapophyseal process, with a constriction separating both. For these reasons, the two distinct portions of the lateral process will be called 'anterior' and 'posterior ramus' in the description of the material to remove any ambiguities. Nevertheless, it is important to note the crushed state of all vertebra, which has influenced their thickness. Since dyrosaurid vertebrae are amphicoelous (*di Stefano, 1903*; *Buffetaut, 1976*), both the lateral processes and the zygapophyses play a key role in orienting the vertebrae anteroposteriorly. One main difference we could observe is that dyrosaurids (e.g. *Congosaurus bequaerti* MRAC 1866 or *Hyposaurus natator* NJSM 23368) do not tend to form a synapophysis (which is the fusion of both articular facets of the lateral process, or transverse process, into a single distal facet) on their last thoracics like crocodylians (e.g. *Mecistops cataphractus* RBINS 18374 or see *de Souza (2018)*). The lumbar vertebrae also possess two distal facets on its lateral process, but the parapophysis is less developed than the diapophysis. This is most probably because no actual ribs were borned by the lumbars, which is part of the essence of being a lumbar vertebra like in crocodylians (*Grigg & Kirshner, 2015*; *de Souza, 2018*).

The sacrals of *Cerrejonisusuchus* are still connected together, and their lateral process facing each other make it easy to identify them. In Crocodylia, the existence of a single distal extremity on sacrals is due to the fusion of the diapophysis and parapophysis into the synapophysis (*de Souza, 2018*). The single caudal of *Cerrejonisusuchus* also bears distinctive features such as a tall and narrow neural spine, small zygapophyses and the presence of prominent chevron facets. In Crocodylia, the lateral process of both sacral and caudal vertebrae is to be called 'costal process' or 'rib' because its origin differs from that of the dorsals (thoracics and lumbars alike) according to *de Souza (2018)*. Here we decided to keep using 'lateral process' because its broader and more basic meaning better serves the goals of this paper.

### The cervical region

The cervical region is composed of five vertebrae: there are four cervicals and an odontoid preserved. *Hyposaurus natator* (NJSM 23368) possessed seven cervicals (comprising the atlas and axis-odontoid as CI and CII respectively) while modern crocodylians reach eight or nine cervicals (*Mook, 1921*; *Grigg & Kirshner, 2015*; *de Souza, 2018*). The odontoid (see Cervicals) presents the typical stretched-hexagonal shape as found in other crocodyliforms, notably among thalattosuchians (e.g. *Thalattosuchus superciliosus* SMNS 10116; *pers. obs.*), dyrosaurids (e.g. *Congosaurus bequaerti* holotype at MRAC ; *Hyposaurus natator* NJSM 23368 or Hyposaurinae indet. AMNH FARB 2390; *pers. obs.*), and crocodylians. Yet, the odontoid of *Cerrejonisuchus improcerus* (UF/IGM 31) is significantly wider laterally thus giving the impression of an ellipsoid (with its greatest axis laterally oriented) in anterior view; its height over width ratio is 0.61, whereas that of

*Congosaurus bequaerti* (MRAC 1839) is 0.78, and Hyposaurinae indet. (AMNH FARB 2390) is 0.82 (*personal observations*).

The anterior facet of the odontoid of *Cerrejonisuchus improcerus* is concave and is bordered laterally and posteroventrally by two small protuberances (the lateral one being the largest). Posteriorly, the center and dorsal portion of the bone are protruding, leaving the lateral and ventral parts hollow. This hump is where the bone connects to the axis (see Fig. 1).

Hyposaurine dyrosaurs (i.e. *Congosaurus bequaerti* and *Hyposaurus natator* (*Schwarz-Wings, Frey & Martin, 2009*)) possess rather long neural spines on their cervicals (see Fig. 1), which is a trait also observed on *Cerrejonisuchus*. Indeed, C'4' (the only one preserved) displays a neural spine whose shape is not unlike that of the cervicals of the aforementioned i.e. *Congosaurus bequaerti* (holotype; *pers. obs.*) and *Hyposaurus natator* (NJSM 23368; *pers. obs.*): among hyposaurine dyrosaurs anterior or middle cervicals possess slender or pointed neurals which become wider distally around the thoracic transition. The neural spine of C'4' measures 66.9 mm and accounts for 166% of the height of the anterior centrum (see Table 1), making it shorter than that of Th'0'. For these reasons, C'4' must be at least a middle cervical vertebrae, i.e. it must have ranged from the position three to five.

The anterior facet of C'2' displays a shield-like shape: its ventral surface is rounded while the dorsal surface is rather flat. In C'4' both facets appear more round than shield-shaped. This variation is also found in *Hyposaurus natator* (YPM VP.000380 – heautotype; *pers. obs.*) and *Hyposaurus natator oweni* (VP.000753 – holotype; *pers. obs.*) where the anterior facet of a cervical is usually round or hexagonal but the posterior facet takes a shield-like shape. This difference is particularly marked in the anterior thoracics (i.e. CIII–CIV) of *Hyposaurus natator* (e.g. NJSM 23368; YPM VP.000380; *pers. obs.*) since the parapophyseal process and anterior facet are more or less joined, thus influencing the silhouette of the anterior facet's margin. Posteriorly, the size of the centrum increases in width, height and length (the length of C'3' may have been overestimated as its actual length is not easily observable; see Table 1).

In C'1' and C'2' the parapophyseal process is shorter than the diapophyseal process, which is a condition also observed on the holotype of *Congosaurus bequaerti* (e.g. MRAC 1868; *pers. obs.*), on Hyposaurinae indet. (AMNH FARB 1421, 2389; *pers. obs.*), and on *Hyposaurus natator* (NJSM 23368; *pers. obs.*), and which is also found in crocodylians (*Grigg & Kirshner, 2015*). However, in AMNH FARB 2389 (Hyposaurinae indet.; *pers. obs.*), the first cervical vertebra directly posterior to the axis (i.e. CIII) shows a slightly longer parapophyseal process, and in AMNH FARB 2390 (Hyposaurinae indet.; *pers. obs.*) both actually seem of relatively equal dimensions. In *Cerrejonisuchus improcerus*, C'2' has both its diapophyseal and parapophyseal processes centered on the lateral sides of the centrum as in the posterior cervicals (i.e. CVI and CVII) of *Hyposaurus natator* (NJSM 23368; *pers. obs.*). Indeed, the diapophyseal and parapophyseal processes of the anterior and middle cervicals (CIII–CV) of *Hyposaurus natator* (e.g. NJSM 23368; YPM VP.000380; *pers. obs.*) and Hyposaurinae indet. (e.g. AMNH FARB 2389; *pers. obs.*) are

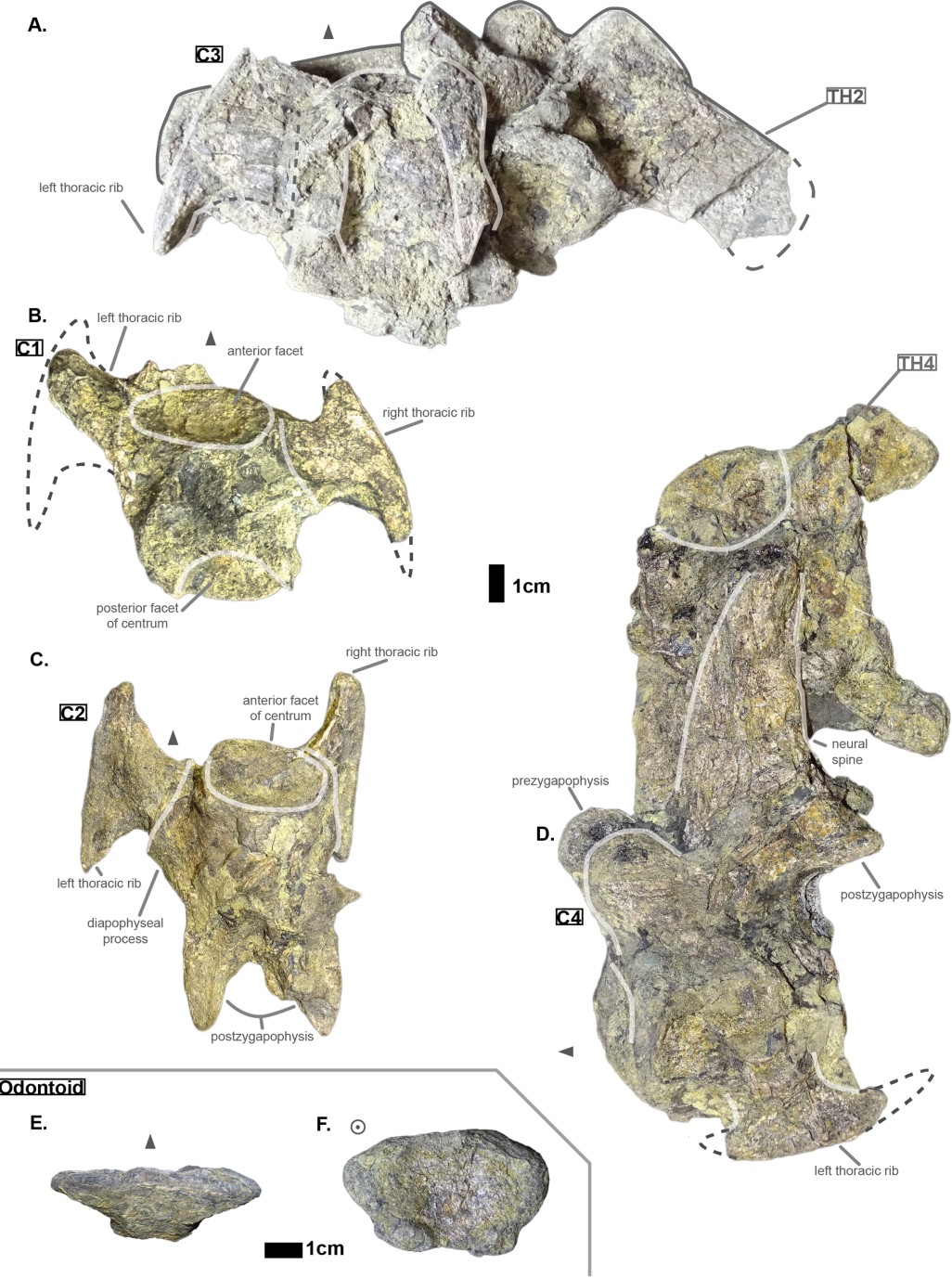

**Figure 1 Cervicals of *Cerrejonisuchus improcerus* UF/IGM 31:** (A) Cervical C'3' in dorsal view; (B) Cervical C'1' in dorsal view; (C) Cervical C'2' in dorsal view; (D) Cervical C'4' in lateral view (left); (E) Odontoid in dorsal view; (F) Odontoid in anterior view. Cervical C'3' and C'4' are respectively fused to Th'2' and Th'4'. Scale bar represents 1 cm. Grey arrow points towards anterior.

Table 1 **Table depicting each cervical vertebrae (ordered) and some measurements.** 'CH' represents the maximal height of the centrum (dorso-ventrally); 'CW' represents the maximal width of the centrum (laterally); 'CL' represents the anterior-posterior length of the centrum; 'N ang' represents the angle (whole number) that the neural spine forms with the horizontal (coronal plane). When the neural possesses a corner, the two angles are separated by an en dash; 'NH' represents the height of the neural spine; 'Para W' stands for the greatest length of the distal parapophyseal facet (which may be tilted regarding the antero-posterior plane) of the lateral process; 'Dia W' stands for the greatest length of the distal diapophyseal facet (which may be tilted regarding the antero-posterior plane) of the lateral process; 'Para L' represents the proximal-distal length of the anterior ramus (or parapophyseal process) of the lateral process; 'Dia L' represents the proximal-distal length of the posterior ramus (or diapophyseal process) of the lateral process; 'Prez Maj' represents the length of the major axis of the elliptic surface of the prezygapophysis; 'Postz Maj' represents the length of the major of the elliptic surface of the post-zygapophysis; 'Prez min' represents the length of the minor axis of the elliptic surface of the pre-zygapophysis (and is usually perpendicular to the corresponding 'Prez Maj'); 'Postz min' represents the length of the minor axis of the elliptic surface of the postzygapophysis (and is usually perpendicular to the corresponding 'Postz Maj'); 'Prez ang' refers to the angle (degree) between the horizontal plane (or coronal plane) and prezygapophysis; 'Postz ang' refers to the angle (degree) between the horizontal plane (or coronal plane) and the postzygapophysis. The lowercase letters 'a' and 'p' stand for 'anterior' and 'posterior' respectively.

|  | C1 | C2 | C3 | C4 |
|---|---|---|---|---|
| CH a (mm) | — | 24.01 | — | 40.29 |
| CH p (mm) | 29.03 | — | — | 40.58 |
| CW a (mm) | — | 29.35 | — | — |
| CW p (mm) | 27.76 | 29.59 | 43.95 | — |
| CL (mm) | 38 | 36.34 | 55.38 | 50.45 |
| N ang (°) | — | — | — | 72 |
| NH (mm) | — | — | — | 66.89 |
| Hyp height (mm) | — | — | — | — |
| Prez Maj (mm) | — | — | — | 19.97 |
| Postz Maj (mm) | — | 12.62 | — | 20.08 |
| Prez min (mm) | — | — | — | 13.9 |
| Postz min (mm) | — | 7.88 | — | — |
| Prez ang (°) | — | — | — | — |
| Postz ang (°) | — | — | — | — |
| NH/CW a (%) | — | — | — | — |
| NH/CH a (%) | — | — | — | 166.02 |
| Dia W/Dia L (%) | — | — | — | — |
| Para W/Para L (%) | — | — | — | — |
| Para L/Dia L (%) | — | — | — | — |
| Dia L/CW a (%) | — | — | — | — |

anteriorly located, and migrate towards the center of the centrum posteriorly (so that the processes are almost centered at CV).

The exact inclination angle of the diapophyseal and parapophyseal processes are lost, but C'2' still shows the remnants of their initial orientation (see Fig. 1): both were ventrally oriented with their distal facet (i.e. diapophysis and parapophysis respectively) facing both anteriorly and laterally.

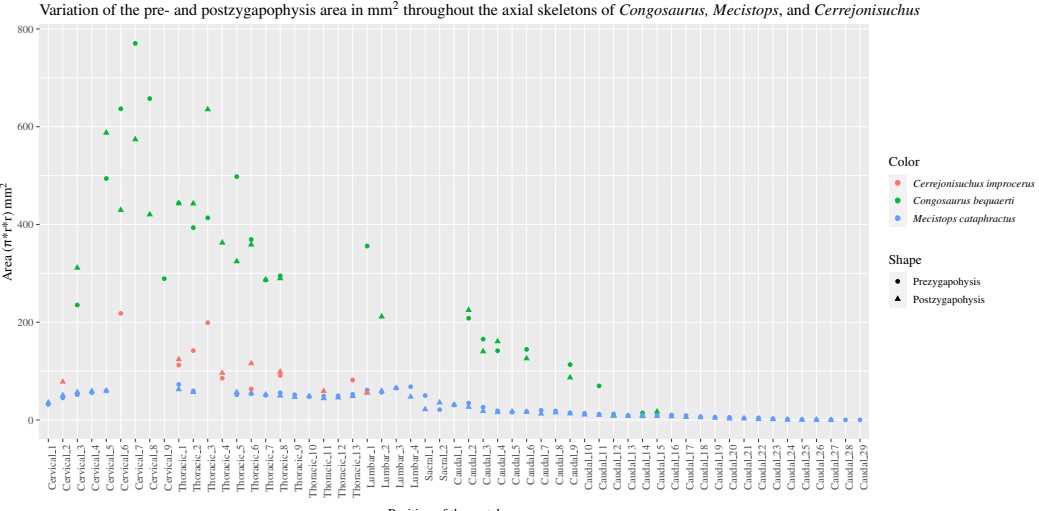

**Figure 2** **Area in mm² of the preserved pre– and postzygapophysis of *Cerrejonisuchus improcerus* (UF/IGM 31), *Mecistops cataphractus* (RBINS 18374), and the holotype of *Congosaurus bequaerti*.** There are two observable peaks for *Cerrejonisuchus improcerus* and, unlike *Mecistops cataphractus* RBINS 18374, the three first anterior thoracics show an increasing trend. For *Congosaurus bequaerti*, note the existence of two peaks: one at the mid-cervicals, and one at the anterior thoracics. For *Mecistops cataphractus*, note the existence of two peaks indicating vertebral transition areas. Cervicals have an increasing trend while all the other parts, excluding the peaks, have a decreasing trend posteriorly.

It is not possible to tell if *Cerrejonisuchus improcerus* possessed a posterior ventral keel on its anterior or middle cervicals like *Hyposaurus natator* (e.g. NJSM 23368; YPM VP.000380 – heautotype; *pers. obs.*) or *Congosaurus bequaerti* (MRAC holotype, e.g. MRAC 1840, 1868), or Hyposaurinea indet. (AMNH FARB 1416, 1432, 2389, 2390; ANSP 8649; *pers. obs.*). This structure (i.e. ventral keel) differs from the hypapophysis as it is of less significant height, and located posteriorly on the centrum as opposed to the anteriorly positioned hypapophysis of the last cervicals (e.g. Hyposaurinae indet. AMNH FARB 1421; *pers. obs.*) or anterior thoracics (e.g. thoracic numbered MRAC 1872 of *Congosaurus bequaerti*; *pers. obs.*).

The pre– and postzygapophyses are already large in this portion of the skeleton compared to their centrum, see Table 1. Also, they increase in size posteriorly as they approach the thoracic region as in the crocodylian *Mecistops cataphractus* or *Congosaurus bequaerti* (see Fig. 2). However, in *Cerrejoniuchus improcerus*, the first thoracics still follow the increasing trend initiated among the cervicals and the decreasing trend occurs here more posteriorly (i.e. among the thorarics; see Fig. 2) than in the crocodylian *M. cataphractus*. In *Congosaurus*, the anterior thoracics do not follow the same increasing trend as in *Cerrejonisuchus* but rather form a plateau before starting the decreasing slope posteriorly. Yet, both *Cerrejonisuchus* and *Congosaurus* show a peak among the anterior thoracics (see Fig. 2) which totally breaks from the other thoracics. This feature is not present in *Mecistops* (see Fig. 2).

### The anterior thoracic vertebrae UF/IGM 31 Th'0', Th'1', Th'2', Th'3', Th'4' and Th'5'

Region is composed of 12 vertebrae: ten thoracics and two lumbars are preserved. Among the thoracic vertebrae, five (UF/IGM 31 Th'0', Th'1', Th'2', Th'3', Th'4' and Th'5') belong to a more anterior portion of the thoracic region, two others (UF/IGM 31 Th'6' and Th'7') are certainly middle thoracic vertebra, and the two remaining ones (UF/IGM 31 Th'8' and Th'9') are more posterior. The actual number of vertebrae is unknown for *Cerrejonisuchus*, so we decided to order the thoracics relatively to one another. This classification is based on the evolution of several key features throughout the axial skeleton, which are: the parapophyseal and diapophyseal processes, the neural spine and the hypapophysis. The thorough investigation of the holotype of *Congosaurus bequaerti* (MRAC) and the reconstruction of the hyposaurine skeleton in *Schwarz-Wings, Frey & Martin (2009)* revealed that the height of the neural spine was a good ordering trait for hyposaurine dyrosaurids. We then applied the same trend to *Cerrejonisuchus* as we inferred it would be similar among basal and derived dyrosaurids. The neural spines of thalattosuchians and crocodylians does not vary much posteriorly: for this reason the extensive variation of the neural spine has been considered (yet hypothetically) a general dyrosaurid feature. The next important ordering traits after the height of the neural spine are the dimensions of both rami (length, distal facet and base width), and mostly their proportions with regards to the centrum. Indeed, it is well known that the distal extremities (parapophysis and diapophysis) of the thoracic lateral process encounter that of their corresponding thoracic rib, and also that a larger distal surface means a larger rib. Moreover, these larger ribs are not found anteriorly nor posteriorly; they represent the stoutest part of the thoracic skeleton, which gradually increases in the cervical region and decreases towards the lumbar region (*Schwarz-Wings, Frey & Martin, 2009*). Therefore, the change in size of the parapophysis and diapophysis is sorting feature of vertebrae. We quantify the evolution of these traits in *Cerrejonisuchus improcerus* following different ratios, such as total length over the centrum's width or total length over distal thickness, all of which are detailed below. The centrum width has been taken as the reference because it is more often preserved than the height of the centrum. The absolute dimensions of the centrum and the different processes are considered. The tables below contain all of the measurements (see Table 1 and Table 2) used for classifying and describing the thoracic vertebra.

The anterior-posterior sequence of anterior thoracic vertebrae is as follow: **UF/IGM 31 Th'0'**, **UF/IGM 31 Th'1'**, then **UF/IGM 31 Th'2'** and **UF/IGM 31 Th'3'**, and finally **UF/IGM 31 Th'4'**. Classification details are presented here below.

The anterior thoracic vertebrae are recognizable because of the relatively short anterior ramus (or parapophyseal process), plus the long neural as well as the presence of a hypapophysis (see Fig. 3). Indeed, middle and posterior thoracics possess a longer anterior ramus (both absolute and relative) but, in contrast, a shorter neural spine.

The parapophyseal and diapophyseal processes are borne on the neural arch but their orientation remains unknown (see Fig. 3). Similarly, the orientation of the distal
**Table 2 Table depicting each thoracic vertebrae (ordered) and some measurements.** 'CH' represents the maximal height of the centrum (dorso-ventrally); 'CW' represents the maximal width of the centrum (laterally); 'CL' represents the anterior-posterior length of the centrum; 'N ang' represents the angle (whole number) that the neural spine forms with the horizontal (coronal plane). When the neural possesses a corner, the two angles are separated by an en dash; 'NH' represents the height of the neural spine; 'Hyp height' represents the maximal height of the hypapophyse, 'Lat W' represents the base width (antero-posterior) of the lateral process; 'Lat H' represents the base height (dorso-ventral) of the lateral process; 'Para H' represents the height of the distal parapophyseal facet of the lateral process and is taken perpendicular to 'Para W'; 'Dia H' represents the height of the distal diapophyseal facet of the lateral process and is taken perpendicular to 'Dia W'; 'Para W' stands for the greatest length of the distal parapophyseal facet (which may be tilted regarding the antero-posterior plane) of the lateral process; 'Dia W' stands for the greatest length of the distal diapophyseal facet (which may be tilted regarding the antero-posterior plane) of the lateral process; 'Para L' represents the proximal-distal length of the anterior ramus (or parapophyseal process) of the lateral process; 'Dia L' represents the proximal-distal length of the posterior ramus (or diapophyseal process) of the lateral process; 'Prez Maj' represents the length of the major axis of the elliptic surface of the prezygapophysis; 'Postz Maj' represents the length of the major of the elliptic surface of the postzygapophysis; 'Prez min' represents the length of the minor axis of the elliptic surface of the prezygapophysis (and is usually perpendicular to the corresponding 'Prez Maj'); 'Postz min' represents the length of the minor axis of the elliptic surface of the postzygapophysis (and is usually perpendicular to the corresponding 'Postz Maj'); 'Prez ang' refers to the angle (degree) between the horizontal plane (or coronal plane) and prezygapophysis; 'Postz ang' refers to the angle (degree) between the horizontal plane (or coronal plane) and the postzygapophysis. The lowercase letters 'a' and 'p' stand for 'anterior' and 'posterior' respectively.

| | Th'0' | Th'1' | Th'2' | Th'3' | Th'4' | Th'5' | Th'6' | Th'7' | Th'8' | Th'9' |
|---|---|---|---|---|---|---|---|---|---|---|
| CH a (mm) | 31.23 | 37.15 | — | 28.02 | 29.62 | 33.01 | 37.24 | — | 36.33 | 34.56 |
| CH p (mm) | 31.36 | — | 36.05 | 30.98 | — | 31.75 | 39.64 | — | 34.74 | 31.38 |
| CW a (mm) | — | 31.17 | — | 42.68 | 42.09 | 38.9 | 43.65 | 35.63 | 35.02 | 39.09 |
| CW p (mm) | 32.3 | 29.95 | 35.25 | 39.02 | — | 37.39 | 44.02 | — | 33.92 | 41.67 |
| CL (mm) | 32.53 | 36.1 | 39.45 | 30.17 | — | 21.29 | — | — | 35.39 | 40.39 |
| N ang (°) | 90–65 | 90–65 | 90–61 | — | — | — | — | — | 84–58 | — |
| NH (mm) | 98.82 | 86.37 | 78.8 | 58.78 | 57.9 | 47.46 | 44.71 | 43.47 | 33.34 | — |
| Hyp height (mm) | — | 22.2 | 42.78 | 21.5 | 23.61 | 22.7 | — | — | — | — |
| Lat W (mm) | — | — | — | 26.61 | 27.08 | 23.27 | 24.41 | 27.76 | 23.3 | 25.05 |
| Lat H (mm) | — | — | — | 7.65 | — | 8.94 | 7.23 | 7.39 | — | 8.39 |
| Para H (mm) | — | — | — | 4.72 | 4.02 | 5.76 | 5.23 | 7.84 | 3.5 | — |
| Dia H (mm) | — | — | — | 5.62 | 6.21 | 7.23 | 5.65 | 6.49 | 4.06 | 3.64 |
| Para W (mm) | — | — | — | 8.84 | 11.76 | 9.57 | 10.02 | 10.56 | 6.36 | — |
| Dia W (mm) | — | — | — | 13.5 | 14.4 | 17.74 | 16.43 | 18.46 | 15.8 | 13.2 |
| Para L (mm) | — | — | — | 8.19 | 17.41 | 20.82 | 22.87 | 29 | 26.01 | — |
| Dia L (mm) | 18.65 | — | — | 31.17 | 39.97 | 45.61 | 42.13 | 47.61 | 42.58 | 44.07 |
| Prez Maj (mm) | 13.6 | 14.26 | 18.1 | 13.71 | — | 10.48 | 14.06 | — | — | 11.04 |
| Postz Maj (mm) | 14.09 | — | 18.46 | 12.88 | 13.87 | 14.3 | 12.63 | — | 12.2 | — |
| Prez min (mm) | 10.51 | 12.67 | 14 | 7.92 | 11.66 | 7.71 | 8.25 | — | — | 9.44 |
| Postz min (mm) | 11.21 | — | — | 9.52 | — | 10.31 | 9.95 | — | 6.17 | — |
| Prez ang (°) | 45 | 45.9 | — | 22.9 | 39.91 | 45 | — | — | — | 15 |
| Postz ang (°) | 45 | — | — | 41 | — | 45 | 41.7 | — | — | — |
| NH/CW a (%) | — | 277.09 | — | 137.72 | 137.56 | 122.01 | 102.43 | 122 | 95.2 | — |
| NH/CH a (%) | 316.43 | 232.49 | — | 209.78 | 195.48 | 143.77 | 120.06 | — | 91.77 | — |
| Dia W/Dia L (%) | — | — | — | 43.31 | 36.03 | 38.89 | 39 | 38.77 | 37.11 | 29.95 |
| Para W/Para L (%) | — | — | — | 107.94 | 67.55 | 45.97 | 43.81 | 36.41 | 24.45 | — |
| Para L/Dia L (%) | — | — | — | 26.28 | 43.56 | 45.65 | 54.28 | 60.91 | 61.09 | — |
| Dia L/CW a (%) | — | — | — | 73.03 | 94.96 | 117.25 | 96.52 | 133.62 | 121.59 | 112.74 |

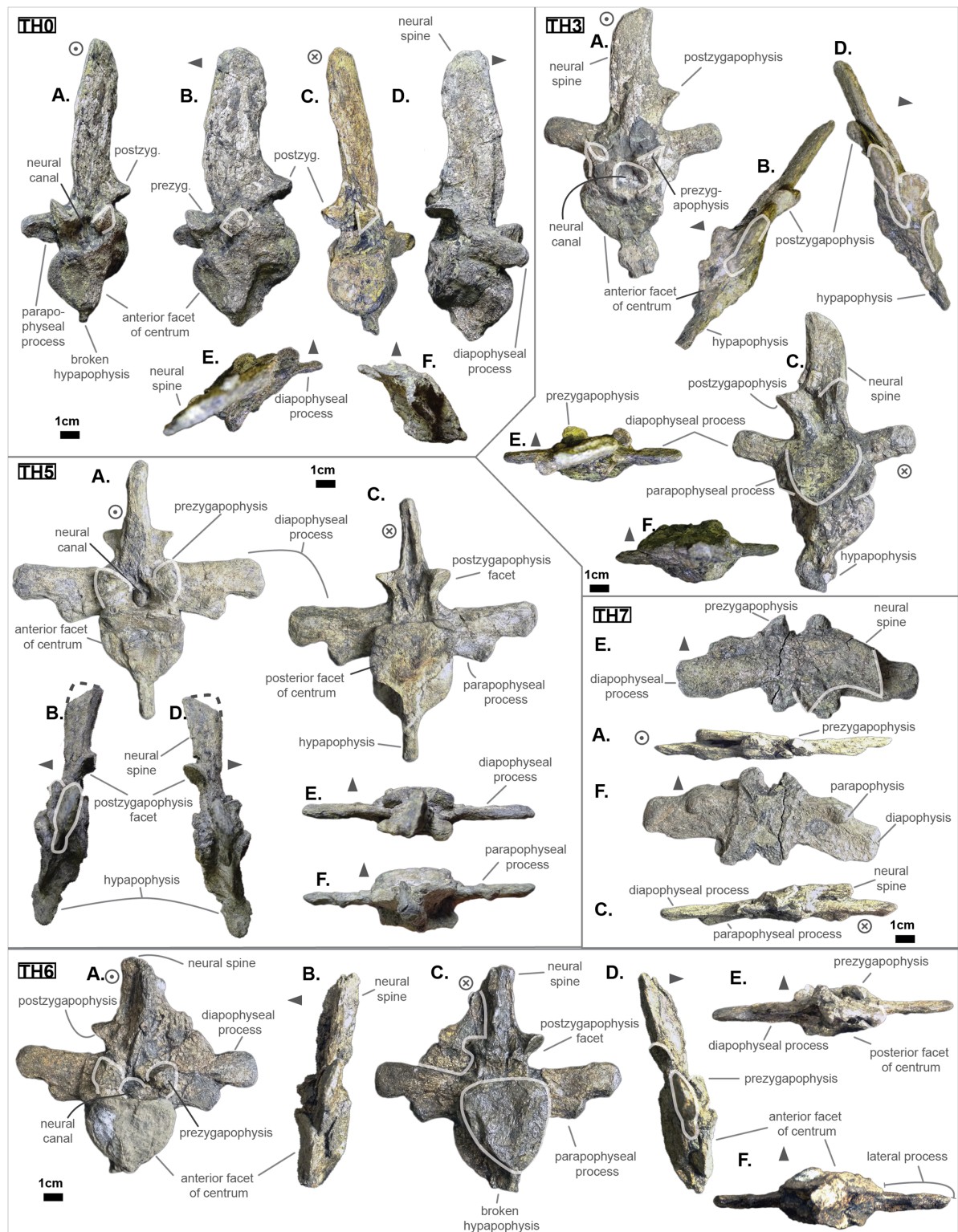

**Figure 3 Thoracics of *Cerrejonisuchus improcerus* UF/IGM 31.** (A) Anterior view; (B) left lateral view; (C) posterior view; (D) right lateral view; (E) dorsal view; (F) ventral view. Grey arrow points towards anterior.

extremities of the processes (parapophysis and diapophysis) has not been preserved. Yet, it is likely that both distal facets exhibited some sort of oval shape before fossilization (i.e. longer anteroposteriorly than high). In the anteriormost thoracics (i.e. Th'0', Th'1', and Th'2'), the parapophyseal process is close to the diapophyseal process so that both share the same base (called the 'lateral process'; see Fig. 3). Furthermore, the parapophyseal process (or 'anterior ramus') is strictly ventral to the diapophyseal process (or 'posterior ramus') as it is the case for *Hyposaurus natator* (NJSM 23368, *pers. obs.*). Yet another argument supporting the aforementioned classification of Th'0', Th'1', and Th'2'. Unsurprisingly, the exact position of the lateral process as a whole is not certain but, from what is apparent on lateral sides of Th'2' and Th'1', it appears to have been centered on the centrum like *Congosaurus bequaerti* (holotype; *pers. obs.*) or *Hyposaurus natator* (NJSM 233368; *pers. obs.*), Hyposaurinae indet. (YPM.VP000764; *pers. obs.*), and unlike *Alligator mississippiensis* (*Storrs, 1986*) or *Mecistops cataphractus* (RBINS 18374; *pers. obs.*) where they are more anteriorly located.

The neural spine is rather straight (see Fig. 3): both the anterior and posterior surfaces are parallel and tilted by more or less 75° (in relation to the horizontal plane, see Table 2). However, there is a portion at the base of the anterior surface that is vertical, leading to the presence of a flexion point along the surfaces. Hence the neural spine is bevel-shaped, with the distal extremity being posteriorly pointed with a smooth and convex anterior. The vertebrae Th'0', Th'1' and Th'2' greatly resemble one another both in the shape of their neural spine and their lateral processes, meaning that they were probably closer together than to the other anterior thoracics.

There is also the presence of a posterior notch which runs between half the total length of the neural and almost all of it: on Th'0' the notch appears restricted to the area between the postzygapophyses, whereas on Th'3' the notch stops at about 1/6th from the top (see Fig. 3). The notch separating the postzygapophyses ventrally (visible in Th'0', Th'1', Th'3' and Th'4') shows the absence of a hyposphene and thus the non-existence of a *hyposphene-hypantrum* articulation (*Stefanic & Nesbitt, 2019*). Still, the existence of a notch along the posterior surface of the neural conveys the thoracic vertebra's capacity to interlock to a certain extent. It means that the vertebrae were probably quite close, i.e. the intervertebral disc was rather thin than thick. The presence of a notch may be an attempt to enhance the column flexibility in the dorsal plane by creating extra space for flexion. The original inclination of the neural spine is not preserved, but its straight outline does not indicate any change of angle dorsally (see Fig. 3). Nevertheless, it seems the neural spine was quite vertical (close to 80–90°).

Th'2' has both the neural and hypapophysis (yet incomplete) preserved, and the length of both processes (see Table 2) place it as either the last cervical of one of the first thoracics. The wide and rounded tip of the neural further supports this hypothesis: indeed, among hyposaurine dyrosaurs (i.e. *Congosaurus bequaerti*, holotype, and *Hyposaurus natator*, NJSM 23368), anterior or middle cervicals possess slender or pointed neurals which become wider distally around the thoracic transition. Also, the outline of the neural of Th'2' greatly resembles that of Th'0' and Th'1'.

Both facets of the centrum are concave (amphicoelous), heart- or shield-shaped (larger dorsally and pointed ventrally) and ventrally united by a process (the hypapophysis; see Fig. 3). Based on the preserved dimensions (see Table 2), it seems that the posterior facet of the vertebra is slightly taller than the anterior one (which is a feature also found among some thalattosuchians, and on the holotype of *Congosaurus bequaerti*; *pers. obs.*). Nevertheless, both facets are wider than tall, this feature being more emphasized for the anterior facet. The hypapophysis starts from the anterior portion of the centrum, and is linked to the ventral margin of the anterior facet. Its anterior surface is vertical while its posterior one is slightly concave since it stretches out towards the posterior facet. Furthermore, the hypapophysis was long (unfortunately broken in Th'0' and Th'1') and exceeded the centrum's height or width, at least in the anteriormost portion of the thoracic region (see Table 2). Where it is preserved the hypapophysis appears straight with no specific orientation (see Fig. 3), which is a condition also observed in Hyposaurinae (*Schwarz, Frey & Martin, 2006*) like *Hyposaurus natator* (NJSM 23368; *pers. obs.*) and *Congosaurus bequaerti* (MRAC holotype; *pers. obs.*) (while counter-example could be the condition observed in *Mecistops cataphractus* for instance; *pers. obs.*); its shape is of a rectangle with a possibly curved tip (see Fig. 3). The presence of the hypapophysis indicate that those vertebrae were rather anteriorly positioned thoracics, and its decreasing length posteriorly (when preserved) helps ordering the vertebra.

The prezygapophysis facet is mostly oriented dorsally (see Fig. 3), with an angle (taken from the coronal plane) ranging from about 23° to roughly 40–45° (see Table 2). The postzygapophysis appears to be facing mainly ventrally with an angle of about 40–45°. Yet the crushed condition of the vertebrae makes it hard to secure the validity of these measurements (see Fig. 3). Nonetheless, the postzygapophysis value is plausible (when compared to the holotype of *Congosaurus bequaerti*; *pers. obs.*).

The pre- and postzygapophyses are quite large compared to the centrum (see Tables 2 and 3). In Th'0', the prezygapophysis represents 42.1% of the width of the anterior facet, while the postzygapophysis accounts for 43.6%. In Th'1', he maximal size of the articular facet of the prezygapophysis is quite important as it reaches 45.75% of the anterior facet's width. In Th'2', the prezygapophysis represents 51.3%, and the postzygapophysis reaches 52.4% of the width of the anterior facet. In Th'3', the greater axis of the prezygapophysis accounts for 32.1% of the width of the anterior facet (and 35.1% of the posterior one). Also, the postzygygapophysis reaches 30.2% of the width of the anterior facet (and 33% of the posterior one). In Th'4', only the postzygapophysis is present, and it represents 32.9% of the anterior facet's width. And finally, in Th'5', where both the prezygapophysis and postzygapophysis are present, the prezygapophysis accounts for 26.9% of the anterior facet's width while the postzygapophysis reaches 36.7%.

The non-gradual changes between each of the anterior thoracic vertebrae (see Table 2) supports their non-adjacent state, but also helps position them relatively. These great differences are of course emphasized by the lack of transitional vertebrae between them. Hence, the succession from anterior to posterior is likely to be: Th'0', Th'1', then Th'3', and finally Th'4'. We will presently describe the whole trend and compare the two pairs of Th'1'–Th'3' and Th'3'–Th'4' to prove it. The strongest evidence for ordering the vertebrae

**Table 3 Table depicting each lumbar vertebrae (ordered) and some measurements.** 'CH' represents the maximal height of the centrum (dorso-ventrally); 'CW' represents the maximal width of the centrum (laterally); 'CL' represents the anterior-posterior length of the centrum; 'N ang' represents the angle (whole number) that the neural spine forms with the horizontal (coronal plane); 'NH' represents the height of the neural spine; 'Hyp H' represents the maximal height of the hypapophyse, 'Lat W' represents the base width (antero-posterior) of the lateral process; 'Lat H' represents the base height (dorso-ventral) of the lateral process; 'Para H' represents the height of the distal parapophyseal facet of the lateral process and is taken perpendicular to 'Para W'; 'Dia H' represents the height of the distal diapophyseal facet of the lateral process and is taken perpendicular to 'Dia W'; 'Para W' stands for the greatest length of the distal parapophyseal facet (which may be tilted regarding the antero-posterior plane) of the lateral process; 'Dia W' stands for the greatest length of the distal diapophyseal facet (which may be tilted regarding the antero-posterior plane) of the lateral process; 'Para L' represents the proximal-distal length of the anterior ramus (or parapophyseal process) of the lateral process; 'Dia L' represents the proximal-distal length of the posterior ramus (or diapophyseal process) of the lateral process; 'Prez Maj' represents the length of the major axis of the elliptic surface of the prezygapophysis; 'Postz Maj' represents the length of the major of the elliptic surface of the postzygapophysis; 'Prez min' represents the length of the minor axis of the elliptic surface of the prezygapophysis (and is usually perpendicular to the corresponding 'Prez Maj'); 'Postz min' represents the length of the minor axis of the elliptic surface of the postzygapophysis (and is usually perpendicular to the corresponding 'Postz Maj'); 'Prez ang' refers to the angle (degree) between the horizontal plane (or coronal plane) and prezygapophysis; 'Postz ang' refers to the angle (degree) between the horizontal plane (or coronal plane) and the postzygapophysis. The lowercase letters 'a' and 'p' stand for 'anterior' and 'posterior' respectively.

| | L'1' | L'2' |
|---|---|---|
| CH a (mm) | 37.15 | 34.75 |
| CH p (mm) | — | 32.43 |
| CW a (mm) | — | 28.18 |
| CW p (mm) | — | 28.6 |
| CL (mm) | 39.7 | 35.05 |
| N ang (°) | 90 | — |
| NH (mm) | 35.46 | — |
| Hyp height (mm) | — | — |
| Lat W (mm) | 23.97 | 28.94 |
| Lat H (mm) | — | — |
| Para H (mm) | 3.37 | — |
| Dia H (mm) | 4.32 | 3.26 |
| Para W (mm) | 6.46 | — |
| Dia W (mm) | 15.18 | 12.67 |
| Para L (mm) | 27.17 | 26.67 |
| Dia L (mm) | 38.06 | 36.23 |
| Prez Maj (mm) | — | — |
| Postz Maj (mm) | 10.5 | — |
| Prez min (mm) | — | — |
| Postz min (mm) | 6.83 | — |
| Prez ang (°) | — | — |
| Postz ang (°) | — | — |
| NH/CW a (%) | — | — |
| NH/CH a (%) | 95.45 | — |
| Dia W/Dia L (%) | 39.88 | 34.97 |
| Para W/Para L (%) | 23.78 | — |
| Para L/Dia L (%) | 71.39 | 73.61 |
| Dia L/CW a (%) | — | 128.57 |

is here going to be the size (both absolute and relative) of the neural spine since we are in the anteriormost part of the thoracic region (thanks to the holotype of *Congosaurus bequaerti*; *pers. obs.*). Indeed, the neural spine of each anterior thoracic is tall (± 98.8 mm for Th'0', ± 86.4 mm for Th'1'; ± 58.8 mm for Th'3', ± 57.9 mm for Th'4' and ± 47.5 mm for Th'5') and greatly exceeds the dimensions of their centrum, which confers them a unique identifiable look (see Fig. 3). When compared to the anterior width (or height for Th'0') of their respective centrum (3): Th'0' has by far the greatest ratio with ± 316% (in relation to its height), Th'1' equals almost three times its width with ± 277%, while Th'3' is worth ± 137.7%, Th'4' reaches ± 137.5% and Th'5' about ± 122%. On the opposite, the hypapophysis shows a rather constant length of about 22 mm among all of the thoracic vertebrae where it is preserved. Therefore its ratio with the respective centrum width is not relevant as it would only reflect the centrum's dimensions. Nevertheless, the hypapophysis of the posterior cervicals exceeds by far the length of that of the anteriormost thoracics (see Figs. 1 and 3), probably because their actual length is not preserved (i.e. the hypapophysis of Th'0' and Th'1' would be broken in that case). Looking at Th'0', Th'1', and Th'3' (see Fig. 3), it appears clear from the absolute and relative height of the neural spine (see Table 2), doubled with the absolute width of their centrum, that Th'0' is the anteriormost thoracic and that Th'1' is anterior to Th'3' (rather than the opposite). Unfortunately, the lateral processes of Th'0' and Th'1' appear to be missing and cannot be used to further support this sorting. The neural spine of Th'1' also suggests the existence of a posterior notch at about one-third of its height. Th'3' and Th'5' show the same structure but here it appears to run almost the full height of the neural. The information is not available on Th'4' (see Table 2).

The difference of size of the neural spine between Th'3' and Th'4' is here more subtle (with ± 58.78 mm versus ± 57.9 mm respectively, see Table 2), which makes it more difficult to determine position. As it has been discussed in the section just above, there are also other important features which can corroborate this sorting: the relative length of the rami plus the size of their distal extremities (see Table 2). The proportional length of the anterior ramus (i.e. parapophyseal process) accounts for ± 19% of the posterior ramus (i.e. diapophyseal process) for Th'3', and this number increases to ± 43% for Th'4'. When compared to the width of their respective centrum, the posterior ramus of Th'3' reaches up to ± 64% of the length while Th'4' shows a greater proportion with ± 95% (see Table 2). If we look even further along the thoracic region, the length of the lateral process (i.e. parapophyseal and diapophyseal processes) shows an increasing trend posteriorly both proportionally to the corresponding centrum and absolutely. And not to mention that the extremity of both rami also increases from Th'3' to Th'4', with 8.84 mm for the length from the anterior ramus of Th'3' versus 11.76 mm for that of Th'4', and with 13.5 mm for the length from the posterior ramus of Th'3' versus 14.4 mm for that of Th'4' (see Table 2). These last length measurements represent the greater extend of the distal surface, even if it is slightly tilted compared to the anteroposterior plane. Indeed, the length of the rami, and thus the overall attaching site of the ribs, considerably increases in size up to a certain point among the middle thoracics, where it starts to slowly shrink

towards the lumbar (where the ribs finally disappear). As a consequence, the short rami (in both their lengths, and dimension of their distal facets) of Th'3' must be placed anterior to that of Th'4'. Additionally, the shortness of the rami of **UF/IGM 31 Th'3'** (see Fig. 3) could imply a position of the vertebra among the very first thoracics, probably from the third through fifth thoracic position as the first and second thoracic would be expected to show resorbing parapophyseal processes like in Crocodylia and Thalattosuchia.

The neural spine can be used again to order Th'4' and Th'5' as their difference in size and shape is obvious. While the neural of Th'4' is long (± 57.9 mm) and blade-like, that of Th'5' is shorter (± 47.5 mm) and broader at its extremity (see Fig. 3). This posterior modification of the neural shape is not unlike *Hyposaurus natator* (NJSM 23368; *pers. obs.*), where the mid-thoracics and posterior thoracics show square-like neurals. The relative length of the anterior ramus (i.e. parapophyseal process) to the posterior ramus (i.e. diapophyseal process) in Th'5' equals 45.6%, which is a small increase compared to Th'4'. Yet the anterior ramus hasn't reached its maximum length at this point as it usually happens more posteriorly towards the lumbar region (like in *Hyposaurus natator* (NJSM 23368; *pers. obs.*) or *Congosaurus bequaerti* MRAC 1874). In overall, the articular facets of Th'5' (i.e. parapophysis and diapophysis) are greater than those of Th'4', which definitely resolves the ordering debate since the ribs are only getting bigger posteriorly at this stage (i.e. in the anterior portion of the thoracic region).

In overall, the anterior thoracics present a substantial decrease in the height of the neural spine posteriorly, while the hypapophysis remains quite stable (see Table 2). The centrum is also of equal dimensions throughout the anterior thoracic, with the height of Th'1' being the sole exception. The postzygapophysis slightly increases in size posteriorly while the opposite trend strikes the prezygapophysis. The zygapophyses are also quite large compared to the centrum, but they are decreasing posteriorly (see Table 2). Lastly, the lateral process increases in length laterally as one goes posteriorly, so that both rami develop an increased articular facet (the diapophysis being the larger than the parapophysis) to hold even larger ribs. It seems the lateral process originated from the neural arch, however due to the crushed condition of each vertebrae, it is not possible to determine whether it was positioned at the base of the arch or on the same level as the neural canal. Likewise, assessing the orientation of the lateral process remains doubtful.

The size of the neural canal in Th'4' is 9.87 mm in height and 12.31 mm in width, while that of Th'1' reaches 13 mm in height and 12.38 mm in width. The neural canal of Th'3' is obstructed and cannot be measured. The neural canal of Th'5' is 12.9 mm high and 14.66 mm wide. It is slightly wider than that of Th'1', but is overall similar as it would be expected from two closely related thoracics among crocodyliforms (like for example in *Mecistops cataphractus* RBINS 18374 see Fig. 4). On the scatter plot graph of *Congosaurus beqauerti* (see Fig. 4), the evolution of the size of the neural canal reveals an overall decreasing trend posteriorly throughout the axial skeleton, which slightly differs from the cervicals of *Mecistops cataphractus* (see Fig. 4). There is still a discernible increase occurring at the lumbar-caudal transition, highlighting the switch in vertebral regions.

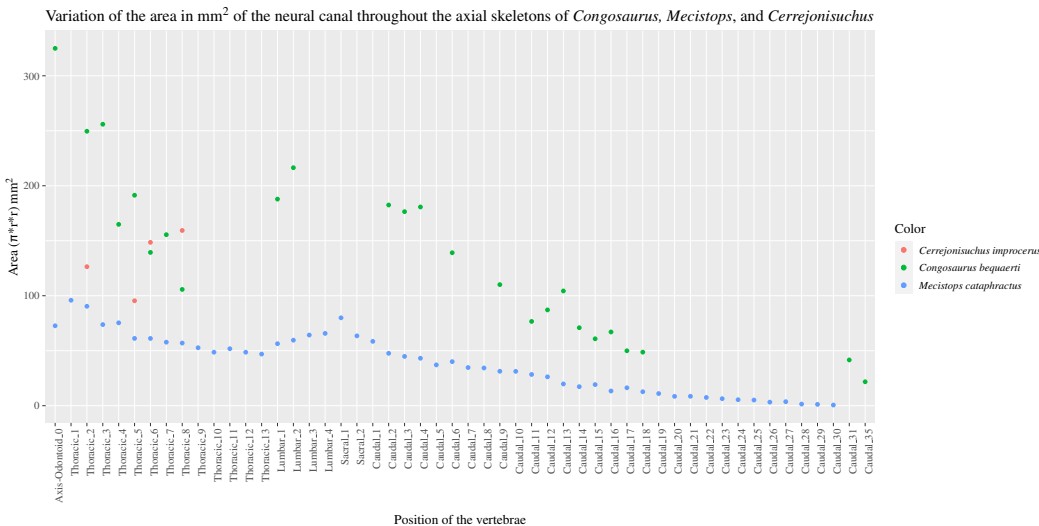

**Figure 4** Scatter plot of the area variation of the neural canal throughout the axial skeleton of *Cerrejonisuchus improcerus* (UF/IGM 31), *Mecistops cataphractus* (RBNIS 18374), and the holotype of *Congosaurus bequaerti*.

## The middle thoracic vertebrae UF/IGM 31 Th'6' and Th'7'

The anterior-posterior sequence of the middle thoracic vertebrae is as follow: **UF/IGM 31 Th'6'** then **UF/IGM 31 Th'7'** (see Fig. 3).

The middle thoracics have been ordered following the same process of classification than mentioned in the previous section, which has furthermore been improved here using to the presence of a preserved lateral process. Hence, the most important classifying features are (with the same degree of importance) the dimensions of the centrum, the size of the neural spine and that of the parapophyseal and diapophyseal processes (including the dimensions of the parapophysis and diapophysis). In parallel, it is important to mention that both the anterior and posterior thoracic vertebrae help understand the organisation among the middle thoracics. Indeed, the anterior and posterior thoracics being the easiest to identify and order, these were resolved in the first place. And their own characteristics were used as a reference to classify those in between them (i.e. the middle thoracics).

The characteristics of the middle thoracics that stand out are: the widening of the lateral process (which flares out anteroposteriorly from the neural arch to the distal ends), a further increase in the size of the ribs attachment sites (i.e. area of the parapophysis and diapophysis) and finally a neural spine decreasing in height posteriorly (which is therefore smaller than that of the anterior thoracics). All of these traits were decisive in ordering the middle thoracics, using the global trend previously inferred from the anterior and posterior thoracics.

The centrum is here heart-shaped because its maximal width is obtained more dorsally than ventrally compared to the imaginary horizontal mid-line cutting the centrum (see Fig. 3). Still, like Th'3', the centra are wider than tall. Also, the posterior facet is slightly more elongated dorsoventrally than the anterior one for an even width. Th'6' presents a

bigger centrum than Th'5', indicating that the posteriorly increasing trend initiated with the anterior thoracics is still going (see Table 2). However, Th'7' is drastically smaller than Th'6', much like the posterior thoracics, which would mean that a downward trend has been taking place somewhere between the vertebrae Th'6' and Th'7' (which is also implying their non adjacent state).

The anteroposterior width of the centrum of Th'6' could not be measured, but it seems that the hypapophysis was uniting both facets (see Fig. 3). The length of the hypapophysis cannot be known as the process is broken in Th'6'. In Th'7', the lower portion of the centrum is missing (see Fig. 3), thus making it impossible to assess the presence of a hypapophysis. It is interesting to note that the mid thoracics of *Cerrejonisuchus* still possessed a well-developed hypapophysis, which is not the case for *Congosaurus*.

Conversely, all lateral processes are preserved integrally among the middle thoracics and reflect the trend initiated in the anterior thoracics with the increase of the anterior ramus (i.e. parapophyseal process; see Fig. 3). Starting from Th'6' the length of the anterior ramus greatly increases so that is represents 54.3% of the posterior ramus's length, and then it reaches up to 60.9% in Th'7' (see Table 2). In Th'6', the posterior ramus is almost as long as the centrum is wide while the anterior branch is only equal to its half width (see Fig. 3). However, the posterior ramus is greater than the centrum's width for Th'7'. In overall, the articular facets of each rami (i.e. parapophysis and diapophysis) are the biggest in Th'7' than any other thoracic indicating a peak in the robustness of the ribcage at this point (see Table 2). Before this turning point, distal facets are increasing in size, and afterwards (i.e. in the posterior thoracic part) they slowly begin to decrease. In lateral view, the parapophysis and diapophysis are oval (greater axis usually positioned in the anteroposterior plane, but this is not observable here), with the diapophysis showing a slender and slightly pointed posterior. Sadly, it is not possible to assess the original orientation and inclination of the lateral process as a whole or of any of the rami. Compared to *Congosaurus bequaerti* (MRAC 1855, 1874), the anterior and posterior rami (parapophyseal and diapophyseal processes respectively) of *Cerrejonisuchus improcerus* (UF/IGM 31 Th'5' and Th'6') do not appear to arrange in tiers vertically. Instead, the rami rather appear to have an anterior-posterior relationship (see Fig. 3). In *Hyposaurus natator* (NJSM 23368), the relative position of each ramus changes along the axial skeleton so that their relationship is vertical anteriorly (reminiscent of the diapophyseal and parapophyseal processes of the cervicals), and changes to horizontal posteriorly. These changing arrangements are also reflected on the proximal end of the thoracic ribs in *Hyposaurus natator* (NJSM 23368) and *Congosaurus bequaerti*.

The neural spine of both Th'6' and Th'7' are similar in size (Th'6' may have a portion of the tip cut off; see Fig. 3), and display the decreasing trend initiated in Th'5' (see Table 2). Their shape resembles that of the other thoracics in being rather elongated, with the anterior portion of the tip being lower than the posterior part (see Fig. 3). The extremity appears bevel-shapes. However, while the anterior thoracics did entirely look like a blade due to their almost parallel anterior and posterior outlines, the middle thoracics show a rupture of angle along the anterior outline (see Fig. 3). Indeed, at almost half of its total length, the anterior surface of the neural presents a corner which gives a bent look to

the neural. Unfortunately, the original orientation of the neural cannot be assessed. The neural of Th'6' also reveals the existence of a wide posterior notch running along its full height (wider than the one suggested in Th'1' but about the same as Th'5'). The existence of such indentation most probably allowed some extended dorsoventral movement. The posterior notch is wider, and was probably also deeper, just in between the postzygapophyses (see Fig. 3).

Th'7' is extremely flattened dorsoventrally, with the ventral part of the centrum cut off (see Fig. 3). In Th'6' the prezygapophysis facets are mainly oriented dorsally but it seems that there is a small anterior component. Their tilting angle of both prezygapophysis does not match due to conservation issue: the left one is tilted at about 34° from the horizontal plane and the right one is steeper, with an angle of 42° (see Table 2). It would seem that none of these values reflect the true angle of the prezygapophysis. The right postzygapophysis does not bear any visible sign of deterioration, but its position has certainly been altered as well (even slightly). Its facet is mainly oriented ventrally with a small posterior component; it shows an angle of more or less 41.7° with the horizontal plane. As opposed to Th'3' and Th'1', the dimension of the zygapophysis facets are here small compared to the centrum: the greater axis of the prezygapophysis accounts for 37.75% of the anterior height of the centrum (32.2% of the width) and that of the postzygapophysis makes up 31.86% of the posterior height of the centrum (28.7% of the width). The neural canal is partially visible posteriorly and is wider than tall, almost twice as much with 10.78 mm in height and 18.82 mm in width. Though Th'3' and Th'1' seemed to be rather close, this thoracic was probably situated quite a bit further from them because of the zygapophyses. The neural spine and the hypapophysis would have helped resolving such a case but are unfortunately missing.

## The posterior thoracic vertebrae UF/IGM 31 Th'8' and Th'9'

The anterior-posterior sequence of the posterior thoracic vertebrae is as follow: **UF/IGM 31 Th'8'** then **UF/IGM 31 Th'9'** (see Fig. 3).

The shape and size of each vertebra changes along the axial skeleton, and there are key features that help identify them such as: the absolute and relative size of the lateral process, the shape and size of the neural, and the absence of a hypapophysis (which characterized the more anterior portion of the axial skeleton). Focusing on the lateral process, the anterior ramus increases in length posteriorly so that it starts at 19% (Th'3') and then reaches up to 61% (Th'8') of the posterior ramus length (see Table 2). Hence, the absolute length of both rami also differs for each vertebrae: both rami increase in length posteriorly up to a certain point around the transition with the lumbars where their absolute dimensions are then reduced. Indeed, while 'L1' shows the greatest ratio between the rami, both are also shorter than that of the directly surrounding vertebrae (i.e. the posterior and middle thoracics).

The size and shape of the neural spine, which is now a well known ordering character, is reduced compared to the other thoracics (see Fig. 3). Indeed, the neural reaches 33.3 mm for Th'8', which represents 91.8% of Th'8' anterior facet's height (and 95.2% of its width) while these numbers were greater for the other thoracics (see Table 2 and Table 2).

Unfortunately, Th'9' is missing its neural arch. Yet, the decreasing trend in the size of the neural spine is still going on posteriorly. The shape of the neural spine in Th'8' mostly resembles the middle thoracics as its outline looks like a bent bevel shape: the anterior surface is firstly erected at angle of about 84° to the horizontal, then it decreases to 58°. The posterior surface shows nearly the same curve and also possesses a notch, but does not show any hypophene structure (see Fig. 3).

In this portion of the axial skeleton, the centrum is not heart-shaped but has become rather round (i.e. shows similar height and width), with a slightly oval ventral extremity. The length of the centrum increases posteriorly from Th'8' to Th'9' (their crushed condition is different from the other thoracics and has here preserved the length).

Based on the neural spine shape and size, and the absence of hypapohysis, Th'8' resembles the thoracics of *Hyposaurus natator* (NSJM 23368) occupying the positions from eight through ten. It is possible that Th'8' of *Cerrejonisuchus* occupied was of those positions as well. The last thoracic, Th'9' seems to have been placed further posteriorly from Th'8', and were not adjacent.

In summary, the transition from the middle to the posterior thoracics is achieved through a series of reductions, notably: the size of the neural and of the lateral process (both in length of the parapophyseal and diapophyseal processes, and in the dimensions of their articular facets).

## The lumbar region

The lumbar region is characterized by a series of traits: a short neural spine; overall short lateral process with great anterior ramus; existence of a ventral keel. Indeed, both L'2' and 'L1' show locally a bump or a keel on the ventral side of the centrum (see Fig. 5). This feature was observed on *Hyposaurus* specimens (notably YPM VP.000380 & VP.000753; *pers. obs.*) and hyposaurinae specimens (notably AMNH FARB 1416 & 2390, NJSM 12293; *pers. obs.*) and thus serves as an indicator of the lumbar region.

The centrum has therefore changed slightly from the posterior thoracic region (see Fig. 3); the dorsal part of both facets is wider than the ventral part, which is in return pointed giving the impression of a shield (see Fig. 5). This is probably influenced by the existence of a ventral keel. L'2' is relatively smaller than 'L1' in the height of its facets and in the length of its centrum while it is not entirely certain that the length of L'1' has not been increased by its crushed state (see Fig. 5). Due to its larger centrum, L'1' was probably closer to the thoracic region (and Th'9') than L'2' was. Also, this ordering is supported by the size of the ventral keel which is more developed in L'2' than in 'L1' (see Fig. 5).

*Congosaurus bequaerti* (MRAC 1865 & 1896, holotype), *Hyposaurus natator* (YPM VP.000380 – heautotype & YPM VP.000985 – holotype), and *Hyposaurus natator oweni* (YPM VP.000753 – holotype) also bear ventral keels on their lumbars (and even the first sacral for YPM VP.000753 and YPM VP.000985), making it an important sorting feature. Yet, the middle thoracics of *Congosaurus bequaerti* (MRAC 1851 & 1874) possess a strong ventral ridge which is but a reminiscence of the hypapophysis.

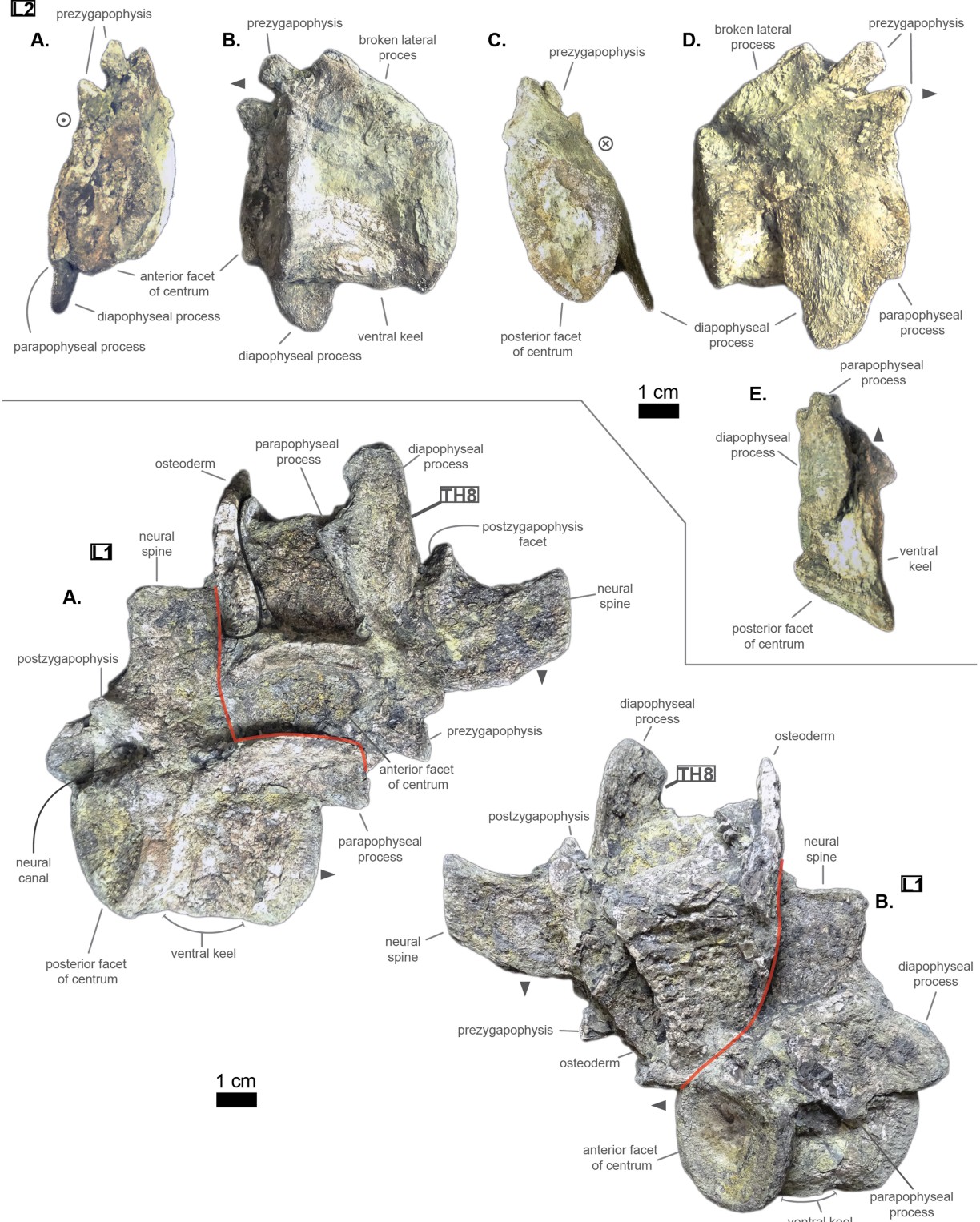

**Figure 5 Lumbars of *Cerrejonisuchus improcerus* UF/IGM 31. L'1'.** (A) Anterior view; (B) left lateral view; (C) posterior view; (D) right lateral view; (E) ventral view. L'2': (A) right lateral view; (B) left lateral view. Grey arrow points towards anterior.

As mentioned earlier, the neural spine is shorter in the lumbar region compared to the thoracic region (see Fig. 5 and Fig. 3). In 'L1' the neural spine accounts for 95.4% of the anterior facet's height, and reaches 35.5 mm in total (see 3). The posterior surface of 'L1' is clearly different from that of Th'8', and its dorsal extremity is slightly broader which gives it a more squared look.

The anterior ramus is still following the increasing trend initiated in the thoracics: the length of the anterior ramus now reaches 71% of that of the posterior ramus in 'L1' while this number equals 73.6% in L'2' (see Table 3). However, their rami are reduced both in total length and size of their distal facets (especially the anterior one), which means that these were probably not able to support actual sturdy ribs like thoracics do. Yet these may have been connected to slender and short ribs, or some cartilaginous structure but it is currently unknown. The overall reduction of the lateral process can be traced back to Th'8' (see Table 2). The distal extremities of the lateral processes of L'2' and 'L1' take the shape of elongated ovals in the anteroposterior direction. The surface of those facets is no longer flat, but rather slightly convex. The major difference between *Cerrejonisuchus* and modern crocodylians is that its lumbars are not fused into a synapophysis (*de Souza, 2018*) (e.g. *Mecistops cataphractus* RBINS 18374) but rather retain reduced but distinct distal facets (see Fig. 5). The postzygapophysis is also smaller in this region of the skeleton compared to the thoracics (see Table 3). There are no evidence of a hypophene structure emanating from the postzygapophysis preserved in the lumbars (*Stefanic & Nesbitt, 2019*).

To sum up, the transition from the posterior thoracic vertebrae to the lumbar region is easily identified thanks to the reduction in both the length and the thickness (dorsoventral) of the lateral process, plus in the size of their distal facets (which connect to the ribs). The overall shortening of the lateral process is proportionally less impressive than the two reductions just mentioned as it decreases more slowly.

## The pelvic region

The pelvic and caudal regions show three vertebrae in total: two sacrals and one caudal. The sacral vertebrae (see Fig. 6) typically bear large but short lateral processes, each pointing towards each other in order to support the ilium. Indeed, the lateral process shows a sturdy base (see Table 4, especially when compared to the centrum length) which flares out distally, forming again two distinct rami. In S1, the anterior ramus exceeds the posterior one and vice versa in S2 (see Table 5). The distal facets of the rami are clearly different from one another (see Fig. 6): the longest ramus (i.e. either the anterior one, here missing, in S1 or the posterior one in S2) is anteroposteriorly elongated but extremely flattened in the perpendicular direction; the shortest ramus takes the shape of an anteroposteriorly elongated triangle with a vertex pointing ventrally. These short rami are oriented towards the junction between S1 and S2 and were probably the main support for the ilium. The relatively short length of the lateral process compared to the centrum places the pelvic girdle closer to the axial skeleton than it is the case in metriorhynchids (e.g. *Thalattosuchus superciliosus* NMI F21731; *pers. obs.*).

The lateral process occupies almost the whole length of each centrum but it is not centered (see Fig. 6): the lateral process stems from the anteriormost portion of the

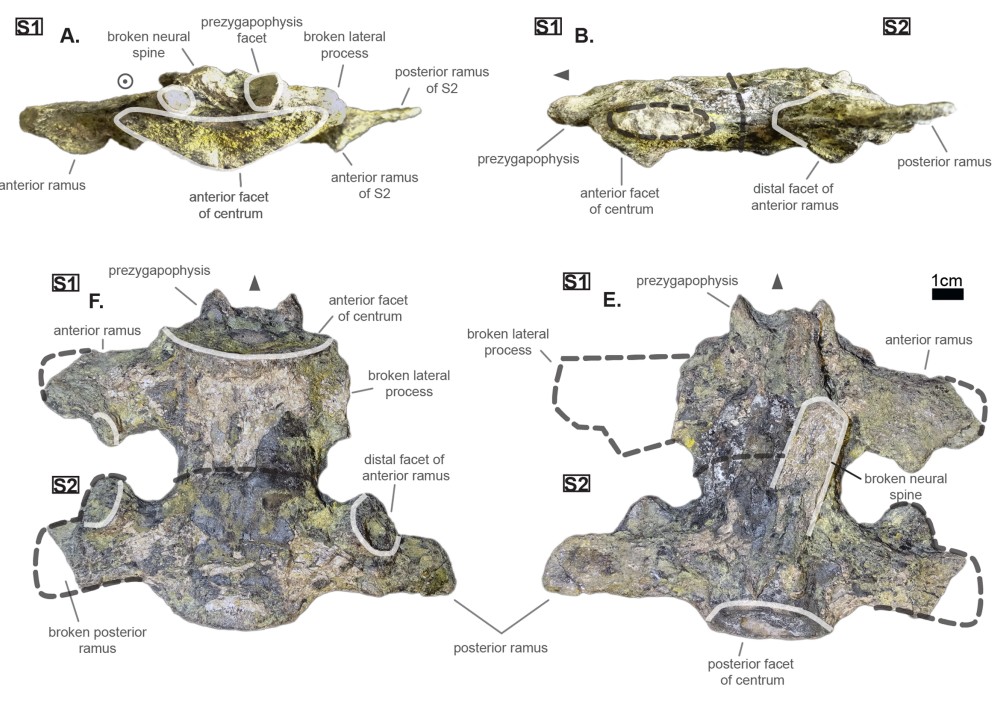

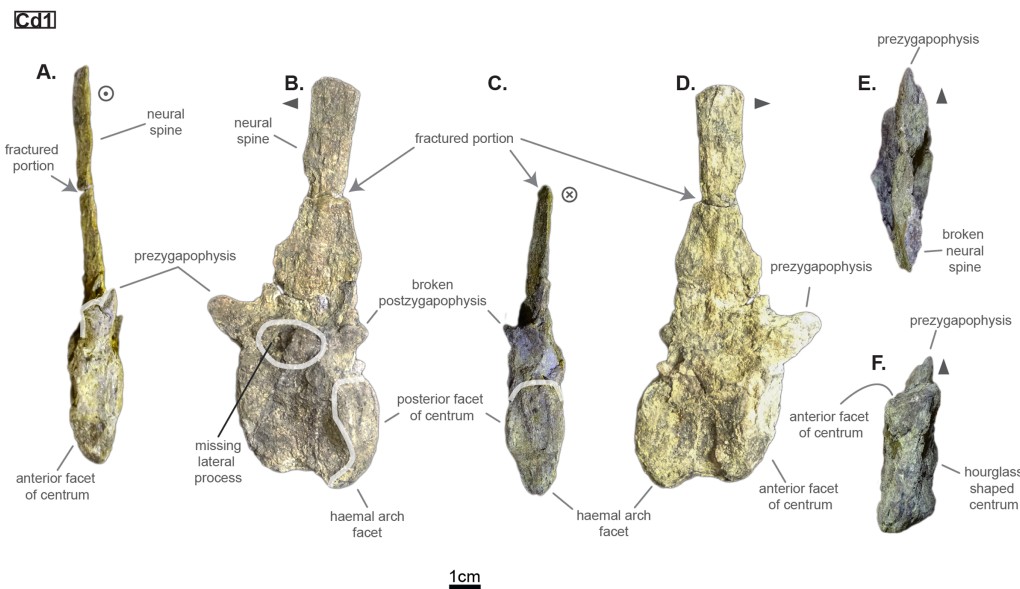

**Figure 6 Sacrals and caudal of *Cerrejonisuchus improcerus* UF/IGM 31.** (A) Anterior view; (B) left lateral view; (C) posterior view; (D) right lateral view; (E) dorsal view; (F) ventral view. Grey arrow points towards anterior.

centrum in S1 while it is located posteriorly in S2 (see Fig. 6). The neural arch is missing on both sacrals, but it appears that the lateral process is entirely born by the centrum.

The anterior facet of S1 slightly resembles the heart-shaped centrum of the anterior thoracics (see Fig. 3), it is however dorsoventrally flattened (see Fig. 6). Ventrally, a crest (or keel) is issued by the anterior facet but fades away before reaching the center of the

Table 4 **Table depicting each sacral vertebrae (ordered) and some measurements.** 'CH' represents the maximal height of the centrum (dorso-ventrally); 'CW' represents the maximal width of the centrum (laterally); 'CL' represents the anterior-posterior length of the centrum; 'N ang' represents the angle (whole number) that the neural spine forms with the horizontal (coronal plane); 'NH' represents the height of the neural spine; 'Hyp H' represents the maximal height of the hypapophyse, 'Lat W' represents the base width (antero-posterior) of the lateral process; 'Lat H' represents the base height (dorso-ventral) of the lateral process; 'Lat a H' represents the height of the anterior distal facet of the lateral process and is taken perpendicular to 'Lat a W'; 'Lat p H' represents the height of the distal diapophyseal facet of the lateral process and is taken perpendicular to 'Lat p W'; 'Lat a W' stands for the greatest length of the distal parapophyseal facet (which may be tilted regarding the antero-posterior plane) of the lateral process; 'Lat p W' stands for the greatest length of the distal diapophyseal facet (which may be tilted regarding the antero-posterior plane) of the lateral process; 'Lat min' represents the proximal-distal length of the shortest ramus of the lateral process; 'Lat Maj' represents the proximal-distal length of the greatest ramus of the lateral process; 'Prez Maj' represents the length of the major axis of the elliptic surface of the prezygapophysis; 'Postz Maj' represents the length of the major of the elliptic surface of the post-zygapophysis; 'Prez min' represents the length of the minor axis of the elliptic surface of the pre-zygapophysis (and is usually perpendicular to the corresponding 'Prez Maj'); 'Postz min' represents the length of the minor axis of the elliptic surface of the postzygapophysis (and is usually perpendicular to the corresponding 'Postz Maj'); 'Prez ang' refers to the angle (degree) between the horizontal plane (or coronal plane) and prezygapophysis; 'Postz ang' refers to the angle (degree) between the horizontal plane (or coronal plane) and the postzygapophysis. The lowercase letters 'a' and 'p' stand for 'anterior' and 'posterior' respectively.

| | S1 | S2 |
|---|---|---|
| CH a (mm) | — | — |
| CH p (mm) | — | — |
| CW a (mm) | 41.35 | — |
| CW p (mm) | — | — |
| CL (mm) | 43.36 | 45.88 |
| N ang (°) | — | — |
| NH (mm) | — | — |
| Hyp height (mm) | — | — |
| Lat W (mm) | 30.51 | 32.39 |
| Lat H (mm) | 14.38 | — |
| Lat a H (mm) | — | 11.49 |
| Lat p H (mm) | 16.53 | — |
| Lat a W (mm) | — | 19.44 |
| Lat p W (mm) | 21.13 | 17.13 |
| Lat Maj (mm) | — | 51.68 |
| Lat min (mm) | 30 | 27.63 |
| Prez Maj (mm) | — | — |
| Postz Maj (mm) | — | — |
| Prez min (mm) | — | — |
| Prez min (mm) | — | — |
| Prez ang (°) | — | — |
| Postz ang (°) | — | — |
| NH/CW a (%) | — | — |
| NH/CH a (%) | — | — |
| Lat p W/Lat Maj (%) | — | 33.15 |
| Lat a W/Lat min (%) | — | 70.36 |
| Lat Maj/Lat min (%) | — | 187.04 |
| Lat Maj/CW a (%) | — | — |

**Table 5 Table depicting each caudal vertebrae (ordered) and some measurements (part 1).** 'CH' represents the maximal height of the centrum (dorso-ventrally); 'CW' represents the maximal width of the centrum (laterally); 'CL' represents the anterior-posterior length of the centrum; 'N ang' represents the angle (whole number) that the neural forms with the horizontal; 'NH' represents the height of the neural spine. The lowercase letters 'a' and 'p' stand for 'anterior' and 'posterior' respectively.

| | CH a (mm) | CH p (mm) | CW a (mm) | CW p (mm) | CL (mm) | N ang (°) | NH (mm) | NH/CH a (%) |
|---|---|---|---|---|---|---|---|---|
| Cd'1' | 24.85 | 31.72 | — | — | 34.32 | 90 | 71.21 | 286.56 |

centrum. S2, on the contrary, shows an oval-shaped posterior facet and does not bear any ventral keel.

The preservation state of the sacrals makes it difficult to assess whether or not the bones were fused in vivo to form a sort of sacrum. Yet, all other vertebrae of *Cerrejonisuchus improcerus* are disrupted, and those fused together (apart from the sacrals) are actually all belonging to distinct portions of the axial skeleton and are also not crushed in the same fashion (see Figs. 1, 3, and 6). Hence, there is a hypothetical possibility that the preservation of the sacrals is due to their solid connection in vivo.

## The caudal region

The sole caudal vertebra retrieved, **UF/IGM 31 Cd'1'** (see Fig. 6), belongs to a rather anterior portion of the tail: its neural spine is long, vertical, and after a thick base, becomes rapidly finer distally; there is some evidence of a lateral process born on the neural arch; and the posterior facet has a ventral surface reserved for the haemapophysis.

Indeed, both *Hyposaurus natator* (NJSM 23368; *pers. obs.*) and *Congosaurus bequaerti* (Holotype, MRAC; *pers. obs.*) show an increase in the size of the neural spine in the anterior portion of the caudal region, which is even more emphasized because the sacrals and posterior caudals possess a shorter neural. Here, the caudal vertebra of *Cerrejonisuchus improcerus* has a 71.2 mm long neural for a 24.85 mm long anterior facet, which gives a ratio value close to the anterior thoracics (see Table 5) not unlike the hyposaurine dyrosaurids. Hence, the presence of long neural spines on the caudal of *Cerrejonisuchus* (see Fig. 6) shows that basal dyrosaurids (*Young et al., 2016*) had already developed a massive tail. The neural spine of Cd'1' also has its posterior and anterior surfaces parallel, giving it a vertical look, with a humped distal extremity like hyposaurine dyrosaurids (i.e. *Hyposaurus rogersii* NJSM 23368 and holotype of *Congosaurus bequaerti*; *pers. obs.*). Cd'1' resembles the 9th caudal of *Congosaurus* (MRAC 1892) with the swollen base of its neural rapidly slimming down distally, coupled with its relative vertical orientation. For this reason, the caudal of *Cerrejonisuchus* must have belonged somewhere around the 10th position.

On the lateral sides of Cd'1', around the base of the neural arch, are circular scars (see ??) indicating the former position of the lateral process as in modern crocodylians (*de Souza, 2018*). The lateral process usually fades away posteriorly along the caudal vertebra (see Fig. 6) as seen in crocodylians (such as *Crocodylus porosus* Aquarium–Museum Liège R.G.294 or *Mecistops cataphractus* RBINS 18374; *pers. obs.*) or in *Congosaurus bequaerti* (Holotype, MRAC 1852 & 1879; *pers. obs.*).

There is not evidence of a hyposphene structure preserved on this caudal.

The facets of Cd'1' are slightly larger dorsally than ventrally, and their outline resembles that of an apple. This type of shape is also found in the tail of other crocodyliforms such as thalattosuchians (e.g. *Thalattosuchus superciliosus* SMNS 10116 4th caudal; *pers. obs.*), hyposaurine dyrosaurids (e.g. *Congosaurus bequaerti* holotype, caudal numbered MRAC 1837; *pers. obs.*); or crocodylians (e.g. *Mecistops cataphractus* RBINS 18374 5th caudal; *pers. obs.*).

In this portion of the skeleton, the centrum is longer (anteroposteriorly) than it is high or wide. This feature is rather common among crocodyliforms, e.g.: *Mecistops cataphractus* (RBINS 18374; *pers. obs.*); *Terminonaris browni* (AMNH FARB 5844; *pers. obs.*); Hyposaurinae indet. (AMNH FARB 2390; NJSM 12293; *pers. obs.*); *Congosaurus bequaerti* (holotype, vertebra numbered MRAC 1846; *pers. obs.*); or even in *Machimosaurus buffetauti* (SMNS 91415; *pers. obs.*).

## Ribs

The cervical ribs of *Cerrejonisuchus improcerus* (UF/IGM 31) are all still attached to the four cervical vertebra. Only the cervical C'2' possesses the complete pair. The cervical ribs bear the typical crocodyliform shape in resembling a 'T' in dorsal and ventral view, with the anterior portion shorter than the posterior one (see Fig. 1).

*Cerrejonisuchus improcerus* UF/IGM 31 only possesses two thoracic ribs, which belonged to a middle portion of the rib-cage (see Fig. 7; see Fig. 8). The thoracic ribs of UF/IGM 31 differ from those of *Congosaurus bequaerti* and *Hyposaurus natator* (NJSM 23368) in being relatively more convex. In doing such, they appear to contrast with both the thoracic skeletons observed in hyposaurine dyrosaurids and modern crocodylians. Indeed, the ribs of *Cerrejonisuchus improcerus* show a bending further distally than in hyposaurines, almost situated at the mid length of the bone which gives the bone an arched aspect (see Fig. 7). The whole lateral outline of the rib is convex, whereas in *Congosaurus bequaerti* (e.g. mid thoracic rib MRAC 1743 from Fig. 7) the rib straightens distally after the bending. Indeed, in *Cerrejonisuchus* (UF/IGM 31) the concavity of the rib does not appear to change before and after the bending as opposed to *Crocodylus porosus* (Aquarium–Museum Liège R.G.294) or *Congosaurus bequaerti* (MRAC 1743) (*Schwarz-Wings, Frey & Martin, 2009*). In the thoracic region, we hypothesized little to no dorsal deviation of the lateral process bearing the thoracic ribs. For these reasons, the bracing of the trunk of *Cerrejonisuchus* appears more cylindrical (see Fig. 8) than in *Crocodylus porosus* (Aquarium–Museum Liège R.G.294), and also less elevated dorsally than in hyposaurine dyrosaurids (*Schwarz-Wings, Frey & Martin, 2009*). It appears more similar to *Anteophtalmosuchus hooleyi* (*Martin et al., 2016*).

Compared to *Cerrejonisuchus improcerus* (UF/IGM 31), the thoracic ribs of *Anthracosuchus balrogus* (UF/IGM 67) are short and wide, with a shallower concavity (see Fig. 7). The ribs of *Anthracosuchus balrogus* (UF/IGM 67) also possess an enlarged distal extremity which would have been connected to a sternum of some kind. This difference stems from the absolute position of the ribs in the axial skeleton. Indeed, the ribs of *Anthracosuchus balrogus* (UF/IGM 67) are more anterior than those of *Cerrejonisuchus*

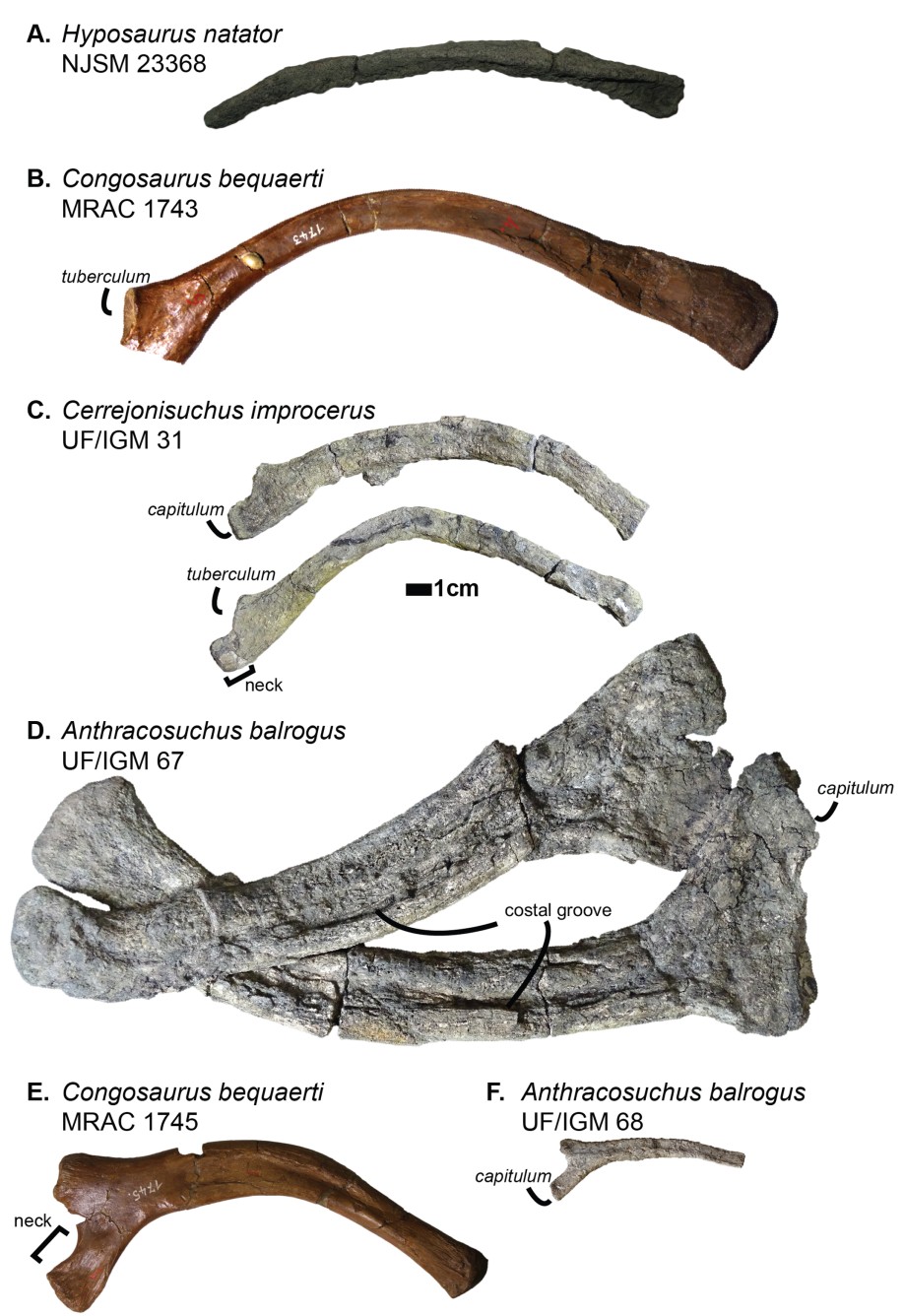

**Figure 7** **Scaled dyrosaurid thoracic ribs. Scale bar represents one cm.** (A–C) Middle thoracic ribs, (D–F) anterior thoracic ribs. (A) *Hyposaurus natator* NJSM 23368; (B) *Congosaurus bequaerti* MRAC 1743; (C) *Cerrejonisuchus improcerus* UF/IGM 31; (D) *Anthracosuchus balrogus* UF/IGM 67; (E) *Congosaurus bequaerti* MRAC 1745; (F) *Anthracosuchus balrogus* UF/IGM 68. Picture of *Hyposaurus natator* courtesy of Wayne Callahan.

*improcerus* (UF/IGM 31) thanks to the presence of a neck on both the capitulum and tuberculum (see Fig. 7).

The relative position of the *capitulum* and *tuberculum* of *Cerrejonisuchus*, which resemble *Anthracosuchus* and *Congosaurus* (see Fig. 7), would hint at a similar vertical

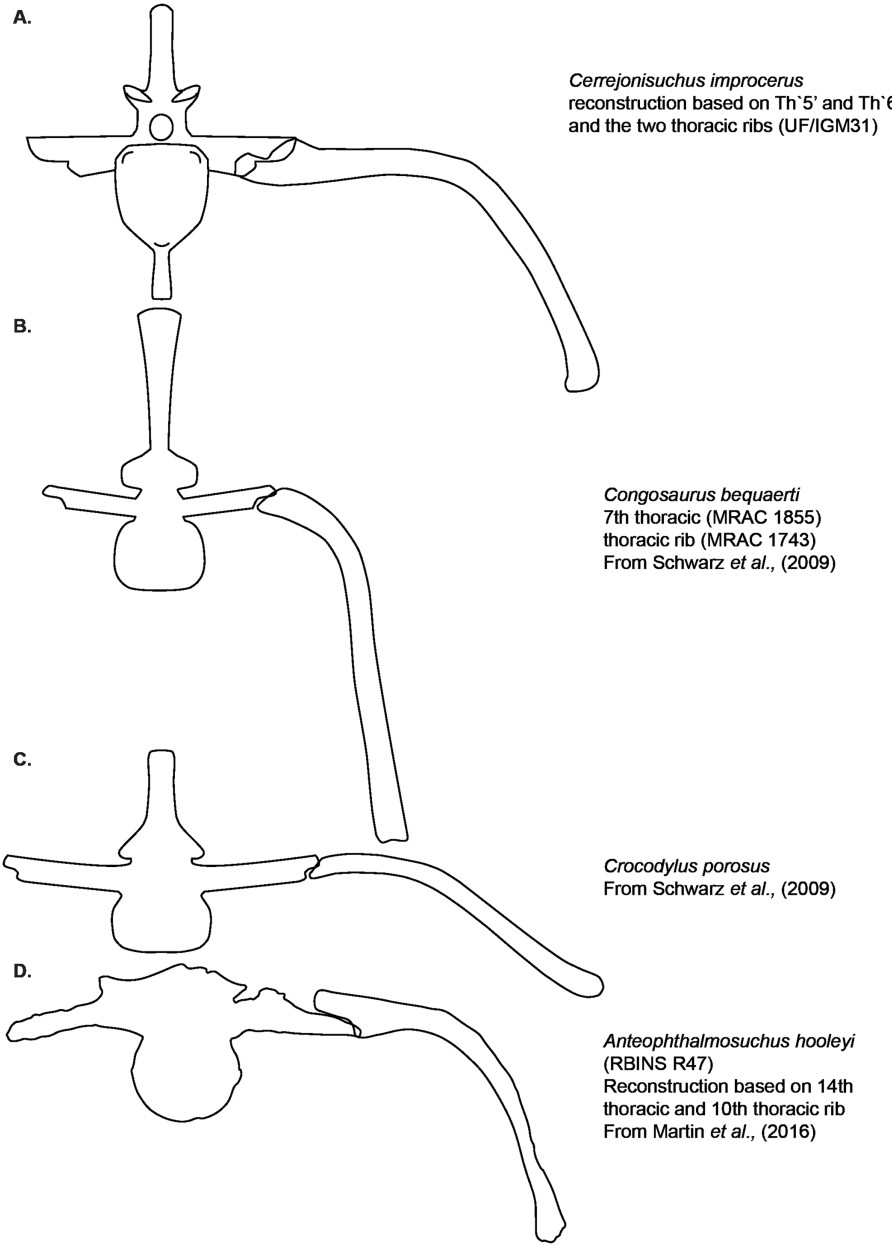

A. *Cerrejonisuchus improcerus* reconstruction based on Th`5' and Th`6' and the two thoracic ribs (UF/IGM31)

B. *Congosaurus bequaerti* 7th thoracic (MRAC 1855) thoracic rib (MRAC 1743) From Schwarz *et al.,* (2009)

C. *Crocodylus porosus* From Schwarz *et al.,* (2009)

D. *Anteophthalmosuchus hooleyi* (RBINS R47) Reconstruction based on 14th thoracic and 10th thoracic rib From Martin *et al.,* (2016)

**Figure 8 Hypothetical reconstruction of a section of the trunk of *Cerrejonisuchus improcerus* based on the thoracic vertebra UF/IGM 31 Th'5' and Th'6' and the two thoracic ribs.** Comparison with reconstruction of the crocodylian, hyposaurine trunk bracing from *Schwarz-Wings, Frey & Martin, 2009*, and reconstruction of the goniopholid trunk bracing from *Martin et al. (2016)*. (A) *Cerrejonisuchus improcerus*; (B) *Congosaurus bequaerti*; (C) *Crocodylus porosus*; (D) *Anteophtalmosuchus hooleyi*.

orientation of muscle attachments on the rib (*Schwarz-Wings, Frey & Martin, 2009*; *Hastings, Bloch & Jaramillo, 2014*). Yet, the relationship between the parapophyseal and diapophyseal processes of the thoracic vertebrae whose position could correspond to that of the thoracic ribs of *Cerrejonisuchus improcerus* (see Th'5' and Th'6' on Fig. 3) seems to indicate a rather horizontal orientation of the said processes (i.e. they do not arrange in

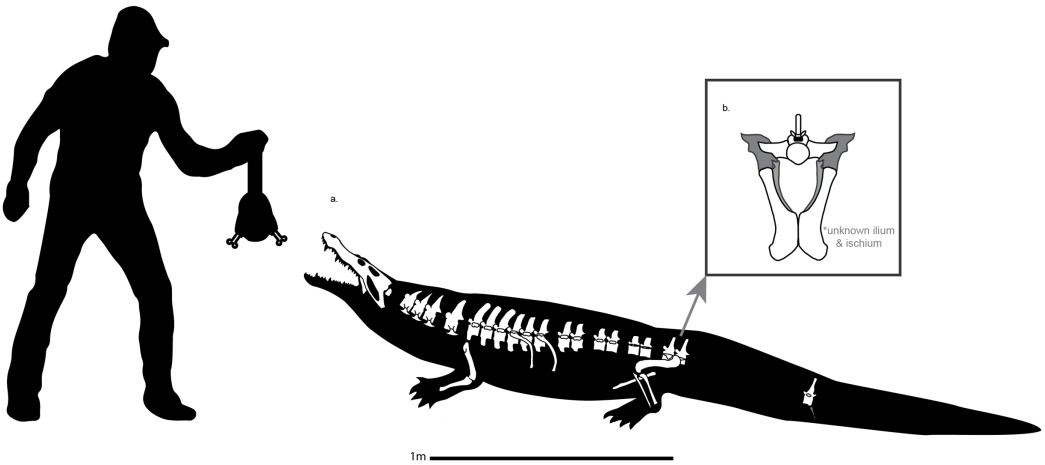

**Figure 9** Schematic reconstruction of *Cerrejonisuchus improcerus* based on the specimen UF/IGM 31. Broken vertebrae have been hypothetically recreated. The precise position of several dorsal (thoracic & lumbar) vertebrae is putative, and based on our morphological comparisons with other taxa. Please do not use this artistic reconstruction to score phylogenetic characters.

tiers vertically, unlike *Congosaurus bequaerti* MRAC 1855 & 1874). Thus, the thoracic ribs of *Cerrejonisuchus* have presumably been flatten and were probably arched posteriorly (see Fig. 9) in the way of modern crocodylians.

## APPENDICULAR SKELETON ANATOMY

### Humerus

The humerus of *Cerrejonisuchus improcerus* (UF/IGM 31, see Fig. 10 Humerus; see Table 6) is similar to those of *Hyposaurus natator* (NJSM 23368; *pers. obs.*) and *Congosaurus bequaerti* (MRAC 1813; *pers. obs.*) by presenting a straight shaft, and lacking the proximal torsion characteristic of crocodylians (e.g. *Mecistops cataphractus*; see also *Stein et al. (2012)*). The humerus of *Cerrejonisuchus improcerus* measures 146.8 mm which accounts for 38.03% of the total skull length (i.e. snout tip to posterior-most portion of the quadrate) and for 83.41% of the femur's length (a similar ratio has been observed among terrestrial crocodyliforms such as *Tarsomordeo winkleri* (*Adams, 2019*)). This number (83.41%) falls under those observed for both *Congosaurus bequaerti* (MRAC 1813 & 1815 ; *pers. obs.*) and *Hyposaurus natator* (NJSM 23368, *pers. obs.*) with 96.32% and 94.47%, respectively. *Cerrejonisuchus improcerus* possessed a humerus shorter both proportionally and globally compared to the hyposaurine dyrosaurids (*Congosaurus bequaerti* MRAC 1813 reached 288.73 mm and *Hyposaurus natator* NJSM 23368 is 188 mm long, *pers. obs.*), which does not fit with the diagnostic limit of 90% set by *Jouve et al. (2006)* for Dyrosauridae. Instead, this trait (i.e. humerus attaining 90% of the femur length) could become mainly indicative of derived dyrosaurids within Dyrosauridae, or could be lowered to 80% to include the basal *Cerrejonisuchus improcerus*. Unfortunately, the stylopodia were not completely recovered for the other two basal

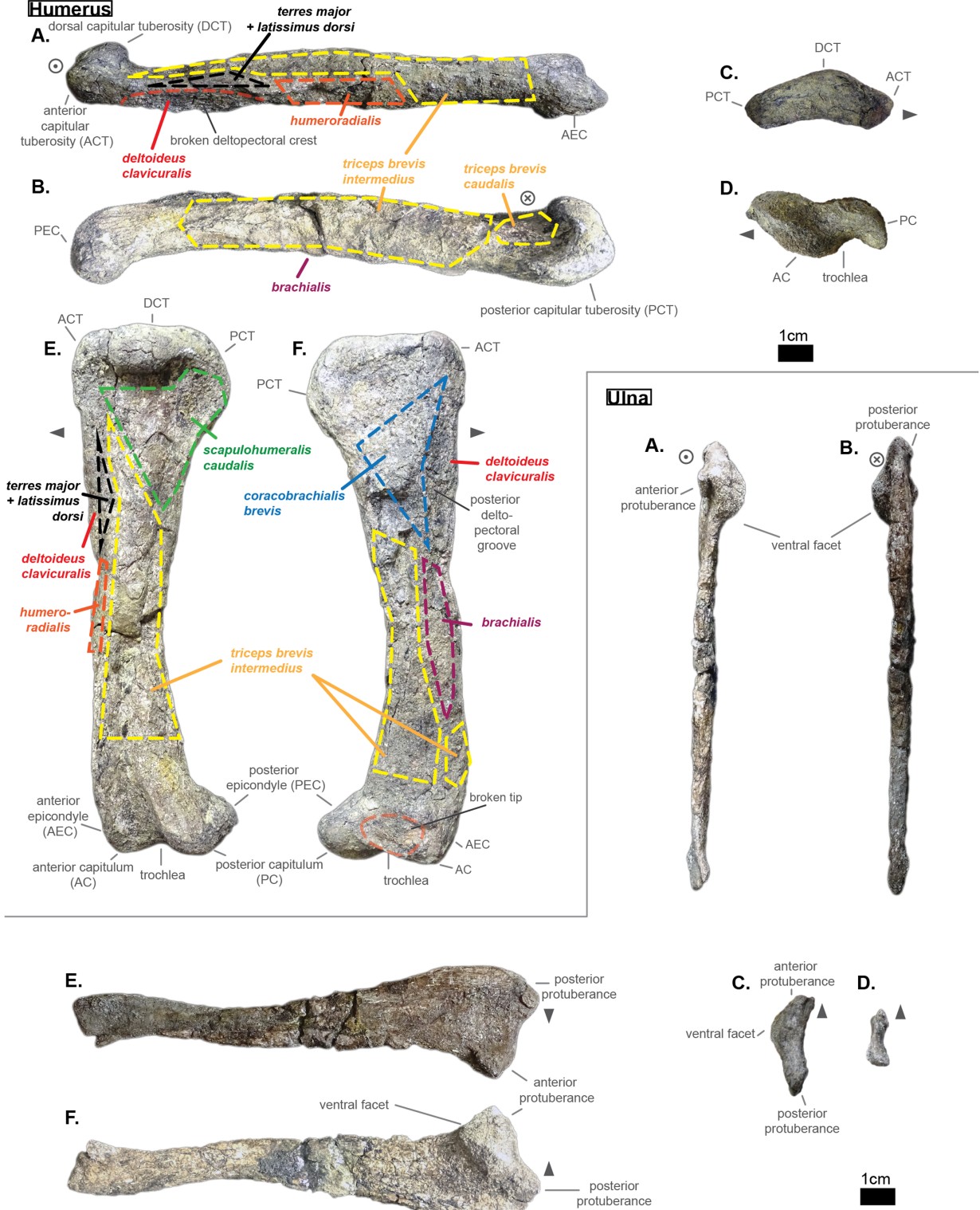

**Figure 10** **Left humerus and right ulna of *Cerrejonisuchus improcerus* UF/IGM 31.** (A) Anterior view; (B) posterior view; (C) proximal view; (D) distal view; (E) dorsal view; (F) ventral view. Grey arrow points towards anterior. Relative position of muscles on the humerus based on the framework of *Meers (2003).*

| Table 6 Table depicting limb proportions for Cerrejonisuchus improcerus. | | |
|---|---|---|
| **Bone** | | **Measurements in mm** |
| Femur | Total length | 176 |
| | Thickness at mid-length | 18.1 |
| | Width at mid-length | 27.21 |
| | Proximal head length | 43.64 |
| | Proximal head width | 13.1 |
| | Posterior capitulum length | 35.28 |
| | Anterior capitulum length | 27.51 |
| | Distal end total width | 18.72 |
| Tibia | Total length | 151.96 |
| | Thickness at mid-length | 9.32 |
| | Width at mid-length | 18.59 |
| | Proximal head length | 40.57 |
| | Proximal head width | 12.25 |
| | Distal end width | 28.96 |
| | Distal end length | 7.85 |
| Fibula | Total length | 140.94 |
| | Thickness at mid-length | 6.61 |
| | Width at mid-length | 11.54 |
| | Proximal head length | 22.59 |
| | Proximal head width | 5.08 |
| | Distal end width | 9.44 |
| | Distal end length | 17.72 |
| Humerus | Total length | 146.8 |
| | Thickness at mid-length | 20.33 |
| | Width at mid-length | 16.14 |
| | Proximal head length | NA |
| | Proximal head width | 44.11 |
| | Distal end width | 28.33 |
| | Distal end length | NA |
| Ulna | Total length | 127.35 |
| | Thickness at mid-length | 4.85 |
| | Width at mid-length | 13.93 |
| | Proximal head length | 11.59 |
| | Proximal head width | 29.25 |
| | Distal end width | 14.85 |
| | Distal end length | 2.95 |

dyrosaurids (i.e. *Anthracosuchus balrogus* & *Acherontisuchus guajiraensis*) from the same locality (*Hastings, Bloch & Jaramillo, 2011*, *2014*).

However, the 90% ratio of *Jouve et al. (2006)* can hardly be used as a diagnostic character of Dyrosauridae within Crocodyliformes because many crocodylians also go beyond this limit, notably: *Alligator mississippiensis* (86–94%), *Alligator sinensis* (82–90%), *Caiman*

*yacare* (76–94%), *Caiman crocodilus* (80–90%), *Crocodylus acutus* (89–96%), *Crocodylus moreletii* (90–96%), *Crocodylus palustris* (87–94%), *Crocodylus porosus* (89–92%), *Tomistoma schlegelii* (81–93%) (*Iijima, Kubo & Kobayashi, 2018*). The wide range of numbers for some species bears witness to a large amount of intraspecific variation which is almost impossible to take into account for fossil species.

*Cerrejonisuchus improcerus* is missing the manus and pes: therefore, the total limb length is restricted to the zeugopodium and stylopodium sum. Following this, the total forelimb and hindlimb lengths of *Cerrejonisuchus* reaches 274.15 mm (right ulna and left humerus) and 327.96 mm (right tibia and left femur) respectively. This difference in length between the limbs of *Cerrejonisuchus* highly contrasts with the hyposaurine dyrosaurids, where forelimb and hindlimb are similarly proportioned (*Denton, Dobie & Parris, 1997*) (e.g. *Hyposaurus rogersii* NJSM 23368 forelimb reaches 91.45% of hindlimb).

At its mid-length, the shaft presents an oval section whose greatest axis is parallel to the anterior-posterior direction; its measurements are: 20.33 mm for the greatest axis, versus 16.14 mm for the minor axis. The deltopectoral process is located more anteriorly than in *Mecistops cataphractus* (for which it is more ventrally positioned). Unfortunately, the actual deltopectoral process is broken off and only its outline remains, which stretches up to 44 mm which is almost 30% of the humerus' total length.

The humerus is also dorsoventrally flattened, and it can be seen easily from the squeezed distal capitula (see Fig. 10 Humerus D). In dorsal view, the overall shape of the bone is upright: its anterior surface forms roughly a straight line (since the deltopectoral crest is broken) while its posterior surface is slightly concave (which may appear accentuated by the flattened distal extremity). There is no proximal torsion of the humerus as in modern crocodylians (*Stein et al., 2012*). In anterior view, the shaft is also upright with a slight sigmoid shape. This is a condition also observed in *Hyposaurus natator* (NJSM 23368, *pers. obs.*) and *Congosaurus bequaerti* (MRAC 1813; *pers. obs.*) meaning that the flattening of the bone has potentially little influence on the shape. However, there are other specimens of crocodyliforms where this sigmoid shape is even more accentuated in anterior and dorsal view, such as *Crocodylus rhombifer* (AMNH FARB 16697; *pers. obs.*), or Hyposaurinae indet. (AMNH FARB 2202; *pers. obs.*).

The anterior epicondyle (or AEC) of *Cerrejonisuchus improcerus* is partially broken in the anterior and ventral directions (see Fig. 10 Humerus F), but its distal outline is preserved and can be observed in dorsal view (see Fig. 10). In distal view, the anterior epicondyle greatly exceeds the size of the posterior epicondyle (or PEC), exactly like in *Congosaurus bequaerti* (MRAC 1813; *pers. obs.*). In *Hyposaurus natator* (NJSM 23368; YPM VP.000985; *pers. obs.*), this difference is present as well but is less marked so that both epicondyles may appear similar. In some crocodylians and crocodyliforms, there are practically no differences between the condyles as for example in: *Crocodylus niloticus* (NHMW 30900; *pers. obs.*), *Alligator sinensis* (NHMW 37966; *pers. obs.*) or *Osteolaemus tetraspis* (NHMW 39338:2; *pers. obs.*) for the crocodylians (all of which are found in different environments); or in *Terminonaris browni* (AMNH FARB 5844; *pers. obs.*). There are, however, counterexamples such as *Caiman crocodilus* (NHMW 31137; *pers. obs.*),

*Crocodylus rhombifer* (AMNH FARB 16697; *pers. obs.*) or *Mecistops cataphractus* (RBINS 18374; *pers. obs.*).

In dorsal view, the length of both epicondyle is almost the same with the PEC extending slightly more distally (see Fig. 10). This is also a feature observed on both *Hyposaurus natator* (NJSM 23368, *pers. obs.*) and *Congosaurus bequaerti* (MRAC 1813; *pers. obs.*). These differences in both epicondyles presumably influenced the positioning and the ROM of the zeugopodia, but unfortunately *Cerrejonisuchus improcerus* is here missing the radius to further test this hypothesis (just like *Congosaurus bequaerti*, holotype *pers. obs.*). Besides, the ulna makes most of the elbow articulation as observed from crocodylians (e.g. *Mecistops cataphractus* RBINS 11839), and thus the modification of the proximal end of this bone could actually be the main influence over the humeral epicondyle shapes.

In dorsal view, *Cerrejonisuchus improcerus* shows a wider proximal end (encompassing all three tuberosities, see Fig. 10) than the distal one (with 44.11 mm and 28.33 mm, respectively), whereas in *Congosaurus bequaerti* (MRAC 1813; *pers. obs.*) and *Hyposaurus natator* (NJSM 23368, *pers. obs.*) both distal and proximal ends are of similar width. The proximal end of those humeri (i.e. the aforementioned *Cerrejonisuchus improcerus*, *Congosaurus bequaerti* MRAC 1813 and *Hyposaurus natator* NJSM 23368; YPM VP.000985; *pers. obs.*) are composed of three tuberosities which are observable on the dorsal side of the articulation. On their ventral side, the proximal articulation is concave, however, thus giving the look of an arrowhead in proximal view like in many other crocodyliforms (e.g. *Alligator sinensis* NMW 37966; *Dacosaurus maximus* SMNS 8203 or *Terminonaris browni* AMNH FARB 5844; *pers. obs.*).

Still in anterior view, the anterior capitular tuberosity (i.e. ACT) and the dorsal capitular tuberosity (i.e. DCT) of *Cerrejonisuchus improcerus* both have their peak anteriorly oriented, while that of the posterior capitular tuberosity (i.e. PCT) is strictly posterior (see Fig. 10). The DCT is mainly responsible for the articulation of the humerus with the scapular girdle, and for this reason it is the biggest of the three tuberosities. The DCT is of similar dimensions (regarding the other tuberosities) in other crocodyliforms, as for example in *Alligator sinensis* (NMW 37966; *pers. obs.*) or in *Terminonaris browni* (AMNH FARB 5844; *pers. obs.*). It is however much more protruding in Hyposaurinae indet. (AMNH FARB 2202, 19205 ; *pers. obs.*), *Congosaurus bequaerti* (MRAC 1813; *pers. obs.*) or in some other crocodyliforms such as *Crocodylus niloticus* (NHMW 30900; *pers. obs.*) or *Osteolaemus tetraspis* (NHMW 39338:2; *pers. obs.*).

The PCT is also protruding posteriorly, thus exaggerating the convex shape of the posterior surface of the bone. Conversely, the ACT does not really protrude anteriorly and is undoubtedly the smallest of the three proximal capitular tuberosities (see Fig. 10).

## Ulna

The only preserved ulna of *Cerrejonisuchus improcerus* (see Fig. 10) unfortunately belongs to the opposite limb of the only preserved humerus: it is a right ulna. It is recognizable thanks to the posteroventral facet of the proximal extremity which meets with a portion of the radius. The total length of the ulna measures 127.35 mm, which is almost as long as the left humerus (thus the ulna accounts for 86.75% of the humerus, see Table 6).

This proportional length could be a more terrestrial feature, similarly to what is observed among Crocodylia (*Iijima, Kubo & Kobayashi, 2018*). The ulna of both *Congosaurus bequaerti* (MRAC 1816; *pers. obs.*) and *Hyposaurus natator* (NJSM 23368, *pers. obs.*) are proportionally shorter than that of *Cerrejonisuchus* because they respectively reach 74.1% and 73.9% of their corresponding humerus.

The ulna of *Cerrejonisuchus improcerus* is dorsoventrally flattened and is thus the thickest in the anterodorsal plane. The proximal extremity is 29.25 mm wide (dorsoventrally) and 11.59 mm thick in the perpendicular direction. Its distal extremity reaches 14.85 mm in the anteroposterior plane, and is 2.95 mm thick dorsoventrally. It is difficult to address the degree of flattening of the bone, but since the proximal extremity shows a subtriangular shape (see Fig. 10) it is more likely that the ulna of *Cerrejonisuchus improcerus* resembled those of *Congosaurus bequaerti* (MRAC 1816; *pers. obs.*) and *Hyposaurus natator* (NJSM 23368, *pers. obs.*): i.e. the ulna must have been relatively flat distally but thicker proximally. It is a trait which appears to be shared among some crocodyliforms at least (e.g. *Mecistops cataphractus* RBINS 18374; *Alligator sinensis* NMW 37966; *Osteolaemus tetraspis* NMW 39338:2; *pers. obs.*) although thalattosuchians do not seem to show this feature (e.g. *Platysuchus multiscrobiculatus* SMNS 9930; *pers. obs.*). Yet, it is likely that the ulna of *Cerrejonisuchus improcerus* may have not been as expanded in the dorsoventral plane as the derived dyrosaurids (*Schwarz-Wings, Frey & Martin, 2009*) or modern crocodylians as mentioned above.

In anterior view, the ulna takes an atypical sigmoid shape where the shaft is protruding anteriorly at more-or-less one-third of the distal extremity (see Fig. 10). Indeed, this shape is unlike those of *Congosaurus bequaerti* (MRAC 1816; *pers. obs.*) and *Hyposaurus natator* (NJSM 23368, *pers. obs.*) where the ulna possesses a rather straight shaft (i.e. anterior and posterior surfaces not undulating), with a slight and general dorsally curved trend (more emphasized at the proximal extremity). Yet, a similar sigmoid shape to *Cerrejonisuchus improcerus* can be found in other crocodyliforms, such as the goniopholid *Anteophtalmosuchus* (*Martin et al., 2016*), the baurusuchid *Pissarrachampsa sera* (*Godoy et al., 2016*) or the teleosaurid *Steneosaurus bollensis* (SMNS 9428, 15712a; *pers. obs.*) but with a lesser intensity. Still, it does not appear as a common shape among dyrosaurids (*i.e. Congosaurus bequaerti* MRAC 1816; *Hyposaurus natator* NJSM 23368; *pers. obs.*) or crocodylians (e.g. *Mecistops cataphractus* RBINS 18374; *Alligator sinensis* NMW 37966; *Osteolaemus tetraspis* NMW 39338:2; *pers. obs.*).

As mentioned earlier, the proximal articulation of the ulna takes the overall shape of a triangle. In ventral view, the proximal condyle almost looks like a heart whose lower part (i.e. the area encompassing the pointed tip; see Fig. 10 Ulna C) which met with the radius is well developed as in the baurusuchid *Pissarrachampsa sera* (*Godoy et al., 2016*) or in the crocodylian *Mecistops cataphractus* (RBINS 18374; *pers. obs.*). This surface is not as expanded in *Hyposaurus natator* (NJSM 23368, *pers. obs.*), and is even less in *Congosaurus bequaerti* (MRAC 1816; *pers. obs.*) in which most of the proximal articulation of the ulna was likely reserved to meet the humerus. The anterior and posterior protuberances, which make the top two rounded tips of the heart in ventral view, also show a dorsal depression between them that can be observed easily in proximal view

(see Fig. 10). In dorsal view though, the depression does not expand much, which could or could not be a consequence of the flattening. Besides *Hyposaurus natator* (NJSM 23368, *pers. obs.*) does not show any cavity on any side of its ulna, while *Congosaurus bequaerti* (MRAC 1816; *pers. obs.*) bears a large hollow area similar to *Cerrejonisuchus improcerus* on its dorsal side.

## Femur

The right femur of *Cerrejonisuchus improcerus* has not been recovered. The left femur of *Cerrejonisuchus improcerus* (see Fig. 11 Femur) displays the typical sigmoid silhouette (*Romer, 1923*) found in many other crocodyliforms (e.g. *Hyposaurus natator* NJSM 23368; *Cerrejonisuchus improcerus* MRAC 1815 & 1817; *Pelagosaurus typus* SMNS 80065, *pers. obs.*; *Wahasuchus egyptensis* (*Saber et al., 2018*), *Pissarrachampsa sera* (*Godoy et al., 2016*), *Mecistops cataphractus* RBINS 18374; see also Fig. 12). This S-shape is mostly apparent in dorsal and ventral view since the bone is flattened dorsoventrally (see Fig. 11 Femur E & F). It is therefore difficult to assess the degree of curvature in anterior or posterior view, yet a slight V-shape can be observed (see Fig. 11 Femur A & B). This kind of shape is also found on both femora of *Hyposaurus natator* (NJSM 23368), and is accentuated by both the size of the fourth trochanter on the ventral side of the bone, and the corresponding dorsal surface which forms a slight depression. Compared to other dyrosaurids, the femur of *Cerrejonisuchus improcerus* shows a strong sigmoid shape in relation to *Congosaurus bequaerti* (MRAC 1815 & 1817), but is similar to *Hyposaurus natator* (NJSM 23368) (see Fig. 12). However, it is not as pronounced as it is for *Acherontisuchus guajiraensis* (UF/IGM 39), an-other basal dyrosaurid from the same locality (*Hastings, Bloch & Jaramillo, 2011*). Indeed, both extremities of the femur of *Acherontisuchus guajiraensis* (UF/IGM 39) are protruding further away from the shaft than those of *Cerrejonisuchus improcerus* (UF/IGM 31). Yet both femora (from i.e. *Cerrejonisuchus improcerus* UF/IGM 31 & *Acherontisuchus guajiraensis* UF/GM 39) are dorsoventrally flattened, which supports that the compaction had little influence over the general shape of the bone (it only emphasized it).

In lateral (dorsal) view (see Fig. 11 Femur E), the proximal head of the femur of *Cerrejonisuchus improcerus* shows a strongly rounded (convex) outline, with a moderately impressive anterior protrusion which is here emphasized thanks to the presence of an anterior underlying depression (which is also partially responsible for highlighting the 'V-shape' in anterior view). In anteroposterior view (see Fig. 11 Femur) a & b), the femoral head does not have a convex outline (like *Hyposaurus natator* NJSM 23368 or *Hyposaurus* sp. DGM 803-R (*de Souza, 2018*)), or *Acherontisuchus guajiraensis* (UF/IGM 39) to a lesser extend) nor flat one (like *Congosaurus bequearti* MRAC 1815 & 1817), but has the shape of a sheared ogive (pointed arch) which appears unique to *Cerrejonisuchus improcerus* (UF/IGM 31). It is not known if there was a deep cavity on the ilium to meet with the femur's requirements, but such a rounded shape must have positively impacted the anterodorsal range of motion of the bone at the hip. However, regardless of the diagenetic flattening, the femoral head is not as smoothly curved dorsoventrally and may have inhibited some dorsal extension compared to a hypothetical perfectly round head.

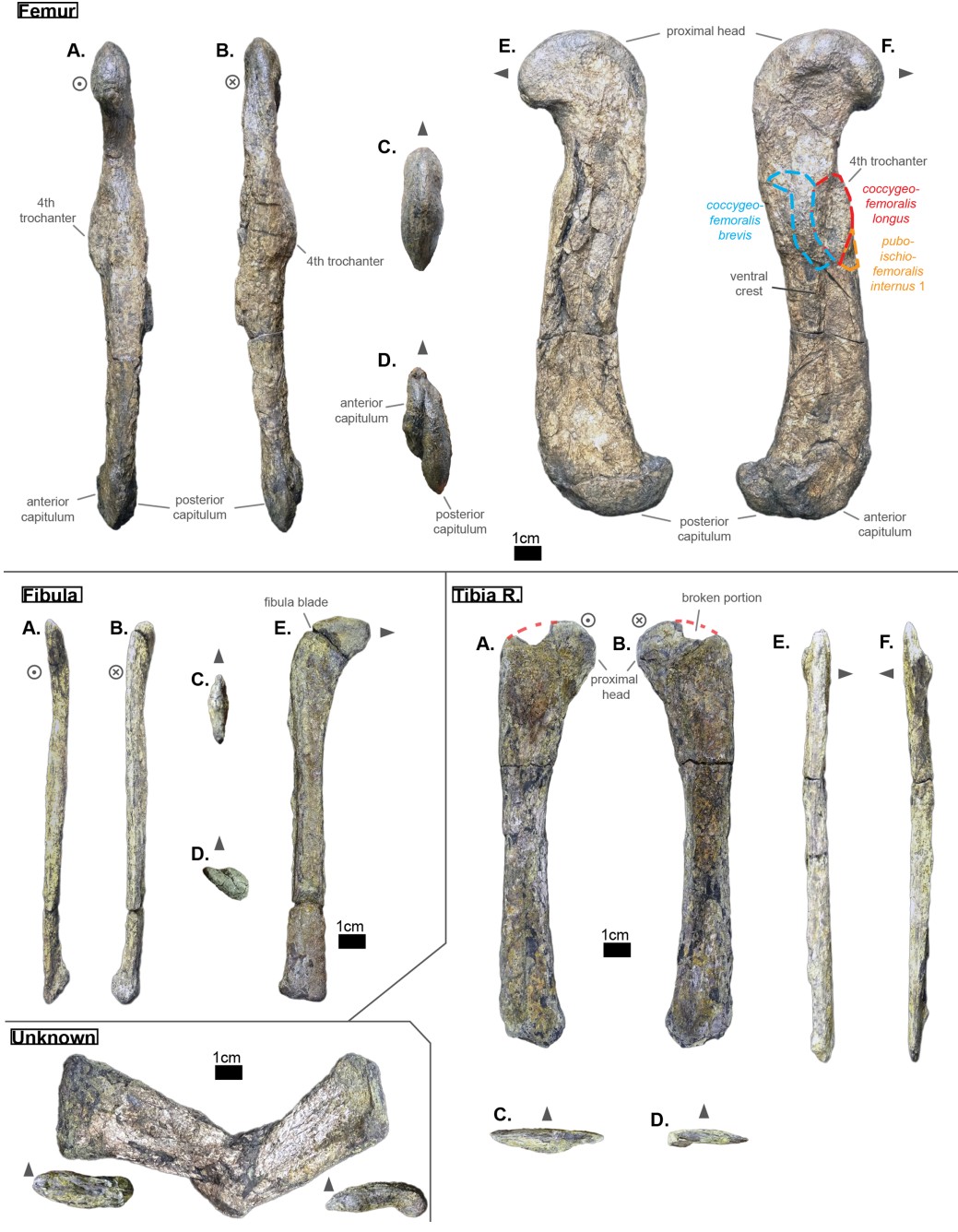

**Figure 11 Left femur, left fibula and right tibia of *Cerrejonisuchus improcerus* UF/IGM 31.** Mysterious bone featured in the bottom left corner. (A) Anterior view; (B) posterior view; (C) proximal view; (D) distal view; (E) dorsal view; (F) ventral view. Grey arrow points towards anterior. Relative position of muscles on the femur based on the framework of *Romer (1923)*.

Comparatively, the femoral heads of *Hyposaurus natator* (NJSM 23368), Hyposaurinae indet. (AMNH FARB 19204, 2202), and more importantly *Acherontisuchus guajiraensis* (UF/GM 39) show a strong anterior protrusion (see Fig. 12). This can however be partially explained through a more intensely sigmoid shape of the bone (i.e. the posterior surface of

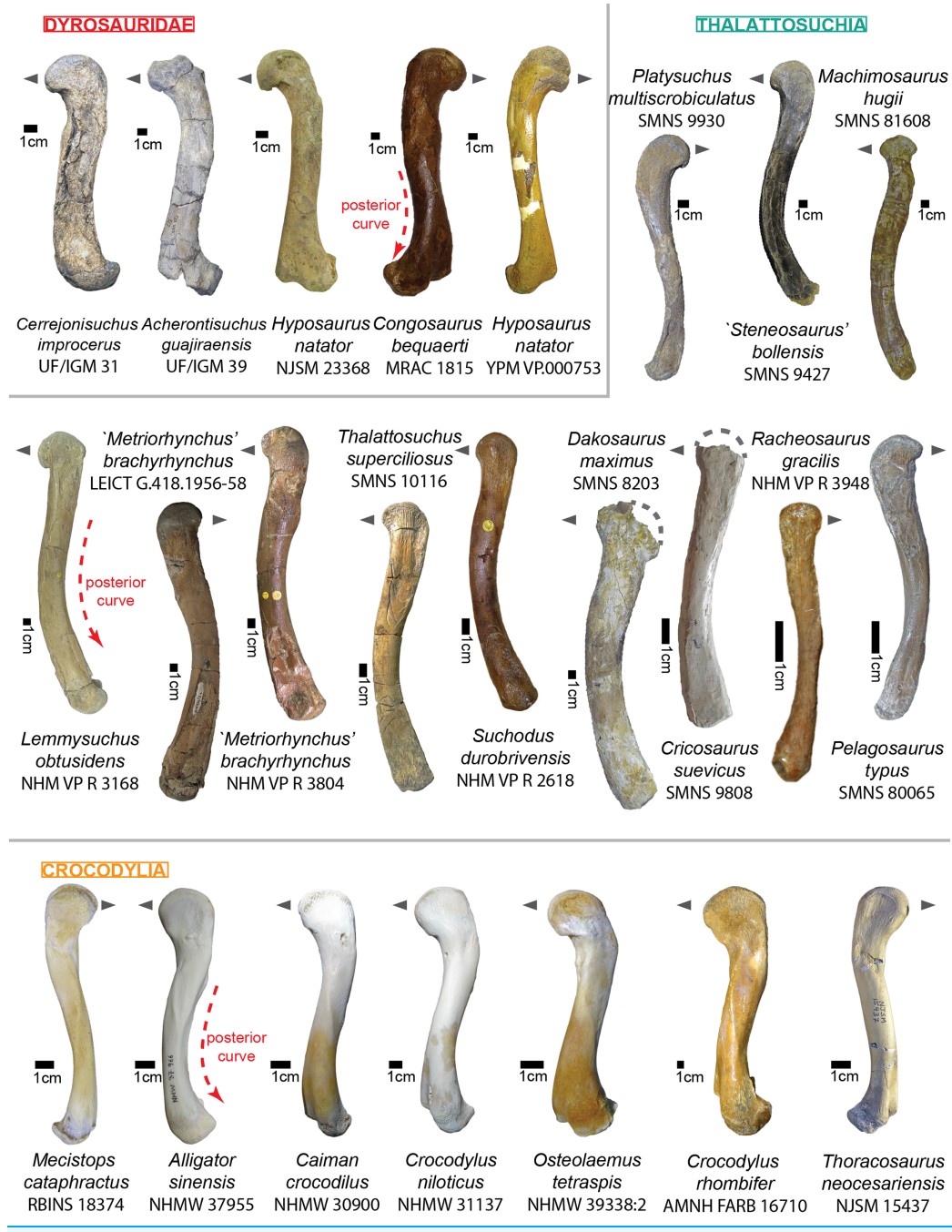

**Figure 12 Comparative figure of crocodyliform femora in dorsal view.** *Metriorhynchus moreli* is now referred to *Thalattosuchus superciliosus*.               

the femur is convex for *Hyposaurus natator* and *Acherontisuchus guajiraensis*, whereas it is straight for *Cerrejonisuchus improcerus*). A fragmentary femur (DGM 803-R) (from the Upper Cretaceous of New Jersey referred to *Hyposaurus* sp. according to *de Souza et al. (2019)*) which is to be considered Hyposaurinae indet. presumably from Danian following *Jouve et al. (2020)*, also possesses a major anterior protrusion whose intensity is situated between *Hyposaurus natator* (NJSM 23368), Hyposaurinae indet. (AMNH FARB

19204, 2202) and *Acherontisuchus guajiraensis* (UF/IGM 39), so that intraspecific variations cannot be ruled out to explain this phenomenon as well. Just like *Hyposaurus natator*, *Congosaurus bequaerti* represents another pole: its femora (MRAC 1815 & 1817) show an even lesser protruding femoral head than *Cerrejonisuchus improcerus* (UF/IGM 31) along with a lesser sigmoidal shaft (altogether giving the look of a straighter femur for *Congosaurus bequaerti*).

An example of a wide, convex and strongly protruding femoral head can be found within *Crocodylus rhombifer* (AMNH FARB 16710), which also happens to be the most terrestrial modern crocodylian.

The distal capitula of the femur indeed appear, as they are not complete, strongly asymmetrical (see Fig. 11 Femur D): the posterior capitulum is greater (in length and width) than the anterior one (base used as reference), but also extends more distally (which is a trait one would expect from a sprawling animal (*Nyakatura et al., 2019*)). Both capitula are well developed compared to the femoral head, and also well curved distally into a half-circle (of about 18 mm each in height). The latter could presumably gives enough room for an extended range of motion with the tibia. Likewise, *Congosaurus bequaerti* (MRAC 1817) and *Hyposaurus natator* (NJSM 23368) possess high and rounded distal capitula. However in *Congosaurus* the anterior capitulum is longer but thinner than the posterior one, and in *Hyposaurus* the posterior capitulum is actually shorter than the anterior one, but wider. Strangely, *Crocodylus rhombifer* (AMNH FARB 16710, one of the most terrestrial crocodylian (*Morgan & Albury, 2013*)) shows a distribution similar to *Cerrejonisuchus improcerus* between its capitula (i.e. large posterior capitulum, short anterior one). Unfortunately, the femur of *Acherontisuchus guajiraensis* (another basal dyrosaurid from the same locality (*Hastings, Bloch & Jaramillo, 2011*)) is missing the anterior capitulum and does not allow any comparisons. Another major feature of *Cerrejonisuchus improcerus* (UF/IGM 31) is the absence of a well defined intercapitular fossa separating the capitula dorsally while showing a clear sulcus running all the way dorsoventrally between the capitula. Even though the bone is broken in this area (*i.e* the dorsal-most part of the distal articular surface is missing), there are no evidence on the surface of the femur indicating an deep dorsal indentation of the articular surface. Therefore, the trochlea of *Cerrejonisuchus improcerus* may have been slightly dorsally positioned, which is a trait found namely in the derived dyrosaurid *Hyposaurus natator* (e.g. YPM VP.000753, holotype of *Hyposaurus natator oweni* as proposed by *Troxell (1925)*, and a junior synonym of *Hyposaurus natator* according to *Parris (1986)*; NJSM 23368). There are other crocodilyforms who have their trochlea more centrally positioned, and thus show well developed dorsal and ventral intercapitular fossa e.g.: *Crocodylus rhombifer* (AMNH FARB 16710, one of the most terrestrial crocodylian (*Morgan & Albury, 2013*)); the hyposaurine *Congosaurus bequaerti* (MRAC 1815); *Mecistops cataphractus* (RBINS 18374, one of the most aquatic crocodylian; *pers. comm.* Dr. O. Pauwels, April 2018).

The femur of *Cerrejonisuchus improcerus* measures 176 mm long (distal-proximal length), which accounts for almost 120% of the humerus' length (see Table 6). Comparatively, the hyposaurine dyrosaurids possess a greater femur: those of *Congosaurus*

**Table 7  Table showing ratios related to the 4th trochanter and the paratrochanteric fossa of several dyrosaurids.**

| Species | Paratrochanteric area (mm²) | Area trochanter/ paratrochanteric fossa | Femur length/area fossa (mm⁻¹) | Length femur/ paratrochanteric fossa |
|---|---|---|---|---|
| *Congosaurus bequaerti* (MRAC 1815) | 615.8 | 1077/615.8 = 1.748 | 299/615.8 = 0.48 | 299/37.41 = 7.99 |
| *Cerrejonisuchus improcerus* (UF/IGM31) | 169 | 422/169 = 2.49 | 176/169 = 1.04 | 176/25.32 = 6.95 |
| *A. guajiraensis* (UF/IGM 39) | 144.13 | 692/144.13 = 4.8 | 248/144 = 1.72 | 248/28.41 = 8.729 |
| *H. natator* (NJSM 23368) | 132.3 | 251/132.3 = 1.89 | 200/132.3 = 1.51 | 200/20.34 = 9.83 |

*bequaerti* (MRAC 1817 & 1815; *pers. obs.* plus *Jouve & Schwarz (2004)*) reach about 299 mm while that of *Hyposaurus natator* (NJSM 23368) measures 200 mm, thus respectively representing 103.9% and 106.3% of their own humerus' length (which is less than *Cerrejonisuchus improcerus*). *Acherontisuchus guajiraensis* (UF/IGM 39) (*Hastings, Bloch & Jaramillo, 2011*), an other basal dyrosaurid from the same formation as *Cerrejonisuchus improcerus* (Cerrejón Formation), shows a femur of intermediate size, yet far greater length than *Cerrejonisuchus improcerus*, at 48 mm. It appears evident that *Cerrejonisuchus improcerus* was a relatively small-bodied dyrosaurid, but seemingly less aquatic than *Acherontisuchus guajiraensis* according to *Hastings, Bloch & Jaramillo (2011)*.

Like in the derived dyrosaurids *Hyposaurus natator* (NJSM 23368), *Congosaurus bequaerti* (MRAC 1817), and the basal dyrosaurid *Acherontisuchus guajiraensis* (UF/IGM 39), the fourth trochanter of *Cerrejonisuchus improcerus* is mainly located on the ventral side of the bone (see Fig. 11 Femur B). It is a rather well developed feature among dyrosaurids (see also Table 7): the base area of the structure reaches about 422 mm² for *Cerrejonisuchus improcerus* (UF/IGM 31), 692 mm² for *Acherontisuchus guajiraensis* (UF/IGM 39), 251 mm² for *Hyposaurus natator* (NJSM 23368), and 1,077 mm² for *Congosaurus bequaerti* (MRAC 1815). Both the basal and the hyposaurine dyrosaurids show a depression anterior to the fourth trochanter, the paratrochanteric fossa, which either hints for a more developed *m. caudofemoralis* (*Romer, 1923*; *Schwarz-Wings, Frey & Martin, 2009*) (which who takes roots in the fossa), or a more developed *pubo-ischio-femoralis internus* (*Romer, 1923*), or both altogether sharing the fossa. *Congosaurus bequaerti* possesses by far the largest fossa (about 615.8 mm²), but *Cerrejonisuchus improcerus* shows the longest one proportionally to the total length of the femur (see Table 7).

## Tibia

The bone referred to as the left tibia (Fig. 11 Unknown) is too altered to bear any representative structures. This bone greatly differs from the right tibia both in shape and size of the proximal and distal extremities; the shaft of this bone is also much thicker than that of the right tibia. For these reasons, we believe it may not belong to *Cerrejonisuchus improcerus* (UF/IGM 31).

The right tibia of *Cerrejonisuchus improcerus* extends as far as 151.96 mm in length, which is about 86.3% of the femur's length (see Fig. 11 Femur & Tibia; see Table 6).

The tibia of *Hyposaurus natator* (NJSM 23368) measures 158 mm which is 79% of the femur's length. The right tibia of *Congosaurus bequaerti* (MRAC 1808 & 1818) is unfortunately incomplete, yet the tibia would have measured in between 240 mm and 250 mm which corresponds to 80% and 83.3% respectively (a ratio close to that of *Hyposaurus natator*). Therefore *Cerrejonisuchus improcerus* possessed a rather long tibia compared to its femur (see Fig. 11), making it in higher stature proportionally to its size.

The overall shape of the tibia of *Cerrejonisuchus improcerus* ressembles that of *Congosaurus bequaerti* (MRAC 1808 & 1818), *Hyposaurus natator* (NJSM 23368) and *Mecistops cataphractus* (RBINS 18374) (see Fig. 12): the shaft is straight and slender, the proximal extremity is wide, and the distal extremity splits into two asymmetrical condyles (see Fig. 11 Tibia). The strongest identifying feature certainly is the combination of a flat anterodorsal surface and a concave posteroventral one, both creating a wide proximal extremity. The posterior portion of the proximal articular surface slightly extends towards the shaft: this is where it partially contacts the fibula proximally. The anterior portion of the proximal articular surface is unfortunately partially broken off, but the articular surface is still not apparent on that side. Indeed, the proximal articular surface of the tibia is not flat but slightly tilted posteriorly (see Fig. 11 Tibia B & F). As the bone is strongly flattened anteroposteriorly, the actual shape of the proximal articulation is altered but still reflects the tuberosities on which the articulation with the femur took place. The distal extremity of the bone splits into two uneven condyles where the posterior condyle extends more distally than the anterior condyle (see Fig. 11 Tibia). Thus, the anterior condyle has its articular surface distally oriented while the posterior condyle is mainly posteriorly facing to meet with the corresponding surface of the fibula (which is therefore anteriorly facing).

## Fibula

The fibula of *Cerrejonisuchus improcerus* measures 140.94 mm, which is almost as long as the tibia (see Fig. 11 Tibia & Fibula; see Table 6). A similar length between the tibia and the fibula is not unexpected because they both connect to the podial elements. The global shape of the fibula resembles that of an upside-down hockey cross: a long straight shaft plus a wide and flat proximal extremity reminding of a spatula (see Fig. 11 Fibula E). This last element is highly unusual as none of the fibulae of *Hyposaurus natator* (NJSM 23368) are known to flare out so intensely (*Congosaurus bequaerti* MRAC 1814 is missing the proximal extremity). The diagenetic flattening of the bone can at best only be partially responsible. Besides the spatula-proximal end, the fibula of *Cerrejonisuchus improcerus* is vaguely similar to that of *Hyposaurus natator* (NJSM 23368) and even to those of modern crocodylians (e.g. *M. cataphractus* RBINS 18374; *C. porosus* Aquarium-Museum Liège R.G.294; *C. niloticus* NMW 31137) in possessing a thin proximal end along with a wide distal end (whose orientations always differ slightly). The distal extremity of the fibula of *Cerrejonisuchus improcerus* is triangular in section (see Fig. 11 Fibula D), with a small portion of the articular surface extending towards the shaft in anterior view to meet with the distal part of the tibia.

## Pubis

Both left and right pubic bones are preserved (see Fig. 13). They possess an overall shape moderately unusual (see Fig. 9) among distal crocodyliforms, like some thalattosuchians (e.g. 'Metriorhynchus' brachyrhynchus NHM VP R3804; Platysuchus multiscrobiculatus SMNS 9930), and crocodylians (e.g. Caiman crocodylus NHMW 30900). Yet, there are some crocodyliforms like 'Steneosaurus' bollensis SMNS 9428 that show a more similar shape to that of Cerrejonisuchus improcerus.

The pubis of Cerrejonisuchus improcerus (UF/IGM 31) also takes the shape of a distorted spatula because its distal extremity is rectangular (see Figs. 13A & 13B), and not triangular like that of Caiman crocodylus (NHMW 30900; see Fig. 14), Mecistops cataphractus (RBINS 18374; see Fig. 13J and Fig. 14) or Hyposaurus natator (NJSM 23368; see Fig. 14). There are no gradual expansion of the shaft, which is rather elongated and makes up for a little more than half the total length of the bone. The shaft is yet comparatively shorter than in Hyposaurus natator (NJSM 23368). While the lateral part of the shaft is straight, the medial part is concave thus creating a prominent peak at its intersection with the distal outline (see Figs. 13A–13D). Indeed, the medial margin of the distal extremity is rather straight, but becomes convex distally thus forming the ventral margin of the distal blade. In contrast, the dorsolateral portion of the distal blade, which connects to the shaft, is concave. There is also a short and convex surface, strictly lateral, that delimits both dorsolateral and ventromedial outlines, thus giving a rectangular look to the distal portion of the bone (see Figs. 13A & 13B). The medial margin of the pubic apron (see Fig. 13A) is the portion where both pubes met in vivo (see Fig. 13I), and is called the pubis diaphysis (Claessens & Vickaryous, 2012). The presence of an elongated diaphysis is what helps creating the distinctive shape of Cerrejonisuchus improcerus and distinguishes it from modern crocodylians (see Fig. 14), which only display a reduced diaphysis (Claessens & Vickaryous, 2012). In this way, Cerrejonisuchus improcerus appears more similar to basal crocodyliforms (Claessens & Vickaryous, 2012), and some teleosauroids (e.g. 'Steneosaurus' bollensis SMNS 9428 or Lemmysuchus obtusidens NHM VP R3168; see Fig. 14). On the posterior side of the bone (see Fig. 13B), there is a shallow notch near the proximal end of the shaft which was probably the attachment site for the pubo-ischio-femoralis externus 2. This muscle also covered most of the distal part on both sides (Romer, 1923).

## SYSTEMATIC PALEONTOLOGY

Crocodylomorpha Walker, 1970
Crocodyliformes Hay, 1930
Mesoeucrocodylia Whetstone and Whybrow, 1983
Dyrosauridae de Stefano, 1903
Cerrejonisuchus improcerus Hastings, 2010

**Type species**: Cerrejonisuchus improcerus (Hastings et al., 2010)

**Range**: Middle to late Paleocene of Colombia (Hastings et al., 2010)

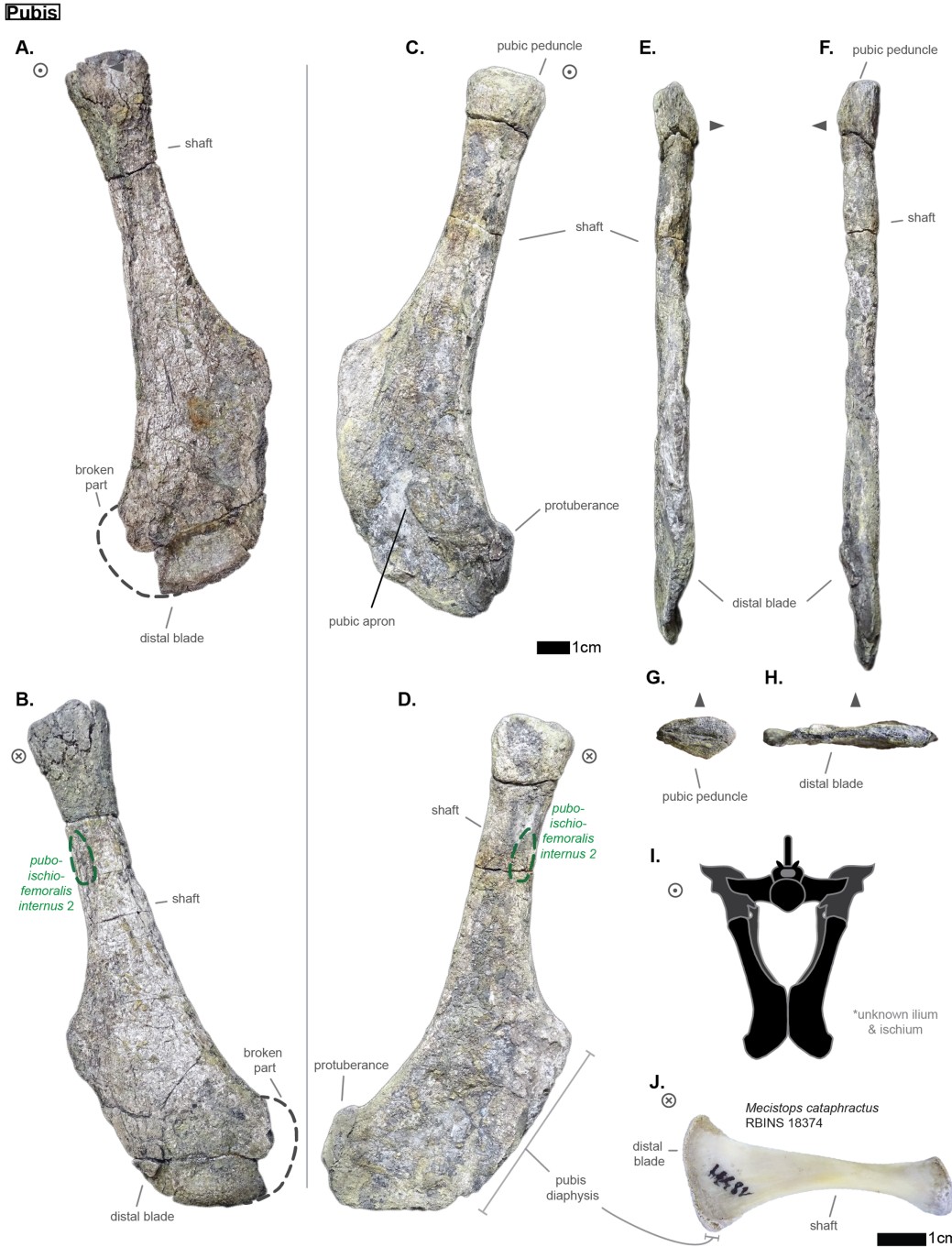

**Figure 13 Right and left pubis of *Cerrejonisuchus improcerus* UF/IGM 31.** Right pubis is partially broken and located on the left side of the line. (A) Right pubis anterior view; (B) Right pubis posterior view; (C) Left pubis anterior view; (D) Left pubis posterior view; (E) medial view; (F) lateral view; (G) proximal view; (H) distal view; (I) reconstruction of the pelvic girdle of Cerrejonisuchus improcerus in anterior view; (J) right pubis of *Mecistops cataphractus* (RBINS 18374) in posterior view (flat). Grey arrow points towards anterior.

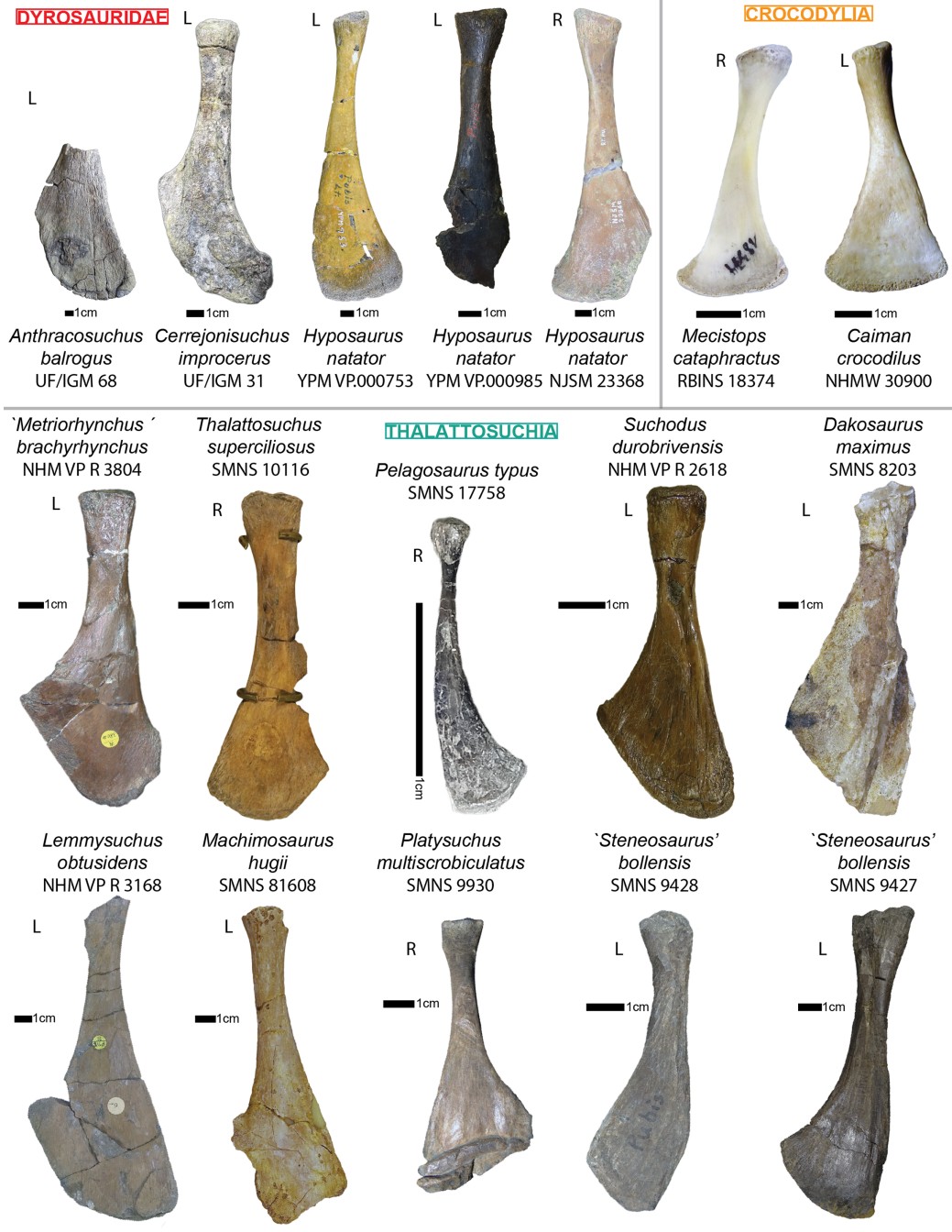

**Figure 14 Comparative figure of crocodyliform pubes in anterior view.** Right pubes are indicated by letter R. Left pubes are indicated by letter L.

**Emended diagnosis**:

We expanded the craniodental diagnosis of *Hastings et al. (2010)* with postcranial characters. *Cerrejonisuchus improcerus* (UF/IGM 31) shows these autapomorphic characters:

– Each maxillary possesses 11 teeth, and eight of those are anterior to the orbits (*Hastings et al., 2010*);

– Fibula with extended proximal fibula blade, greatly protruding from the shaft;

– Pubis with elongated, rectangular distal portion and large pubis diaphysis (rather than triangular in many crocodyliforms);

– Ulna presenting double concavity (usually single concavity in crocodyliforms);

– Proximal head of femur is round in dorsoventral views (whereas it is elliptic in hyposaurine; *de Souza et al. (2019)*) and takes the shape of a Lancet arch in anteroposterior views;

– Odontoid has an elliptic shape, with the greatest axis laterally oriented. Its height over width ratio is much smaller than hyposaurine dyrosaurids with about 0.6 (*contra* 0.8 for *Hyposaurus natator* and *Congosaurus bequaerti*);

*Cerrejonisuchus improcerus* (UF/IGM 31) shows these unique combinations of characters:

– Among Dyrosauridae, snout is the shortest with about 54–59% of the dorsal skull length (*Hastings et al., 2010*);

– Among Dyrosauridae, only one to possess a wide interfenestral bar which has a square shape in cross-section with *Chenanisuchus* (*Hastings et al., 2010*);

– Among Dyrosauridae, only one to possess a reduced fourth premaxillary tooth with *Phosphatosaurus* (and possibly *Arambourgisuchus*) (*Hastings et al., 2010*);

– In dorsal view, lacks a 'festooned' lateral margin of the snout thus differing from *Phosphatosaurus* and *Sokotosuchus* among Dyrosauridae (*Hastings et al., 2010*);

– Possesses a medio-laterally straight posterodorsal margin of the parietal, thus differing from *Hyposaurus, Rhabdognathus, Atlantosuchus*, and *Guarinisuchus* among Dyrosauridae (*Hastings et al., 2010*);

– Possesses well-developed occipital tuberosities, thus differing from *Chenanisuchus* and *Sokotosuchus* among Dyrosauridae (*Hastings et al., 2010*);

– Skull is ornamented continuously across dorsal and lateral surfaces with no interruption across sutures, and in addition the orbits are medially and dorsally placed all of which differ from *Chenanisuchus* among Dyrosauridae. The position of the orbits most closely approximates that of *Dyrosaurus* among Dyrosauridae (*Hastings et al., 2010*);

– Teeth possess straight anterior carinae rather than twisted, thus differing from *Hyposaurus natator* among Dyrosauridae (*Hastings et al., 2010*);

– Among Dyrosauridae, long zeugopodia in relation to stylopodia (zeugopodia attaining >85% of the length of the stylopodia), especially for the ulna as the opposed to 74% for *Hyposaurus natator* and *Congosaurus bequaerti*);

– Short humerus shaft with wide proximal head but poorly developed proximal tuberosities (none of the three tuberosities stand out);

– Among Dyrosauridae, humerus proximodistal length attaining less than 90% of the femoral proximodistal length (this value is >90% in all dyrosaurids for which this feature is known; *Jouve et al. (2006)*);

– Among Dyrosauridae, the mesiolateral length of the lateral process (parapophyseal and diapophyseal processes) in middle thoracics is the greatest in relation to the diameters of the centrum facets (>100%);

– The humerus possesses an extremely reduced posterior epicondyle;

– Among Dyrosauridae, thoracic ribs are short and strongly arched;

– Among Dyrosauridae, lumbar vertebrae possess a ventral keel (shared with *Hyposaurus natator* and *Congosaurus bequaerti*).

## MORPHOSPACE OCCUPATION

Dyrosauridae, Crocodylia, and Thalattosuchia are dissimilar, occupying clearly separated areas of the morphospace (see Fig. 15). Even though some taxa share similar lifestyles, the three clades are clearly separated along the first axis of the PCoA (which accounts for 23.9% of the relative eigenvalue) meaning that this axis appears strongly influenced by the phylogenetic signal. Whereas this was expected for thalattosuchians, our results indicate that dyrosaurids also have a distinctive postcranial anatomy.

The phylogenetic influence is less prominent along the second axis (which accounts for 14.2% of the relative eigenvalue): Crocodylia and Dyrosauridae are still distinct from one another but are enclosed within the range of Thalattosuchia. More precisely, dyrosaurids occupy the same range as metriorhynchoids (plus *Lemmysuchus obtusidens*) and thus cannot be simply discriminated; the hypothesis of existing convergence between those two clades cannot be entirely ruled out and need to be further investigated. For example, *Hyposaurus* has been considered to venture in the open-sea similarly to metriorhynchids, while being able to easily wander over land (*Buffetaut, 1978b*; *Denton, Dobie & Parris, 1997*). In more recent studies though, the hyposaurine lifestyle (including *Congosaurus*) was regarded as more similar to that of teleosauroids (*Schwarz, Frey & Martin, 2006*; *Schwarz-Wings, Frey & Martin, 2009*) suggesting them as ambush predators instead of pursuit predator. Within Dyrosauridae, there is a clear demarcation between *Cerrejonisuchus* and all the other dyrosaurids.

The wide range of Thalattosuchia is essentially due to the Toarcian (late Early Jurassic) teleosauroid *Platysuchus multiscrobiculatus* which is clearly separated from other thalattosuchians along the second axis (see Fig. 15). The other teleosauroid of the dataset, the Callovian *Lemmysuchus obtusidens* (*Andrews, 1909*; *Johnson et al., 2017*) is rather close to the metriorhynchoids but without being included in their convex hull. This gap in morphospace occupation between *Platysuchus multiscrobiculatus* and *Lemmysuchus obtusidens* suppports the idea proposed by *Foffa et al. (2019)* in which Teleosauroidea must be split in two subclades: 'T-subclade' and 'S-subclade', grouping *Mycterosuchus nasutus, Aelodon priscus, Bathysuchus megarhinus, Teleosaurus cadomensis, Platysuchus multiscrobiculatus,* 'Steneosaurus' brevior plus a Chinese teleosauroid; and 'Steneosaurus' bollensis, 'Steneosaurus' leedsi, 'Steneosaurus' larteti, 'Steneosaurus' herberti, 'Steneosaurus' edwardsi, Lemmysuchus obtusidens, Machimosaurus buffetauti, Machimosaurus mosae, Machimosaurus rex,* and *Machimosaurus hugii*, respectively (*Foffa et al., 2019*).

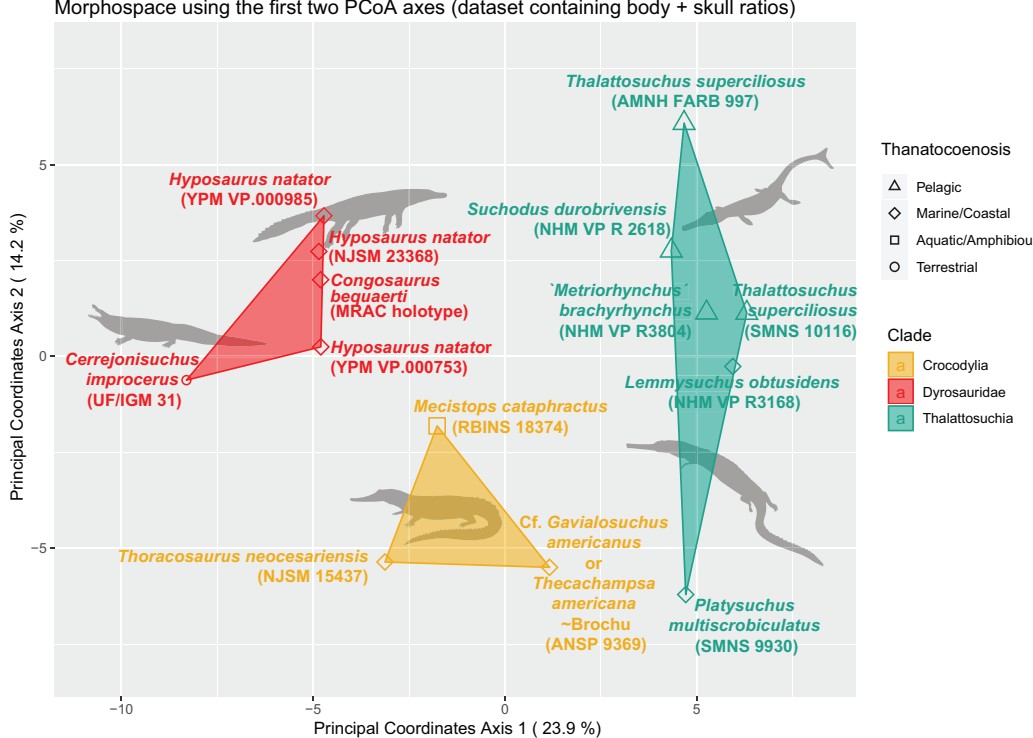

**Figure 15 Morphospace representing dissimilarity between Dyrosauridae, Crocodylia and Thalattosuchia using the first two PCoA axes, and with the complete (skull+body) ratio dataset.** Polygons demarcate families while colored symbols illustrate lifestyles. Threshold completeness: column 30%, row: 40%. Missing entries: 33.44%. Crocodylia vector by Smokeybjb (vectorized by T. Michael Keesey); Dyrosauridae vector Nobu Tamura (vectorized by Zimices) Thalattosuchia by Gareth Monger.

This chasm between *Platysuchus multiscrobiculatus* and *Lemmysuchus obtusidens* in the morphospace also backs the results of *Johnson, Young & Brusatte (2020)*, where *Platysuchus* is retrieved among Teleosauridae ('Family-T'), whereas *Lemmysuchus* in found within the new family Machimosauridae ('Family-M'). At any rate, these preliminary results for thalattosuchians suggest the existence of an expected disparity in the postcranial anatomy of teleosauroids. The wide space occupation of Thalattosuchia is not completely unexpected as thalattosuchians comprise highly differing morphologies between and even within metriorhynchoids and teleosauroids due to highly different lifestyles (*Buffetaut, 1981*; *Fernández & Gasparini, 2008*; *Young et al., 2010*; *Johnson et al., 2017*; *Wilberg, Turner & Brochu, 2019*).

Crocodylia is almost as dissimilar to Dyrosauridae than to Thalattosuchia, therefore it appears obvious that modern crocodylians cannot account for extinct lineages, at least not entirely (*Pierce, Angielczyk & Rayfield, 2009*).

Within Crocodylia, a horizontal line (understand only depending on axis 2) can be traced isolating *Mecistops* from *Gavialosuchus/Thecachampsa* and *Thoracosaurus* which would either reflect phylogeny (similarly to Thalattosuchia) between Crocodyloidae and Gavialoidae, a lifestyle demarcation, or a combination of both. In this case, phylogenetic and ecological signals are indistinguishable.

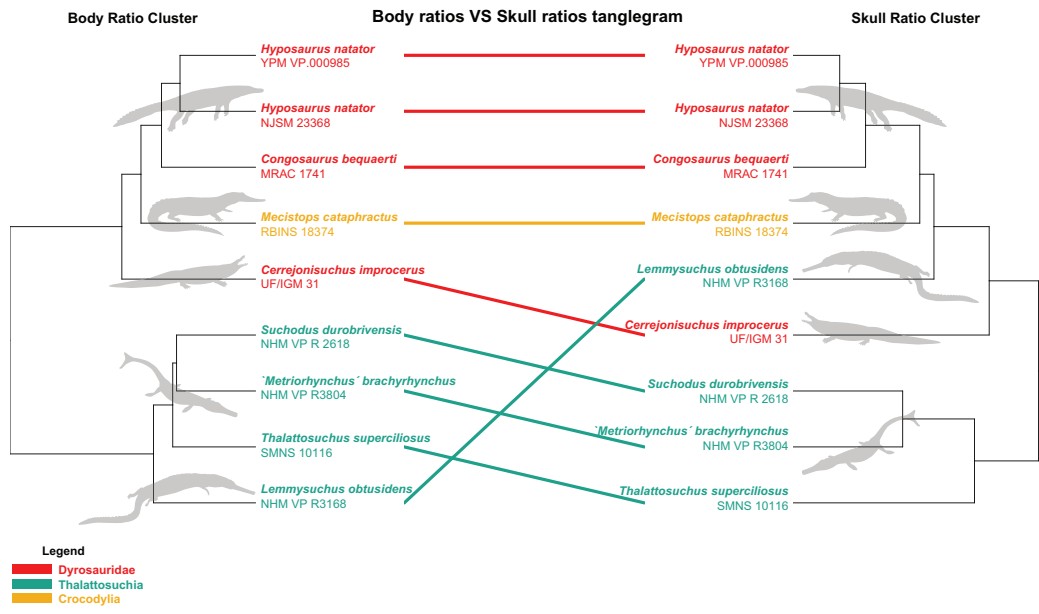

**Figure 16** **Tanglegram between the postcranial based cluster tree (left) and the cranial based cluster tree (right).** Radically differing evolutionary histories are obtained for Dyrosauridae, Crocodylia, and Thalattosuchia. Crocodylia vector by Smokeybjb (vectorized by T. Michael Keesey); Dyrosauridae vector by Nobu Tamura (vectorized by Zimices); Thalattosuchia vector by Gareth Monger.

There seems to be no modularity between the skull and body ratio based datasets as the tanglegram from Fig. 16 reveals no major differences between both cluster dendrograms. Both postcranial and cranial signals appear consistent. Though, the postcranial cluster more correctly reflects the phylogeny, thus indicating that dyrosaurids do possess a distinctive postcranial anatomy.

# DISCUSSION

## Regionalization of the cervical region in *Cerrejonisuchus* and *Congosaurus*

The variation in size of the zygapophyses, neural spine, centrum, and neural canal, along with the inclination of the zygapophyses and neural spine help assessing the flexibility in the axial skeleton of dyrosaurids compared to *Mecistops*. Low-angled zygapophyses (in relation to the coronal plane) better support mediolateral bending than dorsoventral (*Hua, 2003*; *Molnar, Pierce & Hutchinson, 2014*), and vice versa (*Langston, 1995*; *Denton, Dobie & Parris, 1997*). Large zygapophyses increase stiffness in all directions (*Molnar, Pierce & Hutchinson, 2014*). Long and vertical neural spines are negatively correlated to stiffness mediolateraly, whereas tilted neural spines would prevent mediolateral bending. Also tall neural spine mechanically limit dorsal flexion (*Molnar, Pierce & Hutchinson, 2014*). Large centra mediolaterally and anteroposteriorly also constraint mediolateral flexibility, whereas dorsoventrally tall centra inhibit dorsoventral flexion (*Molnar, Pierce & Hutchinson, 2014*). In parallel, the variation in size of the neural canal appears to indicate the relative position of the girdles.

Like *Molnar, Pierce & Hutchinson (2014)*, we took the assumption that mechanical constraints observed on modern crocodylians apply to extinct forms. Even if the prezygapohysis angles appear to suffice to limit mediolateral bending in modern crocodylians (*Hua, 2003*; *Molnar, Pierce & Hutchinson, 2014*), we decided to report angles for both zygapophyses. In *Congosaurus* and *Cerrejonisuchus*, the peak values of pre- and postzygapophysis areas are shifted more anteriorly along the axial skeleton than it is for *Mecistops* (see Fig. 2). The exact same phenomenon is observed with the zygapophysis angles of *Congosaurus* and *Hyposaurus* (see Fig. SI 3 in Supplemental). Therefore, it seems that the regionalization of the vertebral column of *Cerrejonisuchus*, and hyposaurine dyrosaurids was different from *Mecistops* and presumably other crocodylians. A rise in zygapophysis area could pinpoint increased stiffness in those portions of the axial skeleton, as observed for *Crocodylus niloticus* (*Molnar, Pierce & Hutchinson, 2014*). Additionally, the increase in zygapophysis area in *Congosaurus bequaerti*, which would increase stiffness in any orientation, coincides with a decrease in the inclination (in relation to the coronal plane) of the said zygapophysis, presumably resulting in a greater mediolateral, but not dorsoventral, flexibility (*Molnar, Pierce & Hutchinson, 2014*). In *Crocodylus niloticus* (*Molnar, Pierce & Hutchinson, 2014*) for example, vertically oriented zygapophyses (i.e. a high inclination angle) are known to enable greater dorsoventral flexion (*Hua, 2003*; *Molnar, Pierce & Hutchinson, 2014*). In parallel, as observed for *Hyposaurus*, the high inclination angle of the zygapophyseal facets with the coronal plane would inhibit increased lateral flexion, but not vertical flexion (*Langston, 1995*; *Denton, Dobie & Parris, 1997*). Yet, in *Congosaurus* the mediolaterally stiffest portion of the cervicothoracic region appears limited to the last cervicals (C8–C9, see Fig. SI 3 in Supplemental Information), and does not encompass the anterior thoracics as for *Hyposaurus* (see also (*Langston, 1995*)). Based on thoracics only, *Cerrejonisuchus* seems to follow the same trend as *Congosaurus*. Conversely, there is no variation in the neck of *Mecistops* (see Fig. SI 3 in Supplemental Information). This suggests the existence of difference in flexibility among hyposaurine taxa, with *Hyposaurus* presumably possessing more flexibility at the base of its neck. To sum up, the stiffness in the neck of dyrosaurids increases posteriorly in all directions, which probably prevented them from performing the dorsoventral shaking of brevi- and mesorostrine crocodylians during prey capture (*Hua, 2003*; *Grigg & Kirshner, 2015*). As the dorsal bending was more restricted than the lateral one, dyrosaurids probably used lateral shaking of the head instead (like *Gavialis gangeticus*; *Hua (2003)*), which corroborates the suggestion of *Schwarz-Wings, Frey & Martin (2009)*. The 'death roll' behavior has recently been witnessed in almost all extant crocodylian species, whether as a defense or predation mechanism (*Drumheller, Darlington & Vliet, 2019*). As this behavior is present across the different families, and appears to be independent from morphology and diet (*Drumheller, Darlington & Vliet, 2019*), it is possible that dyrosaurids could perform this behavior as well. However, in the absence of osteological correlates, it is an inference we will not make here.

In crocodylians, the height of the neural spine is negatively correlated to stiffness in lateral direction as opposed to the centrum width, and centrum length (*Molnar, Pierce & Hutchinson, 2014*). All dyrosaurids show two major peaks in the relative length of the

neural spine: in the posteriormost cervicals and anteriormost caudals (see Fig. SI 4). In contrast, the neural spines of *Mecistops* remain relatively constant, with a peak present in the first thoracics. Thus, there is an anterior shift of the region containing the highest neural spine in Dyrosauridae compared to Crocodylia, as it was the case for pre- and postzygapophysis areas. *Congosaurus* strongly stands out from the other dyrosaurids by possessing the greatest amplitude and steepest increase, but also the largest absolute values. The greater length of the neural spines locally create epaxial muscles with high-oval sagittal sections (*Schwarz-Wings, Frey & Martin, 2009*), and allow greater mediolateral bending of the vertebral column by increasing the leverage potential of the said epaxial muscles (*Molnar, Pierce & Hutchinson, 2014*). In the case of dyrosaurids, the large size attained by the neural spine putatively limited the dorsal bending of the vertebral column for mechanical reasons. In addition, both peaks of pre- and postzygapophysis areas and angles (indicating vertical orientation) are positively correlated with the neural spine length in the cervical region. The relatively short diapophyseal and parapophyseal processes (see Fig. 1) probably did not allow enough leverage to counter the bending restrictions imposed by the size and inclination of the pre- and postzygapophysis. For these reasons, the base of the neck in dyrosaurids appears to have been stiffer in the mediolateral direction compared to its surroundings.

The mediolateral width of the centrum (see Fig. SI 5 in Supplemental Information) is more or less constant throughout the cervical region of *Mecistops*. A different relationship is observed for *Congosaurus*, *Hyposaurus*, and *Cerrejonisuchus*: where it shows an intense increase in the cervical region. The intensity of the width variation strongly differs from *Mecistops* for all dyrosaurids, and is the highest for *Congosaurus*. Because an increase in centrum width increases mediolateral stiffness (*Molnar, Pierce & Hutchinson, 2014*), dyrosaurids show more mediolaterally flexible necks (especially compared to the thoracic region).

In crocodylians, the height of the centrum is positively correlated to stiffness in the dorsoventral direction (*Molnar, Pierce & Hutchinson, 2014*). Also, the variation of centrum height (see Fig. SI 6) closely mirrors the variation of centrum width (see Fig. SI 5) throughout the axial skeleton. In dyrosaurids, there is a substantial variation of height throughout the neck as opposed to the crocodylian *Mecistops*. This implies a greater gradient of stiffness in the dorsoventral plane between the base and extremity of the neck, so that the intervertebral joints of the anterior portion of the neck were more flexible dorsoventrally than the base.

The change of the neural canal dimensions throughout the axial skeleton of *Congosaurus* closely mirrors that of *Mecistops* (see Fig. 4), which is correlated to the position of the girdles. Yet, the cervicothoracic transition is smoother in *Congosaurus* than *Mecistops* because the cervicals of *Congosaurus* start off with greater area values for their neural canal. This may reflect greater cerebral spinal fluid irrigation (which protects the spinal cord from injury) of the head of *Congosaurus* compared to modern crocodylians, which supports the hypothesis of a 'heavy head' for longirostrine dyrosaurids (*Buffetaut, 1979*; *Storrs, 1986*).

### Regionalization of the thoracolumbar region in *Cerrejonisuchus* and *Congosaurus*

The osteodermal shield of hyposaurine dyrosaurids is known to have limited the dorsal flexibility (but not the ventral and mediolateral flexibility) of the trunk anteriorly, while the posterior portion was less restricted (*Schwarz-Wings, Frey & Martin, 2009*). This dorsoventral stiffness, and mediolateral flexibility, is also reflected in the low angles of the zygapophyses in the anterior portion of the thoracic region of *Cerrejonisuchus, Congosaurus*, and *Hyposaurus* (Th'1' through Th'4', see Fig. SI 3 in Supplemental Information). Following this, the zygapophyseal angles increase posteriorly enabling more dorsoventral flexibility but less mediolateral movement. The high inclination angle of the zygapohyseal facets with the coronal plane at the lumbosacral region of *Congosaurus* reflects an increased mediolateral stiffness in that region too. The overall mediolateral flexibility of the trunk is also made possible thanks to the dorsally and ventrally limited extensions of the osteodermal shield of dyrosaurids (*Schwarz-Wings, Frey & Martin, 2009*), which also presumably allowed lateral undulations during swimming (*Schwarz-Wings, Frey & Martin, 2009*; *Hastings, Bloch & Jaramillo, 2014*).

The variation of the inclination of the neural spine throughout the vertebral column follows the same general trend in Hyposaurines, *Cerrejonisuchus*, and *Mecistops*: differences between *Mecistops* and dyrosaurids are found in the neck, the last thoracics, and the first caudals where the neural spine is more erected for dyrosaurids. These variations hint once more at the existence of distinct regionalizations between the axial skeleton of Dyrosauridae and Crocodylia. *Hyposaurus natator* stands out with its less constrained cervical and lumbar regions compared to other dyrosaurids and *Mecistops*. Yet, all display a reduced flexibility within the first thoracics (i.e. inclined neural spine, inclined zygapophysis, and large zygapophysis), with dyrosaurids showing a steeper decline compared to *Mecistops*. This area possesses the lowest neural spine angles across all specimens, and corresponds to the attachment of the thoracic girdle. This induced joint stiffness mirrors that of the sacral region, and is necessary to weld in place the bony girdle (and muscle mass). Moreover, increased rigidity in this very region probably helped sustain the vertebral column (i.e. prevent lateral undulation) during episodes requiring the active use of forelimb for locomotion, to reallocate ground force responses similarly to the lumbosacral region. In dyrosaurids, and more specifically in hyposaurines (*a.k.a.* where it has been actually studied), the anterior portion of the trunk is encased in a rigid osteodermal shield which provided both (dorsoventral) stiffness and broad support (*Schwarz-Wings, Frey & Martin, 2009*). The low neural spine angle in this case reinforces the stiffness of the anterior portion of the trunk and base of the neck in dyrosaurids. In the meantime, the overall high inclination angle of the neural spine in dyrosaurids allows more flexibility to the vertebral column in all bending directions.

For *Mecistops*, the mediolateral width of the centrum (see Fig. SI 5 in Supplemental Information) is more or less constant throughout the thoracic region. Similarly, the width of the centrum remains constant throughout the thoracic region of *Congosaurus*, *Hyposaurus*, and *Cerrejonisuchus*, but shows a peak at the lumbosacral transition.

As mentioned in the section above, dyrosaurids showed a stiffer trunks and more flexible necks in the mediolateral direction.

Regarding the intervertebral joints of dyrosaurids, the trunk in its entirety (i.e. encompassing the thoracics and lumbars) constitutes the stiffest part of the axial skeleton. Indeed, the mediolateral rigidity increases throughout the trunk with a peak at the lumbosacral region. In parallel, the osteodermal shield presumably did not prevent further lateral bending (*Schwarz-Wings, Frey & Martin, 2009*; *Hastings, Bloch & Jaramillo, 2014*). The dorsoventral stiffness is the highest in the anterior portion of the thoracic region where it starts a constant decreasing trend posteriorly. As mentioned earlier, the anterior portion of the trunk also bears the rigid osteodermal shield, all of which make this portion the stiffest among the thoracics. This lesser dorsoventral stiffness at the lumbosacral region presumably enabled greater stride length in dyrosaurids (*Schwarz-Wings, Frey & Martin, 2009*) than what is observed in crocodylians (*Molnar, Pierce & Hutchinson, 2014*). A second peak in stiffness in all directions is centered at the lumbosacral region. The lumbosacral peak in stiffness in all directions is more evident in dyrosaurids as compared to *Mecistops*; a more rigid pelvic region can be useful for terrestrial locomotion (belly run, high walk, gallop, etc…) to hold in place the lumbosacral region (so it can move along in the dorsoventral plane instead of undulating laterally), and absorb shocks (and transmit them anteriorly) induced by both the movements of the limbs and the large tail (*Molnar, Pierce & Hutchinson, 2014*). Similarly, increased stiffness in the scapular region was probably useful during episodes of terrestrial locomotion, at least to counter forces produced by the forelimb propulsion. A high stiffness in all directions at the pelvic region also helps support large tails (*Molnar, Pierce & Hutchinson, 2014*), which is the case for dyrosaurids (see also section below). Thereby, the trunk of dyrosaurids appears to stand out from that of modern crocodylians in terms of rigidity (as suggested by *Langston (1995)*).

## Regionalization of the sacrocaudal region in *Cerrejonisuchus* and *Congosaurus*

The most distinctive feature of dyrosaurids compared to crocodylians, is their long cervical centra and short caudal centra (see Fig. 17). The exact length of the centra of *Cerrejonisuchus* is unknown and may vary from what has been collected on the seemingly less compressed vertebra. In parallel, *Hyposaurus* clearly differs from *Congosaurus* by showing a less regionalized axial skeleton, with lesser amplitudes of variation of centrum lengths. *Hyposaurus* shows relatively small peaks at each transition of the vertebral column. The greater variation in centrum length of *Congosaurus* implies changing stiffness in localized portions of the vertebral column. Yet, the centrum length of these dyrosaurids does not appear to be constant (*contra Hua (2003)*), but shows a monotonous decreasing trend posteriorly.

Similarly to the cervical region, the caudal region in dyrosaurids displays a considerable variation of the height of the centrum (which is unlike the crocodylian *Mecistops*). This implies a greater gradient of stiffness in the dorsoventral plane between the base and extremity of the tail. Indeed, the intervertebral joints of the posteriormost extremity of the

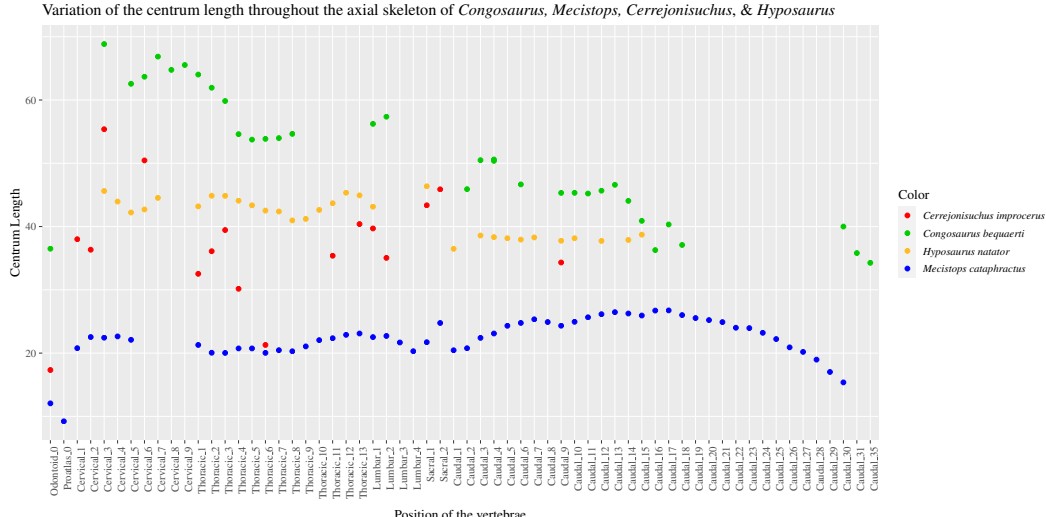

**Figure 17** Scatter plot of the centrum length variation throughout the axial skeleton of *Congosaurus bequaerti*, *Mecistops cataphractus* (RBNIS 18374), *Hyposaurus natator* (NJSM 23368), and *Cerrejonisuchus improcerus* (UF/IGM 31). Position of each vertebra listed on the abscissa axis.

tail were more flexible than its base. Regarding the mediolateral width of the centrum, the caudal centra of *Congosaurus* and *Hyposaurus* strongly decrease in size posteriorly (with *Congosaurus* displaying the greatest variation; see Fig. SI 5 in Supplemental Information), while the decrease is more feeble for *Mecistops* in this region. Because mediolateral stiffness increases with centrum width (*Molnar, Pierce & Hutchinson, 2014*), dyrosaurids had a relatively flexible tail in the mediolateral direction.

The stiffness of the tail also decreases posteriorly in all directions except for the long and inclined neural spines (see Fig. SI 4) of the last caudals (see Fig. SI 13). These observations are consistent with the hypothesis of a steering utility for the posteriormost portion of the tail (*Schwarz-Wings, Frey & Martin, 2009*). The tail of *Congosaurus* was relatively inflexible mediolaterally due to the high inclination angle of the zygapohyseal facets (even greater than the cervicals see Fig. SI 3 in Supplemental Information), with a more flexible base contrasting with *Mecistops*. Indeed, after a small depression at the lumbocaudal junction, the zygapophyseal angles drastically increase posteriorly with a small depression around the middle of the tail. Yet, the drop in area of the zygapophyseal facets at the extremity of the tail of *Congosaurus* and *Mecistops* indicates a loss of stiffness at the end of the tail, which led *Schwarz-Wings, Frey & Martin (2009)* to suggest a steering utility, like modern crocodylians, for the extremity of the tail of hyposaurine dyrosaurids. Indeed, a vertical orientation of the prezygapophysis is considered to enable greater dorsoventral flexibility than mediolateral (*Molnar, Pierce & Hutchinson, 2014*), which means that the base of the tail was potentially less stiff mediolaterally than its extremity. Besides, *Congosaurus* displays high neural spines throughout its tail, whose inclination becomes restricting for mediolateral bending towards the end of the tail. In crocodylians, the height of the neural spine is known to enable mediolateral flexion, especially if combined with lateral processes that enable the development of long epaxial

muscles, which in return increase the leverage potential and thus mediolateral bending (*Molnar, Pierce & Hutchinson, 2014*). The osteological observations actually suffice (*Molnar, Pierce & Hutchinson, 2014*) to indicate that *Congosaurus* had a laterally stiff but powerful tail. This corroborates the conclusion obtained from muscular reconstruction (*Schwarz-Wings, Frey & Martin, 2009*). As more intense force needs to be allocated to a tight vertebral column to bend it, it leads to higher undulatory frequency, and thus a faster swimming speed (*Molnar, Pierce & Hutchinson, 2014*). This observation supports the interpretations of *Schwarz, Frey & Martin (2006)*, *Schwarz-Wings, Frey & Martin (2009)*, as a tail propelled method (or a paraxial and hybrid swimming for younger individuals (*Schwarz-Wings, Frey & Martin, 2009*)) was most probably a predominant form of swimming for *Congosaurus* (which is also proposed for *Hyposaurus* (*Denton, Dobie & Parris, 1997*)). Yet, the body of dyrosaurids presumably undulated more during swimming (suggested by the lack of an extensive lateral osteodermal shield and axial muscle reconstruction; *Schwarz-Wings, Frey & Martin, 2009*; *Hastings, Bloch & Jaramillo (2014)*) than that of modern crocodylians, for which the trunk only slightly undulates during slow swimming motion but remains stiff during sustained swimming (*Hua, 2003*; *Grigg & Kirshner, 2015*).

In conclusion, dyrosaurids possessed a relatively flexible neck in the mediolateral plane, with the posteriormost part being stiffer. The anterior portion of the trunk was stiff, mainly due to the interlocked osteodermal shields (*Schwarz-Wings, Frey & Martin, 2009*), but also due to wide centra and high neural spines which limited lateral and dorsal bendings respectively. The lumbosacral region and the anteriormost caudals were stiff as well, which helped support the large tail. The tail overall was rigid, but showed an increase in mediolateral flexibility posteriorly. In overall, dyrosaurids differ from extant crocodylians by showing a greater and more localized variation of axial rigidity, as it has been suggested by *Langston (1995)*. A similar conclusion was drawn by *Hua (2003)* for thalattosuchians only.

## The ecomorphology and possible lifestyle of *Cerrejonisuchus*

The homodont dentition of *Cerrejonisuchus* (*Hastings et al., 2010*) with labiolingually-compressed teeth, along with the presence of crenulations on the lingual side of the tooth are features that mostly resemble the dentition of terrestrial, meat-eating crocodyliforms (e.g. Sebecosuchia, Mekosuchinae) (*Turner & Calvo, 2005*). A relatively elevated skull is usually associated with those traits (*Turner & Calvo, 2005*), which is not the case with *Cerrejonisuchus*. The dentition of *Cerrejonisuchus* could represent yet another terrestrial feature adding on to the list of terrestrial evidence for this taxon based on the postcranial skeleton.

The convex mid-thoracic ribs of *Cerrejonisuchus* differ from both the high oval ribs of hyposaurines (*Schwarz-Wings, Frey & Martin, 2009*), and reclining ones of crocodylians (see Figs. 7 and 8), giving *Cerrejonisuchus* a more cylindrical trunk in transverse section, halfway between the high and low oval thoracic skeletons of hyposaurines and crocodylians respectively (*Schwarz-Wings, Frey & Martin, 2009*). In this way, the trunk of *Cerrejonisuchus* must have been more similar to that of the pholidosaurid

*Anteophtalmosuchus* (*Martin et al., 2016*) (see Fig. 8). Bodies with high oblong transverse sections, like those of hyposaurines, are often found in aquatic taxa not thriving within/ limited to a specific water tier (*Motani, 2001*; *O'Keefe et al., 2011*), which would dismiss *Cerrejonisuchus* from this lifestyle.

The appendicular skeleton of *Cerrejonisuchus* contrasts with that of hyposaurine dyrosaurids in having similarly proportioned zeugopodia and stylopodia within each limb (hyposaurines have relatively shorter zeugopodia *Denton, Dobie & Parris, 1997*; *Schwarz-Wings, Frey & Martin, 2009*; *Wilberg, Turner & Brochu, 2019*). In marine thalattosuchians, the zeugopodial elements tend to be extremely reduced: this condition is also found in the most aquatic crocodylian, *Gavialis*. Indeed, *Gavialis* possesses the shortest ulna for its humerus and shows disproportionate ulna and tibia, which are considered to reduce its terrestrial locomotor capacity (*Iijima, Kubo & Kobayashi, 2018*). Therefore, we regard the relatively elongated zeugopodia of *Cerrejonisuchus* as suggestive of frequent terrestrial locomotion.

Another feature of *Cerrejonisuchus* is the presence of disproportioned limbs where the forelimb only reaches 83% of the length of the hindlimb (*contra* 91.45% for *Hyposaurus natator* NJSM 23368). Baurusuchids, such as *Stratiotosuchus maxhechti* show a similar relationship where the total length of the forelimb (excluding the manus and pes) corresponds to 80% of that of the hindlimb, but those crocodyliforms were bipedal (*Riff & Kellner, 2011*). The difference of limb proportion of *Cerrejonisuchus* contrasts rather sharply with the similarly sized limbs of hyposaurine (*Denton, Dobie & Parris, 1997*), and must have provoked a relative imbalance in the posture of *Cerrejonisuchus*. Supposing the thoracolumbar region reached at least 400 mm in length, the vertebral column must have inclined by an angle of 5.7° to 7.6° between the scapular and pelvic girdle. Asymmetrical gaits are well-known in terrestrial crocodyliforms (*Parrish, 1987*; *Adams, 2019*), but *Cerrejonisuchus* certainly did not bear a parasagittal posture as the orientation and position of the femoral condyles and 4th trochanter make it impossible for a vertically positioned femur. Following this, the difference of length between the limbs of *Cerrejonisuchus*, and the short absolute size of the limbs likely rendered the crocodylian 'high walk' difficult to sustain over extended distances, similarly to adult crocodylians (*Grigg & Kirshner, 2015*). Furthermore, the high neural spines dyrosaurids likely prevented them from performing the gallop, a behavior mostly associated with small body size in crocodylians (*Grigg & Kirshner, 2015*). The interlocked osteodermal shield of dyrosaurids had presumably no influence over this type of locomotion like in modern crocodylians (*Salisbury & Frey, 2001*), especially if *Cerrejonisuchus* exhibited the same constraints as the osteodermal shield of *Congosaurus* (*Schwarz-Wings, Frey & Martin, 2009*), as the limited dorsal extension would still be sufficient for what is required during the galloping of modern crocodylians (*Salisbury & Frey, 2001*). In addition, the large tail of dyrosaurids presumably shifted the center of gravity more posteriorly compared to modern crocodylians in elevated tail dragging postures like the 'high walk' (*Willey et al., 2004*; *Grigg & Kirshner, 2015*), and thus induced extra weight on the pelvic area and hindlimbs.

Consequently, the hindlimb would constitute the main body mass support and propulsive force for this type of gait (similarly to *Alligator mississippiensis*; *Willey et al. (2004)*), and be required to generate important work to compensate the braking effect of the large tail (*Willey et al., 2004*) while the pelvic area would need to be sufficiently supported to redistribute ground forces (*Schwarz-Wings, Frey & Martin, 2009*). This indicates that dyrosaurids also possessed imbalanced propulsive and braking impulses across their limbs, perhaps even more than in *Alligator mississippiensis* (*Willey et al., 2004*). The small size of *Cerrejonisuchus* probably played a key role in a more terrestrial lifestyle, as large dyrosaurid individuals likely lost the ability to employ terrestrial locomotion with their mass increasing (*Schwarz-Wings, Frey & Martin, 2009*). The elongated zeugopodia of *Cerrejonisuchus* probably enabled it to move more easily on land compared to *Gavialis* (*Iijima, Kubo & Kobayashi, 2018*) or hyposaurine dyrosaurids, making *Cerrejonisuchus* one of the most terrestrial dyrosaurids while likely retaining a sprawling posture, which is notably suggested by the anteroposterior orientation of the femoral head (*contra* erect archosaurs which present a medially oriented femoral head, see *Parrish (1987)*).

Similarly to mekosuchines, the dyrosaurid humerus differs from that of crocodylians in possessing a straight shaft as well as a more anteriorly positioned and oriented deltopectoral crest (e.g. *Kambara* (*Stein et al., 2012*)). The goniopholid *Anteophtalmosuchus* further differs from those in possessing a deltopectoral crest closer to the anterior margin of the shaft than to the ventral midline of the bone, along with the proximal torsion of crocodylians (*Martin et al., 2016*). The anterior position and orientation of the deltopectoral crest increase the strength of the lever-force of adductor muscles (*Stein et al., 2012*), enabling greater propulsive effort from the forelimbs which is notably useful for terrestrial locomotion (even if the hindlimbs represent the majority of the propulsive impulse, see above) or hybrid swimming.

If, like modern crocodylians, *Cerrejonisuchus* breathed by notably involving the liver (hepatic piston pump) (*Gans & Clark, 1976*; *Uriona & Farmer, 2006*; *Uriona & Farmer, 2008*; *Claessens & Vickaryous, 2012*; *Munns et al., 2012*), then the extensive pubis diaphysis (see Fig. 13B; see Fig. 14) certainly impacted this behavior. Compared to extant crocodylian, the flexibility of movement between the pubes is probably reduced in *Cerrejonisuchus* so that the rotational movement induced by the ischiopubis muscles (*Gans & Clark, 1976*; *Uriona & Farmer, 2006*; *Uriona & Farmer, 2008*; *Claessens & Vickaryous, 2012*; *Munns et al., 2012*) may have been less fluid or more difficult to perform. In the meantime, the wide pubic blade of *Cerrejonisuchus* (see Fig. 14) could have enabled a greater propulsion of viscera (in terms of speed and volume of elements displaced). Unfortunately, the ischium is missing and it is therefore not possible to assess the potential flexibility of the puboischial joint which is the main limiting factor for the development of an hepatic piston pump (like that of extant crocodylians (*Gans & Clark, 1976*; *Uriona & Farmer, 2006*; *Uriona & Farmer, 2008*; *Claessens & Vickaryous, 2012*; *Munns et al., 2012*)). Some modern crocodylians are known to switch the hepatic piston pump on and off depending on the situation and the species. It is possible that dyrosaurids employed a different version of this mechanism.

To conclude, *Cerrejonisuchus improcerus*

## The distinctive postcranial anatomy of dyrosaurids

Dyrosaurid phylogeny overrelies on cranial and mandibular characters (*Langston, 1995*; *Jouve, 2007*; *Bronzati, Montefeltro & Langer, 2012*; *Hastings, Bloch & Jaramillo, 2014*; *Wilberg, Turner & Brochu, 2019*; *de Souza et al., 2019*): the dyrosaurid matrices used in the past 15 years have a proportion of 98.78% (81/82 in *Hastings et al., 2010*; *Hastings, Bloch & Jaramillo, 2011*, *2014*) to 100% (30/30 in *Jouve, Bouya & Amaghzaz (2005)*) of cranial characters compared to 82% in *Wilberg, Turner & Brochu (2019)* for Crocodylomorpha, 73.26–72.112% in *Johnson, Young & Brusatte (2019*, *2020)* for Crocodylomorpha, 47–62% for neoichthyosaurs (42/88–84/134) (*Fischer et al., 2016*; *Zverkov & Jacobs, 2020*), 52% in plesiosaurians (140/270) (*Benson & Druckenmiller, 2014*), 53% in turtles (187/355) (*Evers, Barrett & Benson, 2019*), 26% (125/477) for sauropod dinosaurs (*Tschopp, Mateus & Benson, 2015*), and 9.6% (20/208) for hesperornithiform birds (*Bell & Chiappe, 2015*). Clearly, previous works on Dyrosauridae almost exclusively focused on the skulls and mandibles, claiming the postcrania (and sometimes the basicranium (*Buffetaut, 1976*)) were undiagnostic because they remained constant throughout the clade (*Buffetaut, 1976*; *Buffetaut, 1978c*; *Parris, 1986*; *Storrs, 1986*; *Norell & Storrs, 1989*; *Denton, Dobie & Parris, 1997*). However, our thorough osteological investigation of *Cerrejonisuchus* has underlined several unique postcranial traits which are summed up in the emended diagnosis of *Cerrejonisuchus*. There is also postcranial osteological evidence that differentiates *Hyposaurus*, *Cerrejonisuchus*, and *Congosaurus* from one another and from other crocodyliforms, notably the shape of the femoral head (see also *de Souza et al. (2019)*), the shape of the lateral process of the cervicals and thoracics, the zeugopodial to stylopodia ratio, the overall shape of the humerus (including torsion of the shaft, size and shape of proximal and distal condyles). Dyrosaurids have diagnostic postcranial remains, and this distinction is also reflected in our multivariate analyses using morphological ratios. On our main PCoA (see Fig. 15), which is based on a dataset mixing cranial (15%) and postcranial (85%) ratios, dyrosaurids occupy a distinct portion of the morphospace. In parallel, our tanglegram revealed that postcranial data also seem somewhat more conservative, with a slightly better match with phylogeny than the cranial-restricted cluster dendrogram (see Fig. 16). Yet, both signals of the tanglegram emanating from cranial and postcranial data are consistent, reinforcing the conjecture of the existing distinctive cranial and postcranial anatomy of Dyrosauridae.

The multivariate analyses of our extensive dataset on postcranial plus cranial data revealed the presence of a demarcation between *Cerrejonisuchus* and all other dyrosaurids (see morphospaces Fig. 15, Fig. SI 1, and Fig. SI 2 from Supplemental Information). On both the main PCoA (see Fig. 15), and the Dyrosaurid restricted PCoA (see Fig. SI 2 in Supplemental Information), *Cerrejonisuchus* lies on distinct area of the morphospaces, apart from the hyposaurine dyrosaurids. This supports our suggestion that *Cerrejonisuchus* occupies a niche that is distinct from that of hyposaurines, leading to a basal dyrosaurid—hyposaurine dichotomy overlapped by a freshwater/terrestrial—marine dichotomy. Moreover, this result further backs the hypothesis of an early shift to marine

lifestyle within Dyrosauridae (*Wilberg, Turner & Brochu, 2019*). If we restrict the dataset to Dyrosauridae (Fig. SI 1 and SI 2 in Supplemental Information), the distribution of dyrosaurids adheres to the Dyrosauridae phylogeny with basal dyrosaurids (i.e. *Anthracosuchus* and *Cerrejonisuchus*) occupying distinct position from derived ones (i.e. *Hyposaurus*, *Dyrosaurus* and *Congosaurus*). Yet, our analysis of craniodental morphological data (Fig. SI 1 in Supplemental Information) splits the Hyposaurine cluster of *Schwarz-Wings, Frey & Martin (2009)*, and places *Congosaurus* close to *Dyrosaurus* instead, which complies with the results of *Jouve & Jalil (2020)*. The postcranial-restricted PCoA (Fig. SI 2 in Supplemental Information) also sets *Congosaurus* outside of the *Hyposaurus* collection while lacking other crucial specimens like *Dyrosaurus*. Hence, the postcranial anatomy of Dyrosauridae and other crocodyliforms seems to reflect accurately their phylogenetic relationships, whereas craniodental data appears more volatile (see *Wilberg, Turner & Brochu, 2019*; *Jouve & Jalil, 2020*).

Our investigation of the bivariate distribution of the morphological ratios from the main analysis (see script 'Script_parameters.r' in Supplemental Information) revealed the importance of the shape of the femur as a discriminating feature between clades Crocodylia, Thalattosuchia and Dyrosauridae. More precisely, it is the degree of curvature and the location of this curve along the proximodistal axis of the bone that appear decisive in compartmenting the different clades, along with the proximodistal length of the femur and the thickness of the bone at the 4th trochanter. Theses differences reflect its distinct modes of locomotion. Thalattosuchia, Dyrosauridae, and Crocodylia are known to have thrived in different environments, sometimes overlapping, and have colonized distinct ecological niches leading to particularities echoed in femoral differences. Indeed, the shape of the femur, just like the skull and mandible, is a mixture of inherited and newly evolved morphologies, thus reworking phylogenetic and functional signals.

## CONCLUSIONS

*Cerrejonisuchus improcerus* possesses numerous postcranial morphological traits that differ from other dyrosaurids and crocodyliforms. Those traits, notably a fibula with an extended proximal fibula blade, a pubis with an elongated and rectangular distal portion, an ulna presenting a double concavity, and an elliptic-shaped odontoid, form a new set of features that expands the diagnosis of this taxon, which was previously limited to craniodental features. We reveal *Cerrejonisuchus improcerus* is also characterized by a suite of traits that strongly suggest a terrestrial—semi-aquatic lifestyle for this small-sized dyrosaurid: comparatively long zeugopodia (zeugopodia attaining >85% of the length of the stylopodia), a short humerus (less than 90% of the femoral proximodistal length), an anteriorly positioned deltopectoral crest, a large lateral process in thoracics, and short and strongly arched thoracic ribs. Finally, our osteological analyses of *Cerrejonisuchus improcerus* and hyposaurines also hint at a new distinctive feature of Dyrosauridae that requires further investigation among the other taxa: the presence of ventral keels on the posterior cervicals and lumbars. *Cerrejonisuchus* and hyposaurine dyrosaurids possess distinct regions along the axial skeleton, which can be identified following the variation of vertebral features (such as the inclination of the pre- and postzygapophysis, the size of the

pre- and postzygapophysis, the centrum dimensions, and the inclination of the neural spine). These regions, and the distribution of rigid and flexible areas are not always located in the same position of the axial skeleton, and often differ from what is observed for the modern crocodylian *Mecistops cataphractus*. This suggests the existence of different regionalization and flexibility between the axial skeletons of Dyrosauridae and Crocodylia. We also highlight features in hyposaurine dyrosaurids that are shared with modern crocodylians, such as reduced zeugopodia (proportionally to corresponding stylopodia) and similarly sized stylopodia. Those traits may also hint at less terrestrial habits among hyposaurine dyrosaurids. Multivariate analysis of our extensive morphological dataset describing the anatomy of exemplar dyrosaurids, thalattosuchians, and crocodylians reveals that dyrosaurids possess a distinctive postcranial anatomy among crocodyliformes, indicating that crocodylians cannot be used as functional surrogate for the former.

## ABBREVIATIONS

| | |
|---|---|
| **AMNH** | New York: American Museum of Natural History, USA |
| **ANSP** | Philadelphia: Academy of Natural Sciences Drexel University, USA |
| **DGM** | Brazil: Divisão de Geologia e Mineralogia, Departamento Nacional da Produção Mineral, Brazil |
| **MRAC** | Tervuren: Musée Royal de l'Afrique Centrale, Belgium |
| **NHM** | London: Natural History Museum, England |
| **NJSM** | Trenton: New Jersey State Museum, USA |
| **NMI** | Dublin: Natural History Museum of Ireland, Republic of Ireland |
| **RBINS** | Brussels: Royal Belgian Institute of Natural Sciences, Belgium |
| **SMNS** | Stuttgart: Staatliches Museum für Naturkunde, Germany |
| **UF/IGM** | Gainesville: **UF**, Florida Museum of Natural History, University of Florida, USA/**IGM**, Museo Geológico, at the Instituto Nacional de Investigaciones en Geociencias, Minería y Quimica, Bogotá, Colombia |
| **YPM** | New Haven: Yale Peabody Museum, USA |

## ACKNOWLEDGEMENTS

We would like to thank all the museum staff for smoothly granting us the access of the dyrosaurid collections. We thank Dr. Jonathan Bloch, Dr. Richard Hulbert, and the rest of the staff at the Florida Museum (Florida Museum, Gainesville, USA); Dr. David Parris and Dr. Dana Ehret (New Jersey State Museum, Trenton, USA); Dr. Daniel Brinkman (Yale Peabody Museum, New Haven, USA); Dr. Mark Norell and Dr. Carl Mehling (American Museum of Natural History, New York, USA); Dr. Ned Gilmore (Academy of Natural Sciences of Drexel University, Philadelphia, USA); Dr. Susannah Maidment (Natural History Museum, London, UK) for the help and care. We thank Wayne Callahan for sharing his personal data on the NJSM 23368 specimen, as well as Dr. Rodrigo Pellegrini for the cast humeri and femora of the NJSM 23368 specimen. We warmly thank Prof. Michelle Stocker and one anonymous reviewer for their constructive remarks and criticisms which helped improve our paper.

### Funding

This work was supported by the Fonds de la Recherche Scientifique (F.R.S.-FNRS) (Grant MIS F.4511.19). The funders had no role in study design, data collection and analysis, decision to publish, or preparation of the manuscript.

### Grant Disclosures

The following grant information was disclosed by the authors:
Fonds de la Recherche Scientifique (F.R.S.-FNRS): MIS F.4511.19.

### Competing Interests

The authors declare that they have no competing interests.

### Author Contributions

- Isaure Scavezzoni conceived and designed the experiments, performed the experiments, analyzed the data, prepared figures and/or tables, authored or reviewed drafts of the paper, and approved the final draft.
- Valentin Fischer conceived and designed the experiments, analyzed the data, authored or reviewed drafts of the paper, and approved the final draft.

### Data Availability

The datasets, code files, and Supplementary Information are available in the Supplemental Files.

### Supplemental Information

Supplemental information for this article can be found online at http://dx.doi.org/10.7717/peerj.11222#supplemental-information.

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
