# Peer review of "The postcranial skeleton of Cerrejonisuchus improcerus (Crocodyliformes: Dyrosauridae) and the unusual anatomy of dyrosaurids"

_PeerJ, doi:10.7717/peerj.11222_

## Round 0.1 · original submission · Major Revisions

The two reviewers have detailed but very constructive comments on the paper that will help to revise it; and which cover many points from the structure of the MS and its central questions to interpretations of function, descriptions/images of morphology and more. Please address all points individually in a point-by-point response in a timely fashion. Thank you.

Reviewer 1 ·

Basic reporting

The topic is ambitious, but it is “self contained” and the results are relevant to the hypothesis. A thorough analysis of postcranial characters is relatively rare and, to me, welcome, especially where it can shed light on locomotor mode and ecology. However, there are significant problems with the reporting that need to be addressed. I will break these into text and figures/tables.
Text:
1. Interpretation of literature. Unfortunately, in several cases the literature is misinterpreted or misused:
Line 1206: “In crocodylians, the height of neural spine is positively correlated to stiffness in lateral direction… (Molnar et al. 2014)” This was a NEGATIVE correlation. The mistake is perpetuated in the rest of the paragraph and also in line 1293.
1356: “the interlocked osteodermal shield of dyrosaurids likely prevented them from performing the gallop” – this statement is not supported by the literature. Several references state that interlocking osteoderms would not have prevented galloping:
Salisbury, S. W, and E.F. Frey. “A Biomechanical Transformation Model for the Evolution of Semi-Spheroidal Articulations between Adjoining Vertebral Bodies in Crocodilians.” In Crocodilian Biology and Evolution, 85–134. Chipping Norton, Australia: Surry Beatty & Sons, 2001.
Frey, E. “The Carrying System of Crocodilians: A Biomechanical and Phylogenetic Analysis.” Stuttg Beitr Naturk A 426 (1988): 1–60.
2. Context of study. The introduction is extremely short and does not give enough context for the study, making it harder to understand the rationale (especially for readers who are not experts in fossil crocs). For example, what types of animal are Congosaurus and Hyposaurus, and why is their anatomy important for a paper about Cerrejonisuchus? Is there any particular importance of Cerrejonisuchus among dyrosaurids in terms of phylogeny or ecology, other than it being an early member of the group? More detail about the existing ecological hypotheses for dyrosaurids would also be helpful. Finally, and perhaps most importantly, no groundwork is laid for the assessment of whether crocodylians overlap with dyrosaurids in morphospace and/or whether they are functionally analogous. Who proposed this and what was its basis?
3. Clarity. The English is generally good, but there are some places in which the wording is confusing and the meaning is unclear. For example, what is it that “shows that basal dyrosaurids [Young et al., 2016] had already developed a massive tail.”? (line 712) Also (line 1006) What is meant by “higher on its legs”? Also (line 1258) “mediolateral rigidity reaches its maximum throughout the trunk” – maximum is the single highest point – how can it be throughout an entire body region?
4. Spelling and grammar. There are a few spelling and grammatical errors also, such as “strait” instead of straight and “convexe” instead of convex (746), “parpophysis” instead of parapophysis (line 256), “aren’t” instead of is not (725). There is an extra “colonized” in line 44. Line 1306: I believe you mean integration, not “irrigation”
5. Formatting of manuscript. In some places, paragraphs are preceded by bullet points for no apparent reason (196, 205, 420-432).
6. Choice of references. The article is generally well referenced (although more references are needed in the introduction – see point 2). I suggest an additional reference for vertebral function and morphology in crocodylomorphs, section 9.1:
Hua, Stephane. “Locomotion in Marine Mesosuchians (Crocodylia): The Contribution of the ‘Locomotion Profiles.’” N. Jb. Geol. Palaont. Abh., no. 227 (2003): 139–52.
Figures and tables:
The structure of the manuscript is clear and straightforward, and the figures are excellent. However, I find it hard to follow because the figures and tables are not well integrated with the text and because the tables themselves are hard to read. In addition, the logic for including some tables and figures but not others is unclear.
1. Relevance of tables and figures. The tables and figures should be carefully chosen to convey the main points of the manuscript. Although the limb proportions are discussed at some length, there is only one table on limbs (table 16), and that table only shows a few obscure ratios that are only referenced once. There should be a table showing basic limb proportions and perhaps comparisons, and it should be referenced in the paragraphs that discuss these topics (e.g., the first 3 paragraphs of section 5.1). Some tables are too short to allow comparison; for example, tables 1-3, 4-7, 8-10 could be combined to allow comparison of vertebral measurements (don’t forget units!) and figures 2-4 could be combined (as is done, very effectively, in SI), allowing the reader to compare similar measurements between regions and taxa. Finally, it would be very nice to have a figure showing a reconstruction of the entire skeleton. Likewise for the comparative vertebral morphometrics and regionalization: their discussion covers several pages, but the only supporting data is in SI.
2. Formatting of tables: While I understand that the identities of the preserved vertebrae are not known with certainty, I think it would be better to use your hypotheses of their identities on the tables and graphs instead of just numbering them in order (which is very confusing!). In addition, the labels used in the tables are not intuitive. For example, in table 1 the column for centrum height is labeled “Aa” instead of “CH” or “Centrum ht,” and row 1 is labeled “UF/IGM 31 C1” instead of simply “C1.” This is a problem throughout.
3. Integration of figures and text. The text does not refer to the figures of bones very often or very specifically. For example (lines 901-902): “The anterior and posterior protuberances, which make the top two rounded tips of the heart in ventral view, also show a dorsal depression between them that can be observed easily in proximal view.” This should refer to figure 12, and the dorsal depression should be labeled on that figure. This is just one example, but each anatomical description should refer to the relevant figure. In addition, each figure part should have a unique letter that is used to refer to it in the text (Figure 12 has two 12 a’s, etc.) Also, I would use “change” instead of evolution to describe patterns along the spine (e.g., fig. 3 caption).
4. Raw data. Morphometric data are supplied in SI in graphical format, but a table of raw numbers would be more appropriate and useful. In addition, part of the labels are cut off on the right side and some of the colors are too similar to be easily distinguished (e.g., Fig. SI 3).
5. Fig. 13f. The area of PIFI attachment is not visible – the yellow color is too light.

Experimental design

The work is original, the research question is well defined, and the manuscript will contribute meaningfully to the literature. The investigation is technically rigorous and well described. I think that the analyses are appropriate (but see point 1 below).
1. Rationale for choice of methods. Why not use a Phylogenetic Principal Components Analysis? Line 1162 says, “In this case, phylogenetic and ecological signals are indistinguishable,” suggesting that pPCA might be better. And why use dimensionless ratios as inputs rather than allowing the first component to represent size? Again, I’m not saying that the analyses are inappropriate in any way, just that alternatives should be considered.

Validity of the findings

The data seem to support the conclusions, and the underlying data are provided (except for the raw morphometric measurements, as stated above). Conclusions are well stated. However, there are some problems, most importantly with the use of supporting references but also with clarity.
1. Use of references to support conclusions. In several places, references are missing or inappropriately used. Line 955: please provide reference or rationale for claim that semicircular femoral condyles are a terrestrial feature. Also, there is confusion in lines 1148-1149 “When Pierce et al. [2009] studied thalattosuchian skull disparity using geometric morphometrics, their results showed the opposite trend.” To be clear, it is not a trend; Pierce et al. found that metriorhynchids were more distinct from other thalattosuchians than pelagosaurs. However, they used a different sample of specimens and species, which could yield different results. Also (lines 1262-1263) “This dorsoventral stiffness probably enabled greater step length [Molnar et al., 2014] in dyrosaurids compared to crocodylians.” Molnar et al. only looked at Nile crocodile vertebrae, and I don’t believe they discussed step length – more references/explanation needed. Also (lines 1366-1367) “elongated zeugopodia … making Cerrejonisuchus one of the most terrestrial dyrosaurids while likely retaining a sprawling posture [Molnar et al., 2015] – This reference deals with vertebrae and did not include dyrosaurids, so it’s unclear how it supports this statement. Please carefully check all references and make sure you are accurately representing their findings!
2. Link between conclusions and results. Going back to point 1 under figures and tables, there is no reference to figures or tables in the conclusions. This makes more difficult than necessary for readers to assess whether the conclusions are well founded. In addition, it would be very helpful to summarize the evidence for each conclusion.
3. Epicondyles vs. condyles. I’m confused about the discussion in lines 818-836 regarding “epicondyles” of the humerus. The discussion of asymmetry, especially regarding effect on ROM (832) usually relate to condyles, but “epicondyles” is used in the text and that’s what is labeled on the figures. If epicondyles is meant, please explain the functional relevance.
4. Link to original research question. “Multivariate analysis of our extensive morphological dataset describing the anatomy of exemplar dyrosaurids, thalattosuchians, and crocodylians reveals that dyrosaurids possess a distinctive postcranial anatomy among crocodyliformes, indicating that the latter cannot be used as a functional surrogate for the former.” (lines 1445-1448) I think you mean crocodylians instead of “the latter” (crocodyliformes). Also, going back to point 2 about the text, it is not possible to evaluate the implications of morphological differences between crocodylians and dyrosaurids without any context.
5. Explanation of findings. When comparative anatomy of several taxa is described I find myself getting lost – I don’t know why particular comparisons are being made or what they mean. For example, lines 955-971 compares the distal femoral condyles (“capitula”) of 8 taxa, but it is not clear how they are related or what these comparisons are meant to show.

·

Basic reporting

This manuscript includes important new data on the postcranial skeleton of dyrosaurids. These data have implications for phylogeny, function, and locomotion. Introducing some of these implications (that currently are toward the end of the Discussion) in the Introduction would help set up your study and your hypotheses better. In some places the writing needs to be more clear; this is particularly true in the vertebral section where there are a lot of measurement data that become slightly confusing. The figures are helpful in showing the specimens being discussed, but because the preservation of the material doesn't appear to be great line drawings could be helpful.

Experimental design

The authors include a lot of measurement data for these specimens and list many packages used for their analyses. It would be good to be clearer about what hypotheses are being addressed with this work prior to going into a list of packages. Perhaps those would be better listed in a table. Because one of the goals appears to be identifying taxonomically distinguishing features of the postcrania, a comparative figure or two would be very helpful to illustrate your points.

Validity of the findings

The discussion and conclusion sections of the paper are well written. I had a much better idea of the importance of this work once I got to the end of the Discussion section. The research question and hypothesis need to be more obvious earlier.

Additional comments

This paper provides a lot of very useful data on the postcrania of these dyrosaurid taxa. It's clear that the authors put a lot of time into collecting all the measurements and anatomical data for the vertebrae; those details will be very helpful to other researchers. There are some issues with the grammar and word choice that I tried to address in the marked up pdf. The Introduction can be expanded a bit to incorporate your hypotheses, and the Materials and Methods is the section of the paper that will need the most attention to increase readability. There are a lot of figures, which is good, but they could be a bit more in focus and perhaps need accompanying line drawings because the specimens appear flat. Overall, a lot of good information once the authors clean up some things.

---

## Round 0.2 · Minor Revisions

The reviewers have given helpful tips on improving the prose of the MS. Otherwise, it is looking in good shape thanks to your efforts and the prior input of the reviewers. Please pay heed to their tips and respond accordingly with your revised MS when you are able to.

Reviewer 1 ·

Basic reporting

The text and figures are much improved. I especially appreciate the new reconstruction (Fig. 9).
1) The interpretations of literature have been corrected and sufficient context is provided. A little bit of additional detail about references showing "importance" of postcrania would be helpful (line 94)
2) The introduction has been expanded and provides context and motivation for the study.
3) Clarity, spelling, and grammar are much better, though some minor wording issues remain (see attached pdf). Please avoid imprecise/casual expressions such as "anyhow" and "let's now describe ..." In addition, section 4.1 could still be better organized. Again, to avoid confusion I recommend reserving the word "evolve" for change over time.
6) Supplemental information is accessible and useful.

Experimental design

The experiment is well designed, and I accept the rationale for using PCA instead of PPCA.

Validity of the findings

Clarity and supporting references are much improved.
1) The organization and wording are much clearer, making the link between results and conclusions more obvious. However, the discussion sections are quite long and it is easy lose track of the main points. I suggest adding a short paragraph at the beginning of each section summarizing the findings and outlining the points that will be covered.
2) The rationale for comparisons with crocodiles has been explained, and comparisons with other extinct taxa are more concise and clearly reasoned.
3) I don't understand the rationale for relating neural canal dimensions to blood flow to the head (lines 1533-1544).
4) The term "bracing system" in the referenced literature refers to a combination of bones, muscles, and ligaments. Here you are only describing ribs and vertebrae, so I might replace it with a different term such as "thoracic skeleton" (Figure 8 caption, lines 886 )
5) The discussion of the influence of the tail in terrestrial locomotion is interesting and would benefit from a more precise explanation of its influence on croc locomotion (lines 1596-1597). I suggest the following reference:
Willey, J. S, A. R Biknevicius, S. M Reilly, and K. D Earls. “The Tale of the Tail: Limb Function and Locomotor Mechanics in Alligator Mississippiensis.” Journal of Experimental Biology 207, no. 3 (2004): 553–563.

Additional comments

I appreciate the careful revision. The authors have addressed all of my concerns, and the resulting manuscript is clear and convincing. I think it will make a valuable addition to the archosaur literature.
Please note: I have added minor comments to the "tracked changes" version of the manuscript (attached), and line numbers in the review refer to this version.

Annotated reviews are not available for download in order to protect the identity of reviewers who chose to remain anonymous.

·

Basic reporting

This updated version of the manuscript has much improved flow and clear English throughout. I appreciate the added sections to the Introduction- they greatly clarify the intentions of the manuscript and set the reader up to follow along through the description much more easily.

Experimental design

The hypothesis is much clearer now, and the methods have more detail.

Validity of the findings

No new comments- the authors appear to have addressed all reviewer comments associated with the validity of the findings.

Additional comments

Great job! I think you have made substantial improvements to the presentation of your work. I only have extremely minor changes that are in the attached marked up pdf. I think with those updates the manuscript is ready for acceptance. I look forward to seeing this in print and utilizing your work!

---

## Round 0.3 · accepted · Accept

I am satisfied with revisions-- well done. Congratulations on acceptance!